# FEDQUIT: ON-DEVICE FEDERATED UNLEARNING VIA A QUASI-COMPETENT VIRTUAL TEACHER

## ABSTRACT

Federated Learning (FL) enables the collaborative training of machine learning models without requiring centralized collection of user data. To comply with the right to be forgotten, FL clients should be able to request the removal of their data contributions from the global model. In this paper, we propose FedQUIT, a novel unlearning algorithm that operates directly on client devices that request to remove its contribution. Our method leverages knowledge distillation to remove the influence of the target client's data from the global model while preserving its generalization ability. FedQUIT adopts a teacher–student framework, where a modified version of the current global model serves as a virtual teacher and the local model acts as the student. The virtual teacher is constructed by adjusting the global model's outputs on forget data, penalizing the confidence assigned to the true class while preserving relationships among outputs of non-true classes, to simultaneously induce forgetting and retain useful knowledge. As a result, FedQUIT achieves unlearning without making any additional assumption over the standard FedAvg protocol. Evaluation across diverse datasets, data heterogeneity levels, and model architectures shows that FedQUIT achieves superior unlearning compared to six state-of-the-art methods, while significantly reducing cumulative communication and computational overhead relative to retraining from scratch.

## 1 INTRODUCTION

Federated Learning (FL) trains a shared global model by periodically aggregating ephemeral model updates that are locally computed by users' devices on private data (McMahan et al., 2017), avoiding the transfer and collection of unprocessed data. However, privacy regulations such as the GDPR (European Parliament & Council of the European Union, 2016) also require the enforcement of the right to be forgotten, giving users the ability to request the deletion of their personal data upon withdrawal of usage consent. As widely demonstrated in prior work (Shokri et al., 2017; Song et al., 2019; Song & Mittal, 2021), deep learning models can memorize and leak sensitive information from training data. Consequently, simply deleting the data samples to forget is insufficient, both in centralized (Golatkar et al., 2020; Xu et al., 2023) and in federated settings (Romandini et al., 2024; Liu et al., 2024). To address the latter challenge, a growing body of federated unlearning (FU) mechanisms has emerged. Early FU methods perform unlearning by leveraging stored historical model updates to estimate and subtract the contribution of the data to forget, and then recalibrate

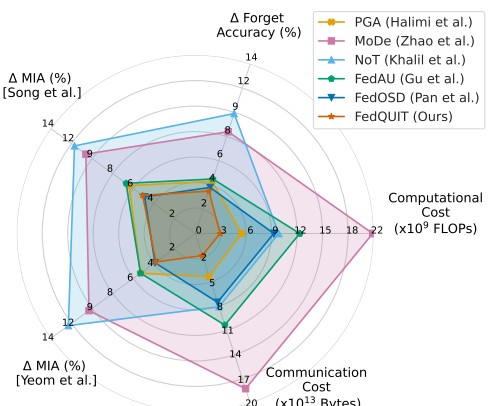

Figure 1: **FedQUIT vs. SOTA on CIFAR-100** (non-IID, ResNet-18; 10 clients, one unlearning request). Radar shows post-recovery absolute gap to *Retrain* for accuracy and MIAs on forget data, and the required computational/communication overhead. Smaller polygons indicate better unlearning; see Sec. 5.

the global model accordingly (Liu et al., 2021), often combined with rapid retraining strategies (Liu

et al., 2022; Wu et al., 2022b; Zhang et al., 2025). More recent approaches avoid retaining historical updates, since this poses storage and privacy risks, and typically adopt a two-stage design: an unlearning phase to remove the influence of the forget data, followed by a recovery phase that restores utility via training on the retained data. Nevertheless, these methods still present drawbacks: some require extensive hyperparameter tuning and can overwrite knowledge to retain, causing model erasure (Halimi et al., 2022); others use multi-round unlearning, keeping the requesting client actively involved and unable to leave the federation (Zhao et al., 2023; Pan et al., 2025), increasing coordination; some introduce auxiliary unlearning modules trained during learning (Gu et al., 2024), forcing changes to local training and model architecture; others depend on a priori identification of critical layers in the model (Zhong et al., 2025); or lack selectivity in removing client-specific contributions (Khalil et al., 2025). Furthermore, as we demonstrate in our empirical evaluations, these methods often fail to be both effective in forgetting and efficient in restoring model utility (i.e., with minimal communication and computation overhead), particularly when data are homogeneously distributed.

In this paper, we propose FedQUIT, a novel mechanism for client and sample unlearning that does not rely on historical updates or auxiliary data and requires a single on-device unlearning round while the target client is still connected (before leaving). During the unlearning phase, the model is trained to mimic the output of a *virtual teacher* on the local forget data. Concretely, our virtual teacher mirrors the output of the current global model on the forget data while selectively penalizing the true-class score and preserving the relationships among the non-true classes. Since high-confidence predictions are indicative of memorization (Yeom et al., 2018; Song et al., 2019; Song & Mittal, 2021; Ye et al., 2024), we explicitly reduce confidence in the correct label to induce forgetting, while preserving the inter-class structure among the non-true classes (Hinton et al., 2015; Phuong & Lampert, 2019; Lee et al., 2022) to preserve model utility.

As shown in Figure 1 (full results in Table 1), among state-of-the-art (SOTA) FU methods, FedQUIT consistently achieves performance closest to the gold-standard retraining baseline across multiple accuracy metrics, while significantly reducing cumulative communication and computation costs.

**Contributions.** The contributions of the paper are summarized as follows:

- We introduce FedQUIT, an efficient on-device FU method that induces forgetting via lightweight knowledge distillation (KD) from a quasi-competent virtual teacher that lowers the-true class score while preserving non-true class geometry.
- We provide theoretical insights showing that distilling to our virtual teacher (preserving the non-true geometry and penalizing only the true-class output) (i) provides a controlled forgetting signal; and (ii) induces a bounded parameter shift, so that FedAvg resumed from the unlearned model retains its standard convergence guarantees.
- We conducted evaluation across four datasets, both IID and non-IID data distributions, and three model architectures, comparing FedQUIT to six SOTA baselines. FedQUIT achieves superior unlearning: it better approximates the gold-standard retrained model in forget metrics, while significantly reducing cumulative communication and computational cost.

Our code is available at: `https://anonymous.4open.science/r/FedQUIT`.

## 2 BACKGROUND

### 2.1 FEDERATED UNLEARNING

**Federated Learning.** With $K$ clients holding private datasets $D_k$, FL optimizes $f(w) = \sum_{k=1}^{K} \frac{n_k}{n} F_k(w)$, where $w$ are global parameters, $n_k = |D_k|$, and $n = \sum_{k=1}^{K} n_k$. Federated Averaging (FedAvg) proceeds in synchronous rounds: the server broadcasts $w$ to a selected client subset, each runs $E$ local epochs and returns updates, and the server updates $w$ via weighted averaging (McMahan et al., 2017; Reddi et al., 2021).

**Federated Unlearning.** In FL, at round $t$, a target client $u$ may request removal of the contribution of a subset $D_u^{\text{forget}} \subseteq D_u$ (the *forget* data) from the global model $w_t$. FU can target *sample*, *client*, *class* (Wang et al., 2022), or *feature* unlearning (Gu et al., 2025). We focus on client unlearning ($D_u^{\text{forget}} = D_u$) and show seamless extension to sample unlearning ($D_u^{\text{forget}} \subseteq D_u$). Let $D_u^{\text{retain}} = D_u \setminus D_u^{\text{forget}}$; the retain data is $D_r = \bigcup_k D_k^{\text{retain}}$. Applying an unlearning algorithm $\mathcal{U}$ to $w_t$

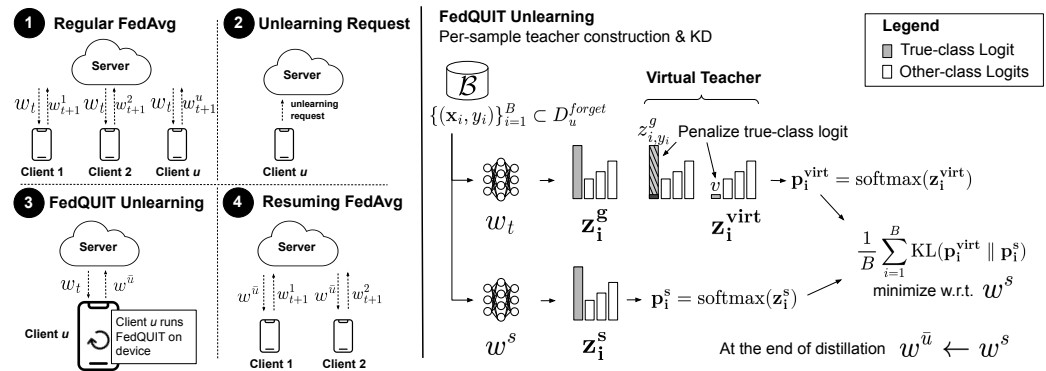

Figure 2: FedQUIT overview. **Left.** ➊ Regular training via FedAvg. ➋ Client $u$ requests unlearning. ➌ The server initiates a special round. Only client $u$ participates, and runs FedQUIT locally starting from the current global model $w_t$. Client $u$ sends back the unlearned model $w^{\bar{u}}$. ➍ Regular training resumes from $w^{\bar{u}}$. **Right.** FedQUIT unlearning phase at client $u$ with forget data $D_u$ (batch size $B$). The global model $w_t$ computes logits $\mathbf{z}_i^g$ on a forget sample $\mathbf{x}_i$. The true-class logit $\mathbf{z}_{y_i}^g$ is replaced with a fixed value $v$, yielding modified logits $\mathbf{z}_i^{\text{virt}}$. After applying softmax, the resulting distribution $\mathbf{p}_i^{\text{virt}}$ serves as a target for the student model $w_{\bar{u}}$, which is trained to mimic it by minimizing the Kullback-Leibler divergence between $\mathbf{p}_i^{\text{virt}}$ and the student's output probability $\mathbf{p}_i^s$.

produces the post-unlearning global model $w_t^{\bar{u}} = \mathcal{U}(w_t)$ ($w^{\bar{u}}$ when $t$ is clear). Typically, a *recovery phase* then runs FL rounds on $D_r$ until model utility is restored at round $t_{\text{rec}}$, producing the post recovery model $w_{t_{\text{rec}}}^{\bar{u}}$, as in (Halimi et al., 2022; Guo et al., 2024; Pan et al., 2025). The target is $w_{t_{\text{rec}}}^{\bar{u}} \approx w_t^r$, where $w_t^r$ is the model retrained from scratch without $D_u^{\text{forget}}$. *Efficiency* means minimizing cumulative costs (communication, computation, storage) of running $\mathcal{U}$ plus recovery; *efficacy* means the post recovery model is approximately indistinguishable from $w_t^r$ Liu et al. (2024).

## 2.2 Knowledge Distillation

**Notation.** We consider $C$-class classification. For an input $\mathbf{x}$ with label $y \in \{1, \ldots, C\}$, a network $h(\cdot; w)$ outputs a logit vector ("logits") $\mathbf{z} = [z_1, \ldots, z_C] \in \mathbb{R}^C$; $z_c$ is its $c$-th component. The temperature-scaled probabilities are

$$\mathbf{p}_\tau = \text{softmax}(\mathbf{z}/\tau), \quad p_{\tau,c} = \frac{\exp(z_c/\tau)}{\sum_{j=1}^C \exp(z_j/\tau)}, \quad \tau > 0,$$

with $\mathbf{p} \equiv \mathbf{p}_1$ when $\tau = 1$. The one-hot label distribution is $\mathbf{q}$ with $q_y = 1$ and $q_c = 0$ for $c \neq y$. The per-sample cross-entropy (negative log-likelihood) is $\ell(\mathbf{p}, y) = -\log p_y$.

**KD objective.** KD transfers knowledge from a fixed *teacher* to a *student* by matching softened output distributions Hinton et al. (2015); Gou et al. (2021). Let $\mathbf{z}^S, \mathbf{z}^T \in \mathbb{R}^C$ be student and teacher logits, and define $\mathbf{p}_\tau^S = \text{softmax}(\mathbf{z}^S/\tau)$ and $\mathbf{p}_\tau^T = \text{softmax}(\mathbf{z}^T/\tau)$. The canonical KD loss uses the (forward) Kullback–Leibler (KL) divergence:

$$\mathcal{L}_{\text{KD}}(\mathbf{p}_\tau^T, \mathbf{p}_\tau^S) = \tau^2 \, \text{KL}(\mathbf{p}_\tau^T \| \mathbf{p}_\tau^S).$$

**Instantiation in FedQUIT.** In our setting, the *student* is the client's local model and the *teacher* is a *virtual* distribution obtained by modifying the global model's logits on forget data (Section 3). We set $\tau = 1$.

## 3 FedQUIT: Unlearning via a Quasi-Competent Virtual Teacher

In the following, we present FedQUIT. Our method performs a single-round unlearning phase, after which regular training resumes without the requesting client. During unlearning, the target client's local model is trained to mimic a *virtual teacher* on its local forget data. The virtual teacher mirrors the outputs of the current global model, while selectively penalizing the true-class score.

---

**Algorithm 1** Local unlearning in `FedQUIT`.

---

**Input:** Global model $w_t$, forget data $D_u^{forget}$, unlearning epochs $E_u$, learning rate $\eta_u$, batch size $B$, replacement value for the true-class logit $v$

**Output:** Unlearned model parameters $w^{\bar{u}}$ to return to the server

1: Client $u$ receives global model $w_t$ from the server
2: **Initialize** student parameters $w^s \leftarrow w_t$               ▷ Start from global model
3: **for** each local unlearning epoch $e_u = 1, 2, \ldots, E_u$ **do**
4:      **for** each minibatch $\mathcal{B} = \{(\mathbf{x}_i, y_i)\}_{i=1}^{B} \subset D_u^{forget}$ **do**
5:          $\mathbf{z}^s \leftarrow h(\mathcal{B}; w^s)$              ▷ Student logits on batch, $\mathbf{z}^s \in \mathbb{R}^{B \times C}$
6:          $\mathbf{p}^s \leftarrow \text{softmax}(\mathbf{z}^s)$
7:          $\mathbf{z}^g \leftarrow h(\mathcal{B}; w_t)$             ▷ Global logits on batch, $\mathbf{z}^g \in \mathbb{R}^{B \times C}$
8:          $\mathbf{z}^{\text{virt}} \leftarrow \mathbf{z}^g$
9:          **for** each $i \in \{1, \ldots, B\}$ **do**         ▷ Modify logits for virtual teacher
10:             $\mathbf{z}^{\text{virt}}[i, y_i] \leftarrow v$
11:          $\mathbf{p}^{\text{virt}} \leftarrow \text{softmax}(\mathbf{z}^{\text{virt}})$
12:          $\mathcal{L}_{\text{KD}} \leftarrow \frac{1}{B} \sum_{i=1}^{B} \text{KL}(\mathbf{p}^{\text{virt}} \| \mathbf{p}^s)$          ▷ Temperature $\tau = 1$
13:          Compute $\nabla_{w^s} \mathcal{L}_{\text{KD}}$ and update $w^s \leftarrow \text{ClientOpt}(w^s, \nabla_{w^s} \mathcal{L}_{\text{KD}}, \eta_u)$
14: $w^{\bar{u}} \leftarrow w^s$                   ▷ Finalize unlearned snapshot
15: Client $u$ sends $w^{\bar{u}}$ back to the server

---

**FU Protocol.** Figure 2 (left) illustrates the process. The client that requested the removal of its contribution performs the unlearning routine itself during a special round, before the regular FL training resumes. During that special round, client $u$ downloads the last version of the global model $w_t$ at the time of the unlearning request, performs FedQUIT locally, and pushes the unlearned model $w^{\bar{u}}$ back to the server. Then, the server can resume the regular FL training from $w^{\bar{u}}$.

**Virtual Teacher Construction and KD.** Figure 2 (right) and Algorithm 1 outline the FedQUIT routine at the target client $u$. The local model acts as the student in a KD framework; the unlearned model returned to the server will be the snapshot of the student model after the FedQUIT routine.

At the beginning of the unlearning round $t$, the target client initializes the student with the current global weights, $w^s \leftarrow w_t$ (Alg. 1, line 2), as in standard FedAvg rounds. For a mini-batch $\{(\mathbf{x}_i, y_i)\}_{i=1}^{B} \subset D_u^{forget}$, the target client computes the logits of the student and the global model on the batch (Alg. 1, lines 5–7). The target client then constructs a virtual teacher by modifying the global model's logits (Alg. 1, lines 9–10): letting $\mathbf{z}_i^g = [z_{i,1}^g, \ldots, z_{i,C}^g]$ be the logits of $h(\cdot; w_t)$ on $\mathbf{x}_i$, define the virtual-teacher logits $\mathbf{z}_i^{\text{virt}} = [z_{i,1}^{\text{virt}}, \ldots, z_{i,C}^{\text{virt}}]$ by

$$z_{i,c}^{\text{virt}} = \begin{cases} v, & c = y_i \\ z_{i,c}^g, & c \neq y_i \end{cases} \quad \text{for } c \in \{1, \ldots, C\}, \tag{1}$$

where $v \in \mathbb{R}$ is a tunable hyperparameter (see **Choice of** $v$ for a parameter-free choice of $v$). The virtual-teacher probabilities are $\mathbf{p}_i^{\text{virt}} = \text{softmax}(\mathbf{z}_i^{\text{virt}})$ and the student probabilities are $\mathbf{p}_i^s = \text{softmax}(\mathbf{z}_i^s)$. The student mimics the virtual teacher on the forget data by minimizing

$$\mathcal{L}_{\text{KD}} = \frac{1}{B} \sum_{i=1}^{B} \text{KL}(\mathbf{p}_i^{\text{virt}} \| \mathbf{p}_i^s), \tag{2}$$

where KL denotes the Kullback–Leibler divergence (Alg. 1, lines 11–13).

**Choice of** $v$. Unless stated otherwise, we set $v_i = \min_c z_{i,c}^g$ for each forget sample $x_i$. This per-sample, data-adaptive rule is parameter-free and guarantees $p_{i,y_i}^{\text{virt}} \leq 1/C$ while preserving the non-true class probability relationships. The procedure with $v_i = \min_c z_{i,c}^g$ is detailed in Appendix Alg. 2. Ablation studies with alternative choices of $v$ are presented in Section 5.4.

**Rationale and Intuition.** In local FU, the method can access only the target client's forget data; retain data are unavailable, so a proxy is required to preserve the utility of the global model. We use the current global model as this proxy via its soft predictions and adopt KD, with the local model (student) guided by a crafted teacher derived from the global model. The intuition relies on two observations: (i) high-confidence predictions, where the true-class logit is much larger than the

others, are strong indicators of memorization (Song & Mittal, 2021); (ii) preserving the inter-class structure among the non-true classes is essential for maintaining model utility (Hinton et al., 2015). Our *quasi-competent virtual teacher* combines these insights by explicitly reducing the true-class logit to induce forgetting, while preserving the relative geometry among non-true classes so that the student does not lose utility-relevant information. Further motivation in Appendix A.

**FedAvg compliance.** In standard FedAvg (McMahan et al., 2017), the server sends the global model to active clients at each round; using it for local inference during unlearning adds no extra assumptions. Building a virtual teacher by editing its outputs has negligible cost.

**Theoretical Analysis.** We analyze three complementary aspects of FedQUIT: (i) *Forgetting signal:* moving the student toward the virtual teacher increases the cross-entropy on forget samples, with strength controlled monotonically by $v$. (ii) *Post-unlearning convergence:* KD updates induce a bounded parameter perturbation, so resuming FedAvg (without client $u$) preserves standard convergence up to a small warm-start term. (iii) *Closeness to retraining* (Theorem 3, Appendix): the divergence between the unlearned model and an ideal retrained model is governed by the introduced model perturbation, which is bounded. Assumptions and derivations appear in Appendix C.

**Lemma 1** (Penalizing the true-class logit provides a forgetting signal). *For a forget sample $(\mathbf{x}_i, y_i)$ let $\mathbf{p}_i^s = \mathrm{softmax}(\mathbf{z}_i^s)$ and per-sample cross-entropy loss $\ell_i := -\log p_{i,y_i}^s$. Define the virtual teacher $\mathbf{p}_i^{\mathrm{virt}}$ as in Eq. 1 with $v < z_{i,y_i}^g$. A single KD step*

$$\mathbf{z}_i^{s+} \leftarrow \mathbf{z}_i^s - \eta_u \, \nabla_{\mathbf{z}_i^s} \mathrm{KL}\big(\mathbf{p}_i^{\mathrm{virt}} \| \mathbf{p}_i^s\big), \quad \eta_u > 0 \text{ small},$$

*satisfies, whenever $p_{i,y_i}^s > p_{i,y_i}^{\mathrm{virt}}$ (as when the student is initialized from the global model, $\mathbf{p}_i^s \approx \mathbf{p}_i^g$),*

$$\Delta\ell_i := \ell_i^+ - \ell_i = -\log p_{i,y_i}^{s+} + \log p_{i,y_i}^s > 0.$$

*Moving the student toward FedQUIT teacher strictly increases the cross-entropy on the forget data.*

**Proposition 1** (Characterization of the forgetting signal). *For a forget sample $(x_i, y_i)$, let $S_i = \sum_{c \neq y_i} e^{z_{i,c}^g}$ and define $p_{i,y_i}^{\mathrm{virt}}(v) = \frac{e^v}{e^v + S_i}$ (non true logits fixed). Then $p_{i,y_i}^{\mathrm{virt}}(v)$ is strictly increasing in $v$, and the cross-entropy change $\Delta\ell_i(v) := \ell_i^+ - \ell_i$ (after a small KD step of size $\eta_u > 0$, initialized at the global) strictly decreases as $v$ increases. In particular, choosing $v = \min_c z_{i,c}^g$ ensures $p_{i,y_i}^{\mathrm{virt}} \leq 1/C$ and yields $\Delta\ell_i\big(\min_c z_{i,c}^g\big) \geq \eta_u \frac{C}{C-1}\big(p_{i,y_i}^g - \frac{1}{C}\big)\big(1 - p_{i,y_i}^g\big) + O(\eta_u^2)$.*

**Lemma 2** (Bounded perturbation induced by FedQUIT). *Consider the geometry preserving teacher of Eq. 1 (true-class logit set to $v$, others fixed) and let the student perform $T_u$ stochastic KD updates with learning rate $\eta_u$ on batches of size $B$, starting from $w_t$. Assume (A4) in Appendix C.2: there exists $G > 0$ such that $\|\nabla_w z(x; w)\|_{\mathrm{op}} \leq G$ for all $(x, w)$. Then the returned snapshot satisfies*

$$\|w^{\bar{u}} - w_t\| \leq \eta_u G \sqrt{2} T_u.$$

*With default choice $v_i = \min_c z_{i,c}^g$, the same bound holds and is typically tighter in practice.*

**Theorem 1** (FedAvg convergence after resuming from $w^{\bar{u}}$). *Let $F_{\backslash u}(w) = \sum_{k \neq u} \pi_k \, \mathbb{E}_{(x,y) \sim D_k}[\ell(h(x; w), y)]$ be the FL objective after removing client $u$, with mixing weights $\pi_k$. Assume (A5)–(A7) in Appendix C.2 (smoothness, bounded variance, bounded data heterogeneity). Run FedAvg on clients $\{k \neq u\}$ with any step-size schedule satisfying the standard conditions ensuring FedAvg convergence (e.g., Li et al. (2020)). When initialized at the snapshot $w^{\bar{u}}$, the FedAvg iterates converge to a stationary point of $F_{\backslash u}$ with the same rate as when initialized at $w_t$. The only effect of FedQUIT is the initial potential shift, with $\|w^{\bar{u}} - w_t\|$ bounded by Lemma 2:*

$$F_{\backslash u}(w^{\bar{u}}) - F_{\backslash u}(w_t) \leq \langle \nabla F_{\backslash u}(w_t), w^{\bar{u}} - w_t \rangle + \frac{L}{2}\|w^{\bar{u}} - w_t\|^2.$$

## 4 EXPERIMENTAL DESIGN

**Datasets and data partitioning.** We conduct experiments on federated versions of three datasets, CIFAR-10, CIFAR-100 (Krizhevsky, 2009) and CUB-200 (Welinder et al., 2010). We partition CIFAR-10 among 10 clients, and CIFAR-100 and CUB-200 among both 10 and 100 clients, and consider both IID and non-IID data distributions. We introduce data heterogeneity using distribution-based label skew, controlled via Latent Dirichlet Allocation (LDA) (Hsu et al., 2019), with concentration parameters of $\alpha = 0.3$ for CIFAR-10 and $\alpha = 0.1$ for CIFAR-100 and CUB-200.

**Model and training implementation.** We employ a standard ResNet-18 (He et al., 2016) and a vision transformer, MiT-B0 (Xie et al., 2021). Before unlearning, we train ResNet-18 from scratch for 200 FedAvg rounds while fine-tuning MiT-B0 for 50 rounds from a pre-trained checkpoint, with one local epoch per round ($E = 1$). The two 100-client configurations (CIFAR-100 with ResNet-18 and CUB-200 with MiT-B0) follow the same protocol but with 600 and 200 rounds respectively. Appendix Table 6 outlines the settings. For each setting, we conduct 10 experiments with different target clients and reported mean and std. dev. In Appendix D.13, we additionally present results for a next-token prediction task on Tiny-Shakespeare corpus (Karpathy, 2015).

**Baselines.** We use the *Retrain* baseline as gold standard for perfect unlearning, and report values for the Original model (global model before unlearning). We compare our method with six FU SOTA baselines, ❶ FedEraser (Liu et al., 2021), ❷ PGA (Halimi et al., 2022), ❸ MoDe (Zhao et al., 2023), ❹ FedAU (Gu et al., 2024), ❺ NoT (Khalil et al., 2025), ❻ FedOSD (Pan et al., 2025). For FedQUIT, we use one epoch for unlearning ($E_u = 1$), and $v_i = \min_c z_i^g$. Appendices D.2 and D.11 report the description and tuning of the baselines, and the results for unconstrained gradient ascent.

**Metrics.** We evaluate two perspectives: *efficacy* in forgetting and *efficiency* in recovering model utility. For unlearning efficacy, the most favorable outcome for an approximate unlearning method is to minimize the absolute difference from the gold-standard *Retrain* baseline. We assess:

- **Test Accuracy (Test Acc.)**: Accuracy on the dataset's held-out test set; for baselines, the recovery phase stops once Test Acc. matches the *Retrain* level, as in (Khalil et al., 2025).
- **Retain Accuracy (Retain Acc.)**: Accuracy on $D_r$.
- **Forget Accuracy (Forget Acc.)**: Accuracy on data $D_u^{forget}$.
- Two **Membership Inference Attack (MIAs)**: (1) a confidence-based predictor (Song & Mittal, 2021) and (2) a loss-based predictor (Yeom et al., 2018). See Appendix D.10.

For unlearning efficiency, we tracked the cost to recover model utility:

- **Communication, Computation, and Storage Costs**: cumulative communication (bytes) and computation (FLOPs) for the unlearning phase and the subsequent recovery phase, as in (Khalil et al., 2025). The recovery phase ends when model utility is recovered, i.e., when Test Accuracy reaches that of *Retrain*. We also report the persistent storage (bytes) required across rounds for each method (e.g., historical updates). Details in Appendix D.3.

## 5 EVALUATION

This section discusses the evaluation of FedQUIT for client unlearning. The experimental results shown here aim to answer the following research questions.

1. Does FedQUIT achieve efficient and effective *client unlearning* under the metrics in Section 4? We compare it against six state-of-the-art baselines, evaluating both *post-unlearning* and *post-recovery* performance, as well as performance under fixed cost budgets (Section 5.1). In Section 5.1 and Appendix D.6, we provide results and discussion for *sample unlearning*. In Appendix D.12, we address multiple client-unlearning requests.

2. How does the model unlearned via FedQUIT behave relative to the Original and Retrain models? We analyze predictive-entropy distributions (Section 5.2).

3. Is preserving the non-true-class structure (rank and geometry) of the teacher fundamental during distillation? We ablate the virtual-teacher design across a spectrum ranging from non-informative (fully flattened) to fully structure-preserving variants, and we additionally ablate the distillation temperature (Section 5.3).

4. How sensitive is FedQUIT to its main design choices? We study the tuning of $v$ (Section 5.4) and ablate the distillation loss used in Eq. 2 (Appendix D.7).

### 5.1 EVALUATION OF UNLEARNING EFFICACY AND EFFICIENCY

**Post-recovery Results.** Table 1 reports post-recovery performance (i.e., once each method's test accuracy matches *Retrain*). **Findings.** Under IID settings (first row), FedQUIT is the only approach

| Setting | Method | Efficacy | | | | | | Efficiency | | |
| | | Test Acc. | Retain Acc. ($\Delta\downarrow$) | Forget Acc. ($\Delta\downarrow$) | MIA$_{[Song]}$ ($\Delta\downarrow$) | MIA$_{[Yeom]}$ ($\Delta\downarrow$) | Avg. Gap ($\downarrow$) | Communication Bytes ($\times\uparrow$) | Computation FLOPs ($\times\uparrow$) | Storage Bytes ($\times\uparrow$) |
|---|---|---|---|---|---|---|---|---|---|---|
| **CIFAR-100**, IID, ResNet-18, $E=1$, 10 Clients | Original | $59.9_{\pm0.0}$ | $79.9_{\pm0.1}$ | $79.8_{\pm0.7}$ | $78.2_{\pm0.6}$ | $71.4_{\pm0.6}$ | — | — | — | — |
| | Retrain | $58.3_{\pm0.5}$ | $82.7_{\pm0.8}$ | $58.0_{\pm0.7}$ | $55.6_{\pm0.7}$ | $48.8_{\pm0.6}$ | | $1.62e^{11}$ (1.0×) | $1.35e^{15}$ (1.0×) | $4.49e^{07}$ (1.0×) |
| | FedEraser | $58.4_{\pm0.5}$ | $\underline{79.5\ (3.2_{\pm0.7})}$ | $59.5\ (1.5_{\pm0.7})$ | $57.6\ (2.0_{\pm0.5})$ | $49.9\ (1.1_{\pm1.1})$ | 2.0 | $1.62e^{11}$ (1.0×) | $6.75e^{14}$ (2.0×) | $8.98e^{10}$ (0.0005×) |
| | PGA | $59.1_{\pm0.6}$ | $76.2\ (6.5_{\pm0.8})$ | $72.8\ (14.8_{\pm1.0})$ | $71.8\ (16.1_{\pm1.8})$ | $63.0\ (14.1_{\pm1.6})$ | 12.9 | $9.86e^{09}$ (16.4×) | $8.54e^{13}$ (15.8×) | $4.94e^{08}$ (0.0909×) |
| | MoDe | $58.7_{\pm0.5}$ | $73.9\ (8.8_{\pm1.0})$ | $69.0\ (11.1_{\pm0.7})$ | $70.2\ (14.6_{\pm2.0})$ | $62.4\ (13.5_{\pm0.8})$ | 12.0 | $1.75e^{10}$ (9.3×) | $1.46e^{14}$ (9.3×) | $8.98e^{07}$ (0.5×) |
| | FedAU | $59.0_{\pm0.5}$ | $75.5\ (7.2_{\pm0.9})$ | $74.7\ (16.8_{\pm1.0})$ | $76.3\ (20.6_{\pm2.1})$ | $61.6\ (12.8_{\pm2.4})$ | 14.4 | $\mathbf{2.67e^{09}}$ **(60.8×)** | $\mathbf{2.23e^{13}}$ **(60.6×)** | $4.49e^{07}$ (1.0×) |
| | NoT | $58.9_{\pm0.4}$ | $75.2\ (7.5_{\pm0.6})$ | $70.5\ (12.6_{\pm0.8})$ | $69.3\ (13.7_{\pm1.9})$ | $63.3\ (14.4_{\pm0.8})$ | 12.1 | $5.25e^{09}$ (30.9×) | $4.39e^{13}$ (30.8×) | $4.49e^{07}$ (1.0×) |
| | FedOSD | $58.8_{\pm0.5}$ | $76.5\ (6.2_{\pm0.9})$ | $71.1\ (13.1_{\pm0.9})$ | $70.9\ (15.3_{\pm1.9})$ | $61.6\ (12.8_{\pm1.2})$ | 11.9 | $1.05e^{10}$ (15.4×) | $8.78e^{13}$ (15.4×) | $8.98e^{07}$ (0.5×) |
| | FedQUIT | $58.7_{\pm0.4}$ | $\mathbf{76.6\ (6.1_{\pm0.6})}$ | $\mathbf{59.4\ (1.4_{\pm1.1})}$ | $\mathbf{60.6\ (5.0_{\pm1.0})}$ | $\mathbf{51.8\ (3.0_{\pm0.8})}$ | **3.9** | $8.17e^{09}$ (19.8×) | $6.82e^{13}$ (19.8×) | $4.49e^{07}$ (1.0×) |
| **CIFAR-100**, Non-IID, ResNet-18, $E=1$, 10 Clients | Original | $53.8_{\pm0.0}$ | $63.8_{\pm0.8}$ | $62.9_{\pm6.9}$ | $75.4_{\pm7.0}$ | $61.0_{\pm7.8}$ | — | — | — | — |
| | Retrain | $51.0_{\pm1.4}$ | $64.6_{\pm1.2}$ | $33.5_{\pm4.5}$ | $44.0_{\pm5.5}$ | $32.0_{\pm5.2}$ | | $1.62e^{11}$ (1.0×) | $1.35e^{15}$ (1.0×) | $4.49e^{07}$ (1.0×) |
| | FedEraser | $51.2_{\pm0.5}$ | $63.0\ (1.6_{\pm1.2})$ | $35.7\ (2.2_{\pm1.6})$ | $49.1\ (3.4_{\pm2.0})$ | $47.4\ (3.4_{\pm1.5})$ | 2.7 | $1.62e^{11}$ (1.0×) | $6.75e^{14}$ (2.0×) | $8.98e^{10}$ (0.0005×) |
| | PGA | $51.4_{\pm0.6}$ | $62.9\ (1.7_{\pm1.3})$ | $37.4\ (4.3_{\pm4.1})$ | $49.2\ (6.4_{\pm4.9})$ | $36.6\ (5.3_{\pm5.3})$ | 4.4 | $5.74e^{09}$ (28.2×) | $5.10e^{13}$ (26.5×) | $4.94e^{08}$ (0.09×) |
| | MoDe | $50.9_{\pm1.1}$ | $61.8\ (2.8_{\pm1.3})$ | $41.9\ (8.4_{\pm3.8})$ | $55.0\ (10.6_{\pm4.0})$ | $42.3\ (10.3_{\pm3.8})$ | 8.0 | $2.19e^{10}$ (7.4×) | $1.83e^{14}$ (7.4×) | $8.98e^{07}$ (0.5×) |
| | FedAU | $51.2_{\pm1.2}$ | $60.8\ (3.8_{\pm1.4})$ | $38.4\ (4.5_{\pm2.2})$ | $51.7\ (6.7_{\pm2.5})$ | $38.3\ (5.3_{\pm1.9})$ | 5.1 | $1.29e^{10}$ (12.5×) | $1.08e^{14}$ (12.5×) | $4.49e^{07}$ (1.0×) |
| | NoT | $51.3_{\pm0.2}$ | $57.8\ (6.8_{\pm1.0})$ | $43.3\ (9.9_{\pm4.4})$ | $55.7\ (11.7_{\pm5.7})$ | $44.3\ (12.3_{\pm5.5})$ | 10.2 | $1.03e^{10}$ (15.7×) | $8.64e^{13}$ (15.6×) | $4.49e^{07}$ (1.0×) |
| | FedOSD | $51.6_{\pm0.7}$ | $63.1\ (1.5_{\pm0.9})$ | $37.3\ (3.8_{\pm2.0})$ | $\mathbf{48.9\ (4.9_{\pm2.4})}$ | $35.9\ (3.9_{\pm2.1})$ | 3.5 | $9.69e^{09}$ (16.7×) | $8.10e^{13}$ (16.7×) | $8.98e^{07}$ (0.5×) |
| | FedQUIT | $52.0_{\pm0.5}$ | $\mathbf{63.2\ (1.4_{\pm1.3})}$ | $\mathbf{34.6\ (3.5_{\pm2.4})}$ | $49.7\ (5.5_{\pm2.8})$ | $\mathbf{33.9\ (3.8_{\pm2.8})}$ | 3.6 | $\mathbf{3.08e^{09}}$ **(52.6×)** | $\mathbf{2.57e^{13}}$ **(52.5×)** | $4.49e^{07}$ (1.0×) |
| **CIFAR-10**, Non-IID, ResNet-18, $E=1$, 10 Clients | Original | $83.7_{\pm0.0}$ | $88.6_{\pm0.3}$ | $88.6_{\pm3.9}$ | $84.4_{\pm5.4}$ | $81.4_{\pm6.3}$ | — | — | — | — |
| | Retrain | $83.5_{\pm1.6}$ | $89.0_{\pm0.6}$ | $81.1_{\pm8.0}$ | $73.1_{\pm13.7}$ | $72.5_{\pm10.8}$ | | $1.62e^{11}$ (1.0×) | $1.35e^{15}$ (1.0×) | $4.49e^{07}$ (1.0×) |
| | PGA | $84.0_{\pm1.0}$ | $88.7\ (0.3_{\pm0.6})$ | $84.3\ (3.2_{\pm4.9})$ | $78.5\ (5.4_{\pm7.7})$ | $77.0\ (4.5_{\pm6.0})$ | 3.4 | $9.38e^{09}$ (17.3×) | $8.14e^{13}$ (16.6×) | $4.94e^{08}$ (0.09×) |
| | MoDe | $83.5_{\pm1.6}$ | $87.9\ (1.1_{\pm0.9})$ | $82.3\ (2.0_{\pm1.7})$ | $72.6\ (2.6_{\pm1.4})$ | $71.8\ (1.5_{\pm1.0})$ | 1.8 | $2.40e^{10}$ (6.7×) | $2.01e^{14}$ (6.7×) | $8.98e^{07}$ (0.5×) |
| | FedAU | $83.8_{\pm1.4}$ | $89.4\ (0.4_{\pm0.7})$ | $85.2\ (4.0_{\pm2.6})$ | $76.9\ (4.3_{\pm3.7})$ | $73.2\ (2.9_{\pm2.9})$ | 2.9 | $5.17e^{09}$ (31.3×) | $4.32e^{13}$ (31.2×) | $4.49e^{07}$ (1.0×) |
| | NoT | $83.8_{\pm1.2}$ | $83.5\ (5.5_{\pm0.6})$ | $84.3\ (3.1_{\pm3.0})$ | $78.0\ (4.9_{\pm4.5})$ | $75.3\ (2.9_{\pm3.4})$ | 4.1 | $1.37e^{10}$ (11.9×) | $1.14e^{14}$ (11.8×) | $4.49e^{07}$ (1.0×) |
| | FedOSD | $83.7_{\pm1.3}$ | $88.6\ (0.4_{\pm0.8})$ | $83.6\ (2.5_{\pm2.2})$ | $76.2\ (3.1_{\pm2.9})$ | $74.9\ (2.4_{\pm1.8})$ | 2.1 | $1.05e^{10}$ (15.4×) | $8.78e^{13}$ (15.4×) | $8.98e^{07}$ (0.5×) |
| | FedQUIT | $83.7_{\pm1.2}$ | $\mathbf{88.8\ (0.2_{\pm0.7})}$ | $\mathbf{81.7\ (1.4_{\pm1.7})}$ | $\mathbf{72.1\ (1.6_{\pm1.8})}$ | $\mathbf{71.1\ (1.3_{\pm1.5})}$ | **1.1** | $\mathbf{3.16e^{10}}$ **(51.3×)** | $\mathbf{2.64e^{13}}$ **(51.1×)** | $4.49e^{07}$ (1.0×) |
| **CUB-200**, Non-IID, MiT-B0, $E=1$, 10 Clients | Original | $60.0_{\pm0.0}$ | $81.2_{\pm1.0}$ | $84.5_{\pm10.4}$ | $71.9_{\pm12.3}$ | $68.9_{\pm10.2}$ | — | — | — | — |
| | Retrain | $56.5_{\pm1.0}$ | $82.5_{\pm0.7}$ | $34.4_{\pm6.2}$ | $22.4_{\pm5.5}$ | $19.7_{\pm4.8}$ | | $1.19e^{10}$ (1.0×) | $3.82e^{15}$ (1.0×) | $1.33e^{07}$ (1.0×) |
| | PGA | $57.1_{\pm0.4}$ | $82.1\ (0.4_{\pm0.3})$ | $39.8\ (5.4_{\pm3.3})$ | $27.8\ (5.3_{\pm3.2})$ | $26.7\ (7.0_{\pm3.1})$ | 4.5 | $2.87e^{09}$ (4.1×) | $9.53e^{14}$ (4.0×) | $1.46e^{08}$ (0.0909×) |
| | MoDE | $57.0_{\pm1.2}$ | $81.8\ (0.7_{\pm0.4})$ | $46.6\ (12.2_{\pm2.8})$ | $29.2\ (6.8_{\pm2.6})$ | $26.9\ (7.2_{\pm2.3})$ | 6.7 | $6.67e^{09}$ (1.8×) | $2.14e^{15}$ (1.8×) | $2.66e^{07}$ (0.5×) |
| | FedAU | $56.9_{\pm1.2}$ | $\mathbf{82.3\ (0.2_{\pm0.5})}$ | $\mathbf{37.8\ (3.4_{\pm2.8})}$ | $26.7\ (4.3_{\pm2.2})$ | $23.0\ (3.3_{\pm1.6})$ | 2.8 | $6.29e^{09}$ (1.9×) | $2.01e^{15}$ (1.9×) | $1.33e^{07}$ (1.0×) |
| | NoT | $57.0_{\pm1.2}$ | $81.2\ (1.3_{\pm0.6})$ | $54.7\ (20.3_{\pm5.1})$ | $40.9\ (18.6_{\pm4.3})$ | $37.5\ (17.8_{\pm3.4})$ | 14.5 | $2.39e^{09}$ (5.0×) | $7.65e^{14}$ (5.0×) | $1.33e^{07}$ (1.0×) |
| | FedOSD | $57.0_{\pm1.0}$ | $80.1\ (2.4_{\pm0.4})$ | $39.5\ (5.1_{\pm2.7})$ | $28.0\ (5.6_{\pm2.5})$ | $24.6\ (4.9_{\pm2.3})$ | 4.5 | $3.82e^{09}$ (3.1×) | $1.22e^{15}$ (3.1×) | $2.66e^{07}$ (0.5×) |
| | FedQUIT | $57.0_{\pm0.9}$ | $\mathbf{82.3\ (0.2_{\pm0.4})}$ | $38.0\ (3.6_{\pm2.7})$ | $\mathbf{24.6\ (2.8_{\pm2.5})}$ | $\mathbf{21.7\ (2.0_{\pm1.5})}$ | **2.2** | $\mathbf{2.20e^{09}}$ **(5.4×)** | $\mathbf{7.05e^{14}}$ **(5.4×)** | $1.33e^{07}$ (1.0×) |
| **CIFAR-100**, Non-IID, ResNet-18, $E=1$, 100 Clients | Original | $35.1_{\pm0.6}$ | $36.6_{\pm0.7}$ | $49.5_{\pm3.8}$ | $59.8_{\pm7.0}$ | $48.0_{\pm7.5}$ | — | — | — | — |
| | Retrain | $35.1_{\pm0.8}$ | $36.8_{\pm0.9}$ | $34.2_{\pm5.2}$ | $45.2_{\pm5.0}$ | $32.5_{\pm4.5}$ | | $5.39e^{11}$ (1.0×) | $4.50e^{14}$ (1.0×) | $4.49e^{07}$ (1.0×) |
| | MoDe | $36.2_{\pm0.7}$ | $38.0\ (1.2_{\pm0.8})$ | $38.9\ (4.7_{\pm2.3})$ | $49.9\ (4.7_{\pm2.4})$ | $36.5\ (4.0_{\pm2.1})$ | 3.7 | $1.07e^{11}$ (5.0×) | $8.92e^{13}$ (5.0×) | $8.98e^{07}$ (0.5×) |
| | FedAU | $36.0_{\pm0.8}$ | $34.7\ (2.1_{\pm0.9})$ | $39.9\ (5.7_{\pm2.6})$ | $50.5\ (5.3_{\pm2.7})$ | $38.0\ (5.5_{\pm2.6})$ | 4.7 | $3.59e^{10}$ (15.0×) | $3.00e^{13}$ (15.0×) | $4.49e^{07}$ (1.0×) |
| | NoT | $36.3_{\pm0.6}$ | $34.8\ (2.0_{\pm0.9})$ | $40.0\ (5.8_{\pm2.8})$ | $51.0\ (5.8_{\pm2.9})$ | $38.7\ (6.2_{\pm3.0})$ | 5.0 | $4.49e^{10}$ (12.0×) | $3.75e^{13}$ (12.0×) | $4.49e^{07}$ (1.0×) |
| | FedOSD | $36.5_{\pm0.8}$ | $38.0\ (1.2_{\pm0.8})$ | $39.9\ (5.7_{\pm2.5})$ | $50.2\ (5.0_{\pm2.5})$ | $37.8\ (5.3_{\pm2.4})$ | 4.3 | $1.26e^{10}$ (42.9×) | $1.05e^{13}$ (42.9×) | $8.98e^{07}$ (0.5×) |
| | FedQUIT | $36.5_{\pm0.7}$ | $\mathbf{37.9\ (1.1_{\pm0.7})}$ | $\mathbf{36.1\ (1.9_{\pm1.5})}$ | $\mathbf{46.8\ (1.6_{\pm1.4})}$ | $\mathbf{34.8\ (2.3_{\pm1.6})}$ | **1.7** | $\mathbf{7.27e^{09}}$ **(74.1×)** | $\mathbf{6.08e^{12}}$ **(74.1×)** | $4.49e^{07}$ (1.0×) |
| **CUB-200**, Non-IID, MiT-B0, $E=1$, 100 Clients | Original | $45.3_{\pm0.4}$ | $57.4_{\pm0.8}$ | $72.0_{\pm5.5}$ | $60.0_{\pm8.0}$ | $57.0_{\pm7.5}$ | — | — | — | — |
| | Retrain | $43.8_{\pm1.0}$ | $56.7_{\pm0.9}$ | $41.3_{\pm6.3}$ | $29.0_{\pm5.5}$ | $26.0_{\pm4.8}$ | | $5.31e^{10}$ (1.0×) | $2.04e^{14}$ (1.0×) | $1.33e^{07}$ (1.0×) |
| | MoDe | $44.4_{\pm1.1}$ | $57.0\ (0.3_{\pm0.5})$ | $43.6\ (2.3_{\pm1.8})$ | $31.6\ (2.6_{\pm2.0})$ | $29.6\ (3.6_{\pm2.3})$ | 2.2 | $3.16e^{10}$ (1.7×) | $1.21e^{14}$ (1.7×) | $2.66e^{07}$ (0.5×) |
| | FedAU | $44.2_{\pm1.0}$ | $56.8\ (0.1_{\pm0.4})$ | $43.6\ (2.3_{\pm1.8})$ | $31.6\ (2.6_{\pm2.0})$ | $29.6\ (3.6_{\pm2.3})$ | 2.1 | $7.97e^{09}$ (6.7×) | $3.06e^{13}$ (6.7×) | $1.33e^{07}$ (1.0×) |
| | NoT | $44.5_{\pm1.2}$ | $56.5\ (0.2_{\pm0.5})$ | $64.0\ (22.7_{\pm4.5})$ | $52.0\ (23.0_{\pm4.8})$ | $50.0\ (24.0_{\pm5.0})$ | 17.5 | $4.25e^{09}$ (12.5×) | $1.63e^{13}$ (12.5×) | $1.33e^{07}$ (1.0×) |
| | FedOSD | $44.5_{\pm1.0}$ | $\mathbf{56.7\ (0.0_{\pm0.4})}$ | $43.0\ (1.7_{\pm1.5})$ | $31.0\ (2.0_{\pm1.8})$ | $29.0\ (3.0_{\pm2.0})$ | 1.7 | $4.25e^{09}$ (12.5×) | $1.63e^{13}$ (12.5×) | $2.66e^{07}$ (0.5×) |
| | FedQUIT | $44.5_{\pm1.1}$ | $56.9\ (0.2_{\pm0.4})$ | $\mathbf{39.7\ (1.6_{\pm1.4})}$ | $\mathbf{27.7\ (1.3_{\pm1.3})}$ | $\mathbf{25.7\ (0.3_{\pm1.0})}$ | **0.9** | $\mathbf{8.23e^{08}}$ **(64.5×)** | $\mathbf{3.16e^{12}}$ **(64.5×)** | $1.33e^{07}$ (1.0×) |

Table 1: Post-recovery performance of FedQUIT vs. baselines. Metrics are *mean ($\Delta \pm$ std)*, where $\Delta$ is the average absolute gap with *Retrain*. Lower $\Delta$ corresponds to better unlearning. The *Avg. Gap* column reports the average across $\Delta$. For efficiency, higher $\times$ means greater reduction vs. *Retrain*. When FedEraser Liu et al. (2021) ranks first, it is underlined to note efficiency limits.

that substantially forgets the target client without storing a persistent history of client updates (as in FedEraser), while reducing cumulative communication and computation by $\sim 20\times$ relative to *Retrain*. Across non-IID settings, both with 10 clients and with 100 clients, FedQUIT closely approximates *Retrain*, achieving the lowest or second-lowest average gap (Avg. Gap) on efficacy metrics, while delivering the largest or second-largest efficiency gains, without the storage burden required by FedEraser. We validate our results also using more local epochs ($E = 10$) in Appendix 7.

**Fixed-budget Evaluation.** To examine behavior beyond recovery, Table 2 reports the performance of all methods under a large fixed communication budget. We consider a budget equal to 50% of the communication cost required for retraining the model from scratch; additional budget levels are provided in Appendix D.5 and Table 11. This setting rules out the possibility that forgetting degrades once the model has already regained its utility. **Findings.** FedQUIT consistently outperforms the baselines, showing smaller gaps in Forget Acc. and often higher Test and Retain Acc. under the same budget. This advantage reflects FedQUIT's ability to restore model utility while maintaining strong selectivity in forgetting.

**Post-unlearning Results.** We analyze the degradation in model utility introduced by each method's unlearning phase. Table 3 reports the Test Accuracy immediately after unlearning (before recovery) and the per-method cost for unlearning only; a full breakdown for all metrics and settings appears in Appendix Table 9. **Findings.** On IID settings, FedQUIT attains the second-highest post-unlearning Test Accuracy after MoDe. However, MoDe relies on a multi-round unlearning phase (see baseline discussion in Appendix D.2), which substantially increases costs — typically $\sim 150\times$ higher FLOPs/bytes than FedQUIT in our settings. Other baselines are less selective on IID data and col-

| Budget | Method | IID | | | Non-IID | | |
|---|---|---|---|---|---|---|---|
| | | Test Acc. ($\Delta$) | Retain Acc. ($\Delta$) | Forget Acc. ($\Delta$) | Test Acc. ($\Delta$) | Retain Acc. ($\Delta$) | Forget Acc. ($\Delta$) |
| 100% | Retrain | 58.31 | 82.72 | 58.02 | 51.02 | 64.61 | 33.52 |
| 50% | FedEraser | 45.48 (-12.83) | 56.28 (-26.44) | 45.42 (-12.60) | 37.76 (-13.26) | 41.32 (-23.29) | 19.99 (-13.53) |
| | PGA | 61.09 (+2.78) | 89.95 (+7.23) | 68.83 (+10.81) | 55.20 (+4.18) | 73.31 (+8.70) | 40.49 (+6.97) |
| | MoDe | 61.44 (+3.13) | 91.98 (+9.26) | 68.50 (+10.48) | 54.95 (+3.93) | 72.45 (+7.84) | 41.18 (+7.66) |
| | FedAU | 61.57 (+3.26) | 91.03 (+8.31) | 66.66 (+8.64) | 55.38 (+4.36) | 74.42 (+9.81) | 41.88 (+8.36) |
| | NoT | 61.05 (+2.74) | 90.59 (+7.87) | 66.36 (+8.34) | 55.10 (+4.08) | 73.27 (+8.66) | 40.39 (+6.87) |
| | FedOSD | 61.25 (+2.94) | 90.32 (+7.60) | 66.88 (+8.86) | 55.30 (+4.28) | 74.14 (+9.53) | 39.76 (+6.24) |
| | FedQUIT | **61.67 (+3.36)** | **91.52 (+8.80)** | **62.58 (+4.56)** | **55.48 (+4.46)** | **74.47 (+9.86)** | **37.51 (+3.99)** |

Table 2: Comparison at fixed communication budget of 50% ($8.10e^{10}$ bytes). Values in parentheses indicate the gap (negative gap in red) with *Retrain* for the corresponding setting (IID or Non-IID).

| Method | CIFAR-100, IID, ResNet-18 | | | CIFAR-100, N-IID, ResNet-18 | | | CIFAR-100, N-IID, MiT-B0 | | | CUB-200, N-IID, MiT-B0 | | |
|---|---|---|---|---|---|---|---|---|---|---|---|---|
| | Test Acc. | Bytes | FLOPs | Test Acc. | Bytes | FLOPs | Test Acc. | Bytes | FLOPs | Test Acc. | Bytes | FLOPs |
| Retrain | $58.3_{\pm0.5}$ | $1.62e^{11}$ | $1.62e^{15}$ | $51.0_{\pm1.4}$ | $1.62e^{11}$ | $1.62e^{15}$ | $73.3_{\pm0.8}$ | $1.19e^{10}$ | $3.82e^{15}$ | $56.5_{\pm1.0}$ | $1.19e^{10}$ | $3.82e^{15}$ |
| PGA | $1.0_{\pm0.2}$ | $8.98e^{07}$ | $3.75e^{12}$ | $1.0_{\pm0.2}$ | $8.98e^{07}$ | $3.75e^{12}$ | $10.1_{\pm0.2}$ | $2.66e^{07}$ | $4.25e^{13}$ | $5.4_{\pm0.2}$ | $2.66e^{07}$ | $4.25e^{13}$ |
| MoDe† | $36.8_{\pm5.6}$ | $1.38e^{10}$ | $1.16e^{14}$ | $17.5_{\pm3.6}$ | $1.38e^{10}$ | $1.16e^{14}$ | $66.6_{\pm2.1}$ | $4.09e^{09}$ | $1.31e^{15}$ | $43.4_{\pm2.2}$ | $4.09e^{09}$ | $1.31e^{15}$ |
| FedAU | $4.2_{\pm1.3}$ | $8.21e^{07}$ | $1.84e^{12}$ | $4.8_{\pm1.5}$ | $8.21e^{07}$ | $1.84e^{12}$ | $1.8_{\pm1.1}$ | $1.03e^{07}$ | $4.61e^{10}$ | $0.8_{\pm0.2}$ | $1.03e^{07}$ | $4.61e^{10}$ |
| NoT* | $12.2_{\pm0.0}$ | 0 | 0 | $9.5_{\pm0.0}$ | 0 | 0 | $41.3_{\pm0.0}$ | 0 | 0 | $25.3_{\pm0.0}$ | 0 | 0 |
| FedOSD† | $4.5_{\pm1.6}$ | $8.08e^{09}$ | $6.75e^{13}$ | $49.5_{\pm0.9}$ | $8.08e^{09}$ | $6.75e^{13}$ | $58.8_{\pm2.1}$ | $2.39e^{09}$ | $7.65e^{14}$ | $52.7_{\pm1.2}$ | $2.39e^{09}$ | $7.65e^{14}$ |
| FedQUIT | $25.5_{\pm1.1}$ | $8.98e^{07}$ | $7.50e^{11}$ | $45.2_{\pm1.0}$ | $8.98e^{07}$ | $7.50e^{11}$ | $50.4_{\pm1.6}$ | $2.66e^{07}$ | $8.50e^{12}$ | $47.8_{\pm1.9}$ | $2.66e^{07}$ | $8.50e^{12}$ |

Table 3: Post-unlearning Test Acc. and cost (Bytes, FLOPs). †: multi-round unlearning with active target client; *: identical unlearned model for any target client (std = 0).

| Method | Test Accuracy | | Retain Accuracy | | Forget Accuracy | | Comm. | Comp. |
|---|---|---|---|---|---|---|---|---|
| | After Unlearning | After Recovery | After Unlearning | After Recovery | After Unlearning | After Recovery | (Bytes $\downarrow$) | (FLOPs $\downarrow$) |
| Original | 51.81 | 51.81 | 63.79 | 63.79 | 66.67 | 66.67 | | |
| Retrain | $50.84 \pm 0.3$ | $50.84 \pm 0.3$ | $64.82 \pm 0.3$ | $64.82 \pm 0.3$ | $55.00 \pm 1.5$ | $55.00 \pm 1.5$ | $1.80e^{11}$ | $1.35e^{15}$ |
| MoDe | $42.20 (8.64 \pm 0.3)$ | $51.53 (0.69 \pm 0.3)$ | $36.30 (28.52 \pm 0.4)$ | $63.20 (1.62 \pm 0.3)$ | $8.51 (46.49 \pm 1.9)$ | $58.70 (3.70 \pm 1.7)$ | $6.19e^{9}$ | $5.18e^{13}$ |
| FedAU | $24.50 (26.34 \pm 0.4)$ | $51.10 (0.26 \pm 0.3)$ | $23.50 (41.32 \pm 0.4)$ | $63.60 (1.22 \pm 0.3)$ | $0.00 (55.00 \pm 2.2)$ | $59.00 (4.00 \pm 1.8)$ | $5.39e^{9}$ | $4.05e^{13}$ |
| NoT | $9.70 (41.14 \pm 0.4)$ | $51.50 (0.66 \pm 0.3)$ | $9.10 (55.72 \pm 0.4)$ | $64.40 (0.42 \pm 0.3)$ | $15.12 (39.88 \pm 2.3)$ | $58.67 (3.67 \pm 1.9)$ | $7.18e^{9}$ | $5.41e^{13}$ |
| FedOSD | $49.30 (1.54 \pm 0.2)$ | $51.93 (1.09 \pm 0.2)$ | $60.51 (4.31 \pm 0.3)$ | $63.95 (0.87 \pm 0.3)$ | **60.00 (5.00 ± 1.8)** | $57.40 (2.40 \pm 1.5)$ | $4.04e^{9}$ | $3.38e^{13}$ |
| FedQUIT | **50.79 (0.05 ± 0.3)** | **52.00 (1.16 ± 0.2)** | **61.20 (3.62 ± 0.3)** | **64.53 (0.29 ± 0.3)** | $41.13 (13.87 \pm 1.3)$ | **56.67 (1.67 ± 1.0)** | **9.87 $e^{8}$** | **7.51 $e^{12}$** |

Table 4: Post-unlearning and post-recovery performance for **sample unlearning** on CIFAR-100 (Non-IID, 1% forget data, ResNet-18, 10 clients). Gaps vs. *Retrain* in parentheses with added $\pm$ std.

lapse utility. On non-IID settings, FedOSD (and, for transformers, MoDe) better preserve utility but at markedly higher cost — typically $\sim 90\times$ more FLOPs/bytes than FedQUIT which still achieves only slightly lower Test Accuracy while operating at orders-of-magnitude lower cost. NoT incurs near-zero unlearning cost (server-side weight manipulation) but substantially degrades utility (costly recovery). PGA and FedAU likewise severely harm utility during unlearning.

**Sample-Unlearning Results.** We evaluate three forget-set sizes: 50% and 10% of local data drawn at random, and, as an extreme case, 1% drawn from the least-represented local classes. Table 4 reports the results for the 1% setting, while Appendix D.6 provides complete results for all configurations, under both Non-IID and IID scenarios. **Findings.** In this extreme setting (1% forget data), FedQUIT outperforms all baselines, showing both lower degradation after unlearning and more accurate forgetting after recovery, while also requiring substantially lower computation and communication costs ($\approx 4\times$). Similar trends hold for the less extreme cases (see Appendix D.6).

## 5.2 EFFECT ON PREDICTIVE ENTROPY

We examine the Shannon entropy of the softmax outputs in Fig. 3a. The plots show the density of predictive entropy for the Original, Retrain, and FedQUIT models, on forget data (left) and on test data (right). **Findings.** The original model is overconfident (i.e., low entropy) on forget data, having seen it during training. In contrast, the retrained model, trained without those data, exhibits higher entropy, indicating more uncertainty. After recovery, the FedQUIT model's predictive entropy distribution on forget data closely matches Retrain (both Retrain and FedQUIT are shifted toward higher entropy) and is well separated from Original (which remains concentrated at low entropy). On test data, the three curves overlap, indicating that, under the predictive entropy lens, FedQUIT achieves selective forgetting on the forget data while preserving behavior on held-out data.

| | Method | After Unlearning | | | R↓ | After Recovery | | |
|---|---|---|---|---|---|---|---|---|
| | | Test Acc. (Δ↓) | Retain Acc. (Δ↓) | Forget Acc. (Δ↑) | | Test Acc. (Δ↓) | Retain Acc. (Δ↓) | Forget Acc. (Δ↓) |
| **IID** | Retrain | 58.31 | 82.72 | 58.02 | | 58.31 | 82.73 | 58.04 |
| | Random | 1.00 (57.3) | 1.00 (81.7) | 1.00 (57.0) | 16.2 | 58.50 (0.2) | 74.50 (8.2) | 69.00 (11.0) |
| | Incompetent | 2.88 (55.4) | 2.88 (79.8) | 2.90 (55.1) | 15.7 | 58.81 (0.5) | 73.09 (9.6) | 67.61 (9.6) |
| | Flatten | 1.61 (56.7) | 1.61 (81.1) | 1.61 (56.4) | 16.9 | 59.20 (0.9) | 74.02 (8.7) | 67.10 (9.1) |
| | Probability Ladder | 13.61 (44.7) | 13.61 (69.1) | 13.05 (45.0) | 8.7 | 59.28 (1.0) | 74.90 (7.8) | 69.91 (11.9) |
| | Logit Ladder | 33.11 (25.2) | 37.24 (45.5) | 34.88 (23.1) | 10.8 | 59.25 (0.9) | 76.14 (6.6) | 66.71 (8.7) |
| | TopK (K=5) | 22.31 (36.0) | 25.16 (57.6) | 22.31 (35.7) | 12.1 | 59.26 (1.0) | 75.42 (7.3) | 61.31 (3.3) |
| | FedQUIT | 25.51 (32.8) | 27.23 (55.5) | 23.54 (34.5) | 10.0 | 58.70 (0.4) | 76.62 (6.1) | 59.42 (1.4) |
| **N-IID (α=0.1)** | Retrain | 51.02 | 64.61 | 33.52 | | 51.02 | 64.61 | 33.52 |
| | Random | 3.40 (47.6) | 4.90 (59.7) | 4.00 (29.5) | 1.4 | 51.70 (0.7) | 62.50 (2.1) | 52.50 (19.0) |
| | Incompetent | 18.30 (32.7) | 20.00 (44.6) | 12.90 (20.6) | 1.3 | 51.80 (0.8) | 62.30 (2.3) | 50.20 (16.7) |
| | Flatten | 16.02 (35.0) | 18.40 (46.2) | 2.10 (31.4) | 1.2 | 52.60 (1.6) | 63.80 (0.8) | 48.50 (15.0) |
| | Probability Ladder | 40.04 (11.0) | 47.42 (17.2) | 22.82 (10.7) | 1.3 | 52.00 (1.0) | 63.30 (1.3) | 47.20 (13.7) |
| | Logit Ladder | 46.93 (4.1) | 58.22 (6.4) | 18.64 (14.9) | 1.3 | 51.90 (0.9) | 63.90 (0.7) | 39.80 (6.3) |
| | TopK (K=5) | 42.74 (8.3) | 53.71 (10.9) | 7.30 (26.2) | 2.0 | 51.20 (0.2) | 63.40 (1.2) | 29.00 (4.5) |
| | FedQUIT | 45.24 (5.8) | 55.82 (8.8) | 8.02 (25.5) | 3.7 | 52.04 (1.0) | 63.22 (1.4) | 34.62 (3.5) |

Table 5: Ablation on non-true structure of virtual teacher. Results just after unlearning and after recovery. $R$ means recovery rounds. In parenthesis, gap with *Retrain*. ResNet-18, CIFAR-100.

### 5.3 ABLATION ON VIRTUAL TEACHER'S NON-TRUE STRUCTURE

We investigate whether distilling the knowledge encoded in non-true output geometry maintains model utility during FedQUIT unlearning. We introduce a spectrum of virtual teachers, from least to most informative in retained non-true structure: flatten (uniform over non-true classes), top-K (keeps only the head, collapses the tail), rank-only logit ladder (preserves rank but not gaps), probability ladder (preserves rank with equal steps in probability), and FedQUIT (preserves non-true structure). All alternatives preserve the total exponential mass of the non-true classes, keeping the post-edit true-class probability unchanged. The incompetent teacher uses a fully uniform distribution, the random teacher provides random hard labels. Table 5 reports post-unlearning and post-recovery results. The temperature ablation in Appendix D.8 and Table 16, where we progressively smooth only the non-true logits, provides a complementary perspective. **Findings.** (i) Rank matters. Right after unlearning, rank-preserving surrogates dominate structure-free ones for test/retain utility (logit ladder > probability ladder > flatten ≈ incompetent > random) and under IID they also require fewer recovery rounds; hence, the ordering among non-true classes by itself carries substantial knowledge. (ii) Relative geometry matters. To align with retrain after recovery, preserving gaps and tail mass is crucial (FedQUIT > Top-K > logit ladder > probability ladder > flatten > random); hence, the full non-true geometry drives the best post-recovery behavior. (iii) Increasing the temperature on non-true logits steadily erodes retain and test utility just after unlearning and shifts FedQUIT toward the behavior of flattened or highly smoothed teachers, which fails to deliver precise unlearning and reflects the patterns observed in the structure ablation. Further analysis in Appendix D.8.

### 5.4 SENSITIVITY ANALYSIS ON $v$

We report results for FedQUIT with $v_i = \min_c z_{i,c}^g$ for each forget sample $x_i$, a parameter-free and data-adaptive choice. We provided theoretical properties for this choice in Section 3. Here we analyze it empirically. Figures 3b and 3c sweep fixed values of $v$ (x axis) and plot the corresponding $\Delta$ Forget Accuracy w.r.t. *Retrain* (y axis) on IID and non IID CIFAR–100 with ResNet–18; the charts also overlay the density of true class logits of the Original model on forget data. $\Delta$ Forget Accuracy serves as a proxy for effective forgetting (lower is better). **Findings.** As $v$ moves toward the left tail of that density, performance approaches *Retrain* (stronger forgetting), in line with Proposition 1, which shows that decreasing $v$ increases the forgetting signal. We also compare two special choices in Appendix Table 17: $v$ to maximizes per-sample entropy ($v^\star$) and an extreme $v \ll \min_c z_{i,c}^g$ so that $p_{i,y_i}^{\text{virt}} \approx 0$. The first underforgets (theoretical intuition in Appendix 3), whereas the latter is not significantly better than $v_i = \min_c z_{i,c}^g$. This supports the default parameter-free $v_i = \min_c z_{i,c}^g$.

## 6 RELATED WORK

**Federated unlearning (FU).** We group prior FU methods by their core mechanism: **Ⓐ History-based subtraction.** These methods retain per–client update histories and reconstruct a sanitized model by removing the target client's contribution. FedEraser (Liu et al., 2021) recalibrates histor-

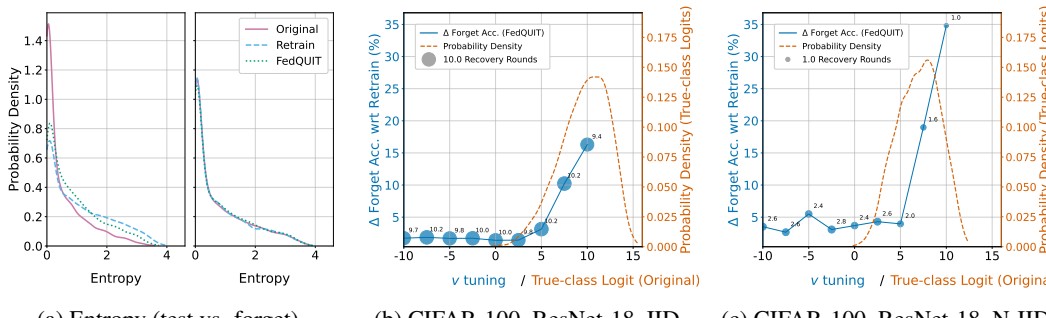

(a) Entropy (test vs. forget).  (b) CIFAR-100, ResNet-18, IID.  (c) CIFAR-100, ResNet-18, N-IID.

Figure 3: **(a)** Predictive-entropy densities of the global model on forget (left) and test (right) data. Setting: CIFAR-100 (Non-IID), MiT-B0. **(b-c)** Forget efficacy of FedQUIT as $v$ is tuned (Eq. 1). Left $y$-axis: absolute after-recovery gap in forget accuracy ($\Delta$) w.r.t. *Retrain* (lower is better). Right $y$-axis: probability density of the original model's *true-class logits* on the forget set (pre-FedQUIT). *Shared x-axis:* the abscissa simultaneously denotes the tuning value $v$ (for the left-axis curve) and the true-class logit (for the right-axis density). Marker size encodes the number of recovery rounds.

ical updates excluding the target contribution; Liu et al. (2022) accelerates retraining using quasi Newton updates. Other works subtract the target's updates and then use KD on public proxy data to recover utility (Wu et al., 2022b; 2023; Zhang et al., 2025). Unlike prior KD-based FU methods that rely on stored historical updates and distill the pre-unlearning global model using external or public data, FedQUIT introduces a different distillation mechanism. History-based approaches first roll back the target client's updates and then use KD on a proxy dataset to recover utility. In contrast, FedQUIT uses no stored updates and no auxiliary or public data. It constructs a virtual teacher directly from the current global model and performs distillation locally on the requesting client's data, enabling a fully on-device and single-round unlearning step. **Ⓑ Gradient manipulation.** These methods manipulate gradients computed on the forget data before applying them to the unlearned model. PGA (Halimi et al., 2022) performs projected gradient ascent on forget data, enforcing an $l2$-norm constraint around a reference model. FedOSD (Pan et al., 2025) introduces a two-stage, multi-round method: first, it enforces forgetting by training the target client with a custom Cross-Entropy and updating the server via an orthogonal steepest-descent direction; then, it post-trains while projecting retained gradients to prevent conflicts. **Ⓒ Direct weight manipulation.** FUSED (Zhong et al., 2025) preemptively identifies critical layers and trains sparse unlearning adapters only on retained clients. NoT (Khalil et al., 2025) negates the weights of the first layer, producing the same unlearned model regardless of forget data. This lack of selectivity leaves residual influence and often prolongs recovery. **Ⓓ Supervision perturbation.** These methods modify the supervision signal on forget data. FedUNRAN (Mora et al., 2024a) fine tunes with random labels; FedAU (Gu et al., 2024) learns an auxiliary module on random labels and combines it with the original model; MoDe (Zhao et al., 2023) proposes a multi-round mechanism that alternates, in each round, degradation and memory guidance using hard labels from a degraded model. Random-label perturbations can be viewed as supervision via a teacher model that outputs random hard labels (see Appendix A). Under our theoretical framework, this type of teacher violates the geometry-preserving assumption in Lemma 2, which is required to guarantee a bounded perturbation during unlearning. FedQUIT (our method) belongs to this supervision-perturbation family, but preserves the non–true-class structure of the current global model (true-class logit penalized, other logits preserved), thereby ensuring a bounded perturbation. Appendix E contrasts methods by their practical constraints.

# 7 CONCLUSION

We presented FedQUIT, a novel and general teacher-student framework for FU. FedQUIT uses a crafted version of the FL global model's outputs as teacher to selectively perform unlearning while preserving model utility of the original global model. In the experiments, we show that using such specially crafted teachers is an effective approach to forget data and favors rapid recovery.

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

We acknowledge that Large Language Models (LLMs) were employed in the preparation of this manuscript. Specifically, we used LLM-based tools to assist with presentation and editorial tasks, including improving grammar, wording, and overall clarity of the text.

## A RATIONALE AND MOTIVATIONS

We aimed to design an FU method that operates on-device and fully adheres to the FL privacy requirements. This entails that the method would have direct access only to the unlearning client's data, while the rest of the data in the federation (the retain data) would not be available for use in the unlearning algorithms, except via some form of proxy. To indirectly utilize the knowledge extracted from the retain data, we leverage KD and use the FL global model as the teacher, which serves as a proxy for the data we cannot access. To balance unlearning of local forget data and retaining of global knowledge, we craft a virtual teacher by modifying the global model output. In the following paragraphs, we first provide an overview of the usage of the global model as a local teacher in FL to combat data heterogeneity. Next, we present (and critically evaluate) the usage of an incompetent teacher to induce forgetting in centralized settings. We call our FedQUIT virtual teacher *quasi-competent* because it preserves the global model's relative ordering and geometry among the outputs of non-true classes, while explicitly penalizing the true-class logit.

**Using a competent teacher to tackle catastrophic forgetting.** In FL, and especially in cross-device FL, data are supposed to be non-IID. This characteristic translates to drift of client models, that diverge from the global model making the next FedAvg aggregation inefficient. Extensive research has investigated the use of the global model as an anchor for local models to mitigate this drift, aiming to minimize the disruptive impact of data heterogeneity on aggregation. An approach to implementing this guidance is via KD, with the global model acting as the teacher and the local model as the student (Yao et al., 2021; Lee et al., 2022; Lu et al., 2023; Guo et al., 2024). This approach preserves global knowledge while incorporating local adaptations, which helps limit the phenomenon of *catastrophic forgetting* – the rapid overwriting of previous knowledge to accommodate the distribution shift imposed by the specific clients participating in each round.

**Using an incompetent teacher to induce forgetting.** On the other hand, inducing forgetting – the complementary of tackling catastrophic forgetting – via KD is typically achieved using an incompetent teacher in place of a competent teacher (e.g., (Chundawat et al., 2023)). The underlying assumption of using an incompetent teacher is that the unlearned model should be indifferent to the forget data, producing non-informative predictions when presented with such data. However, this assumption often does not hold in practice. It is common that the retrained model still produces informative predictions on forget data, which the retrained model has never seen during training, demonstrating its ability to generalize. In Table 1, *Retrain* attains Forget Accuracy comparable to its Test Accuracy and far above the $1/C$ expected accuracy of a random teacher on forget data.

### A.1 USING RANDOM LABELS TO INDUCE FORGETTING

Previous works have used hard random labels as supervision perturbation to induce forgetting (e.g., (Chundawat et al., 2023; Mora et al., 2024a; Gu et al., 2024)). Here, we first conceptually link random-label supervision to the teacher–student framework, and then explain why our virtual teacher has unique characteristics.

Using random hard labels under cross-entropy is equivalent to distillation with a "random teacher" whose output is a one-hot vector. Let $\mathbf{p}^t = e_y$, where $e_y$ is the one-hot vector corresponding to a randomly chosen class $y$. The forward KL divergence used in KD becomes

$$\mathrm{KL}(\mathbf{p}^t \,\|\, \mathbf{p}^d) = \sum_c \mathbf{p}^t(c) \log \frac{\mathbf{p}^t(c)}{\mathbf{p}^s(c)} = -\log \mathbf{p}^s(y),$$

which is exactly cross-entropy with a random hard label. Thus, random-label fine-tuning is a special case of KD with an uninformative teacher.

**Non-preserved non-true-class geometry.** Because $\mathbf{p}^t$ ($\mathbf{p}^{\mathrm{virt}}$ in our analysis) is one-hot, random-label supervision collapses the logit geometry of the non–true classes: all information about their rankings, gaps, and log-odds ratios is discarded. This violates the structural conditions required in our Lemmas (in particular, the assumptions on monotonicity and preserved logit relationships used in Lemmas 1–2).

**Empirical observations.** Our ablations (Table 5) further confirm this distinction: random-label and uniform teachers lead to poor utility recovery, longer recovery phases, and weaker forgetting (large gaps relative to the retrain baseline), while FedQUIT's structured teacher consistently achieves the best trade-off between forgetting accuracy and utility preservation.

In summary, although random-label approaches fit within the broader category of supervision perturbation, they break the non–true-class geometry central to both our theoretical guarantees and our empirical effectiveness. FedQUIT's structured soft-target modification preserves this geometry while reducing true-class evidence, enabling efficient and stable local unlearning.

## A.2 Relation with Self-Supervision and Self-Distillation

In our method – and in general when the global model guides local training – at the beginning of the unlearning routine, the unlearned model and the global model (the student and the teacher) have the same model parameters. In this sense, the FedQUIT unlearning routine resembles a self-supervised learning, with the difference that the self-supervision is from a slightly modified version of self.

## A.3 Similarly Inspired Work in FL

The mechanisms that we present in this paper use a student-teacher framework locally at FL clients to retain the good knowledge from the original model while selectively scrubbing the contributions to forget. We use a crafted version of the FL global model as the teacher, serving as a natural proxy for the retain data that cannot be directly accessed in FL. Recently, KD (Hinton et al., 2015; Buciluǎ et al., 2006) has been widely used in FL to address various issues (Mora et al., 2024b), such as reducing communication costs, enabling model heterogeneity within the federation, improving generalization or personalization performance with heterogeneous data, and even performing unlearning. Most relevantly for this paper, numerous works employ the global model as a KD teacher for the local model to mitigate catastrophic forgetting and client drift in the presence of heterogeneous data (e.g., (Yao et al., 2021; Lee et al., 2022; Lu et al., 2023; Guo et al., 2024)). For example, FedGKD (Yao et al., 2021) uses the global model output as a guide for local models, limiting client model drifts. FedNTD (Lee et al., 2022) demonstrated that excluding the teacher's true-class logit during the distillation phase is beneficial for local-global distillation. Note that FedNTD (Lee et al., 2022) and FedQUIT differ significantly in both objective and methodology: FedNTD addresses data heterogeneity, while FedQUIT enables client unlearning. From a technical perspective, (1) FedNTD applies KL-divergence as a local regularizer for cross-entropy loss, whereas FedQUIT directly optimizes the KL divergence; (2) FedNTD ignores the logits produced by the true classes, while FedQUIT actively penalizes them (an essential step to induce forgetting). Other recent contributions have proposed other label-masking approaches for the global model's output (i.e., the teacher) (Lu et al., 2023; Guo et al., 2024). Unlike these works, our approach modifies the output of the global model, which still acts as a teacher for the local model, to perform unlearning.

## B FedQUIT Algorithm with Auto-Tuning $v$

**FedQUIT (data-adaptive version, Algorithm 2).** At the target client, the student is initialized with the current global weights ($w^s \leftarrow w_t$). For each minibatch of forget data, the client computes student logits $\mathbf{z}^s$ and global logits $\mathbf{z}^g$. A *virtual teacher* is constructed by copying $\mathbf{z}^g$ and replacing, for each sample $i$, the true-class logit with an auto-anchor $v_i = \min_c z^g_{i,c}$, while keeping all non-true logits unchanged. The resulting probabilities $\mathbf{p}^{\mathrm{virt}}$ guide the student through knowledge distillation by minimizing $\mathcal{L}_{\mathrm{KD}} = \frac{1}{B} \sum_i \mathrm{KL}\left(\mathbf{p}^{\mathrm{virt}}_i \| \mathbf{p}^s_i\right)$, updated over $E_u$ local epochs with learning rate $\eta_u$. After unlearning, the client returns the snapshot $w^{\bar{u}}$ to the server. This auto-anchored rule is parameter-free, data-adaptive, and requires only the forget data and current global model.

---

**Algorithm 2** Local unlearning in `FedQUIT` (auto-anchored with per-sample $v_i = \min_c z_{i,c}^g$).

---

**Input:** Global model $w_t$, forget data $D_u^{forget}$, unlearning epochs $E_u$, learning rate $\eta_u$, batch size $B$
**Output:** Unlearned model parameters $w^{\bar{u}}$ to return to the server

1: Client $u$ receives global model $w_t$ from the server
2: **Initialize** student parameters $w^s \leftarrow w_t$                                        ▷ Start from global model
3: **for** each local unlearning epoch $e_u = 1, 2, \ldots, E_u$ **do**
4:     **for** each minibatch $\mathcal{B} = \{(\mathbf{x}_i, y_i)\}_{i=1}^B \subset D_u^{forget}$ **do**
5:         $\mathbf{z}^s \leftarrow h(\mathcal{B}; w^s)$                                        ▷ Student logits on batch, $\mathbf{z}^s \in \mathbb{R}^{B \times C}$
6:         $\mathbf{p}^s \leftarrow \mathrm{softmax}(\mathbf{z}^s)$
7:         $\mathbf{z}^g \leftarrow h(\mathcal{B}; w_t)$                                        ▷ Global logits on batch, $\mathbf{z}^g \in \mathbb{R}^{B \times C}$
8:         $\mathbf{z}^{\mathrm{virt}} \leftarrow \mathbf{z}^g$                                        ▷ Initialize virtual-teacher logits
9:         **for** each $i \in \{1, \ldots, B\}$ **do**                            ▷ Per-sample auto-anchored true-class logit
10:             $v_i \leftarrow \min_{c \in \{1, \ldots, C\}} \mathbf{z}^g[i, c]$                                ▷ Per-sample anchor (parameter-free)
11:             $\mathbf{z}^{\mathrm{virt}}[i, y_i] \leftarrow v_i$                                        ▷ Keep non-true logits unchanged
12:         $\mathbf{p}^{\mathrm{virt}} \leftarrow \mathrm{softmax}(\mathbf{z}^{\mathrm{virt}})$                                        ▷ Temperature $\tau=1$
13:         $\mathcal{L}_{\mathrm{KD}} \leftarrow \frac{1}{B} \sum_{i=1}^B \mathrm{KL}(\mathbf{p}^{\mathrm{virt}}[i] \| \mathbf{p}^s[i])$
14:         Compute $\nabla_{w^s} \mathcal{L}_{\mathrm{KD}}$ and update $w^s \leftarrow \mathrm{ClientOpt}(w^s, \nabla_{w^s} \mathcal{L}_{\mathrm{KD}}, \eta_u)$
15: $w^{\bar{u}} \leftarrow w^s$                                        ▷ Finalize unlearned snapshot
16: Client $u$ sends $w^{\bar{u}}$ back to the server

---

# C    EXTENDED THEORETICAL ANALYSIS

This appendix complements the main-text results (Lemma 1, Proposition 1, Lemma 2, Theorem 1) by stating assumptions, formalizing the evaluation metric, and providing sketch of proofs and comparisons. Our aims are to: (i) formalize *forgetting* via the *evaluation* cross-entropy (CE) on forget data, which is distinct from the KD training loss during unlearning; (ii) prove that the virtual-teacher construction yields a positive, quantifiable forgetting signal; (iii) analyze choices of $v$, showing that the parameter-free $v = \min_c z_{i,c}^g$ (used in our experiments) provides stronger forgetting than the entropy-maximizing $v^\star$ (as we demonstrated also empirically in Appendix Table 17); and (iv) show that FedQUIT induces a bounded parameter perturbation so that FedAvg convergence remains valid when resuming from the unlearned snapshot.

## C.1    NOTATION AND EVALUATION METRIC

For a forget sample $(\mathbf{x}_i, y_i)$, let the student produce logits $\mathbf{z}_i^s$ and probabilities $\mathbf{p}_i^s = \mathrm{softmax}(\mathbf{z}_i^s)$. Define the *evaluation* per-sample CE

$$\ell_i := -\log p_{i,y_i}^s, \qquad \ell_i^+ := -\log p_{i,y_i}^{s+}, \qquad \Delta \ell_i := \ell_i^+ - \ell_i,$$

where $\mathbf{p}_i^{s+} = \mathrm{softmax}(\mathbf{z}_i^{s+})$ follows a KD update. At dataset level, $\mathrm{CE}_{D_u}(w) := \mathbb{E}_{(\mathbf{x},y) \in D_u}[-\log p_y(\mathbf{x}; w)]$. We use the term evaluation cross-entropy (CE) to distinguish it from the KD objective optimized during unlearning. CE is the standard loss for classification, and an increase in CE on the forget data can be taken as evidence of forgetting Yeom et al. (2018).

**Virtual teacher.** As in Eq. 1, the teacher preserves the non-true logits of the global model and sets the true-class logit to $v$:

$$z_{i,y_i}^{\mathrm{virt}} = v, \qquad z_{i,c}^{\mathrm{virt}} = z_{i,c}^g \ (c \neq y_i), \qquad \mathbf{p}_i^{\mathrm{virt}} = \mathrm{softmax}(\mathbf{z}_i^{\mathrm{virt}}).$$

One KD step is

$$\mathbf{z}_i^{s+} \leftarrow \mathbf{z}_i^s - \eta_u \, \nabla_{\mathbf{z}_i^s} \mathrm{KL}(\mathbf{p}_i^{\mathrm{virt}} \| \mathbf{p}_i^s), \qquad \eta_u > 0 \text{ small}.$$

## C.2    ASSUMPTIONS

We adopt two groups of assumptions: (A0)–(A4) are FedQUIT-specific and support the unlearning analysis and the perturbation bound; (A5)–(A8) are FedAvg assumptions in the spirit of Li et al. (2020) and are only needed when we analyze the recovery phase and its approximation to an ideal retraining baseline on the retained clients.

**FedQUIT-specific assumptions.**

(A0) **Initialization at the global model.** At the start of unlearning, the student is initialized at the current global model $w_t$:
$$w^s \leftarrow w_t, \quad \text{so that} \quad p_i^s = p_i^g \text{ on } D_u.$$

(A1) **Overconfidence gap on forget data.** There exists $\Delta_H > 0$ such that
$$\mathbb{E}_{(x,y)\sim D_u}\big[H(p(\cdot \mid x; w_r))\big] \geq \mathbb{E}_{(x,y)\sim D_u}\big[H(p(\cdot \mid x; w_t))\big] + \Delta_H,$$
i.e., the original model $w_t$ is strictly more confident than the retrained model $w_r$ on $D_u$ (empirically verified in Sec. 5.2).

(A2) **Teacher geometry.** The virtual teacher modifies only the true-class logit and preserves all non-true logits of the global model (temperature $\tau = 1$), as in equation 1.

(A3) **Small-step regime.** The unlearning step size $\eta_u$ is small enough that first-order Taylor expansions of the KD update in the student logits are accurate. All "$\approx$" steps in the local CE-change analysis are understood in this first-order sense.

(A4) **Bounded logit Jacobians.** There exists $G > 0$ such that
$$\|\nabla_w z(x; w)\|_{\mathrm{op}} \leq G \quad \text{for all } (x, w).$$
This assumption is only used in the bounded-perturbation argument (Lemma 2 / Proposition 4) to control $\|w^{\bar{u}} - w_t\|$ via the KD logit gradients.

**FedAvg recovery assumptions (after removing client $u$).** Let $F_k(w)$ denote the local objective of client $k$, $\pi_k$ the client weights, and
$$F_{\setminus u}(w) = \sum_{k \neq u} \pi_k F_k(w)$$
the global objective restricted to the retained clients. Let $F_{\setminus u}^* = \min_w F_{\setminus u}(w)$ and $F_k^* = \min_w F_k(w)$.

(A5) **Smoothness and strong convexity.** Each client objective $F_k$ (for $k \neq u$) is $L$-smooth and $\mu$-strongly convex:
$$\|\nabla F_k(w) - \nabla F_k(w')\| \leq L\|w - w'\| \quad \forall w, w',$$
$$F_k(v) \geq F_k(w) + \langle v - w, \nabla F_k(w)\rangle + \frac{\mu}{2}\|v - w\|^2 \quad \forall v, w.$$
We write $\kappa = L/\mu$.

(A6) **Stochastic gradient noise (as in Li et al. (2020), Assumptions 3–4).** Let $\xi_k = (x, y) \sim D_k$ and $g_k(w, \xi_k) = \nabla_w \ell(h(x; w), y)$ be a stochastic gradient on client $k$. There exist constants $\sigma_k^2$ and $G_{\mathrm{sgd}}^2$ such that, for all $k \neq u$ and all $w$,
$$\mathbb{E}_{\xi_k}\big\|g_k(w, \xi_k) - \nabla F_k(w)\big\|^2 \leq \sigma_k^2, \qquad \mathbb{E}_{\xi_k}\big\|g_k(w, \xi_k)\big\|^2 \leq G_{\mathrm{sgd}}^2.$$
The notation $G_{\mathrm{sgd}}$ to avoid confusion with the Jacobian bound $G$ in (A4).

(A7) **Bounded data heterogeneity.** We adopt the heterogeneity measure of Li et al. (2020), restricted to the retained clients:
$$\Gamma_{\setminus u} = F_{\setminus u}^* - \sum_{k \neq u} \pi_k F_k^* \geq 0.$$
In the IID case $\Gamma_{\setminus u} = 0$. We assume $\Gamma_{\setminus u} < \infty$, i.e., the mismatch between the global optimum and the local optima on the retained clients is bounded.

(A8) **FedAvg step-size schedule.** During the recovery phase, FedAvg on $F_{\setminus u}$ uses the step-size schedule of Li et al. (2020):
$$\eta_t = \frac{2}{\mu(\gamma + t)}, \qquad \gamma = \max\{8\kappa, E\},$$
where $E$ is the number of local SGD steps per round.

### C.3 Forgetting signal: statements and sketches

**Lemma 3** (Penalizing the true-class logit provides a forgetting signal). *Fix $(\mathbf{x}_i, y_i) \in D_u$ and define the evaluation CE $\ell_i = -\log p^s_{i,y_i}$. Construct the teacher by keeping all non-true logits of the global model and replacing the true-class logit with $v < z^g_{i,y_i}$ (Eq. 1). Under (A0)–(A3), after one KD step, if $p^s_{i,y_i} > p^{\mathrm{virt}}_{i,y_i}$ (e.g., at initialization), then*

$$\Delta\ell_i = \ell^+_i - \ell_i > 0 \qquad and \qquad p^{s+}_{i,y_i} < p^s_{i,y_i}.$$

*Thus moving the student toward the teacher strictly increases evaluation CE on the forget sample.*

*Sketch.* Using $\nabla_{\mathbf{z}^s}\ell_i = \mathbf{p}^s_i - \mathbf{e}_{y_i}$ and $\nabla_{\mathbf{z}^s}\mathrm{KL} = \mathbf{p}^s_i - \mathbf{p}^{\mathrm{virt}}_i$, a first-order expansion gives $\Delta\ell_i \approx -\eta_u\langle\nabla\ell_i, \nabla\mathrm{KL}\rangle$. If $p^s_{i,y_i} > p^{\mathrm{virt}}_{i,y_i}$, the inner product is negative, so $\Delta\ell_i > 0$ for small $\eta_u$. The same sign argument yields $p^{s+}_{i,y_i} < p^s_{i,y_i}$. $\qquad\square$

**Proposition 2** (Quantified forgetting for $v = \min_k z^g_{i,k}$). *Under (A0)–(A3), initialize at $\mathbf{p}^s_i = \mathbf{p}^g_i$ and choose $v = \min_k z^g_{i,k}$. After one KD step with step size $\eta_u > 0$,*

$$\ell^+_i - \ell_i = \eta_u \left(p^g_{i,y_i} - p^{\mathrm{virt}}_{i,y_i}\right)\left[(1 - p^g_{i,y_i}) + \frac{\sum_{c\neq y_i}(p^g_{i,c})^2}{1 - p^g_{i,y_i}}\right] + O(\eta^2_u),$$

*and since $p^{\mathrm{virt}}_{i,y_i} \leq 1/C$,*

$$\ell^+_i - \ell_i \geq \eta_u \frac{C}{C-1}\left(p^g_{i,y_i} - \frac{1}{C}\right)(1 - p^g_{i,y_i}) + O(\eta^2_u).$$

*If the student matches the teacher on $(\mathbf{x}_i, y_i)$, the asymptotic increase is $\ell^\star_i - \ell_i = \log\frac{p^g_{i,y_i}}{p^{\mathrm{virt}}_{i,y_i}} \geq \log(C\, p^g_{i,y_i})$.*

*Sketch.* Use $p^{\mathrm{virt}}_{i,c} = (1 - p^{\mathrm{virt}}_{i,y_i})\frac{p^g_{i,c}}{1-p^g_{i,y_i}}$ for $c \neq y_i$, expand $\Delta\ell_i$ at $\mathbf{p}^s_i = \mathbf{p}^g_i$, and note $v = \min_k z^g_{i,k} \Rightarrow p^{\mathrm{virt}}_{i,y_i} \leq 1/C$. $\qquad\square$

**Proposition 3** ($v = \min_k z^g_{i,k}$ dominates the max entropy $v^\star$). *Let $v_{\min} = \min_k z^g_{i,k}$ and let $v^\star$ be the value that maximizes the teacher's predictive entropy, i.e.,*

$$p^{\mathrm{virt}}_{i,y_i}(v^\star) = \frac{1}{1 + e^{H(r_i)}},$$

*where $r_i$ is the non-true conditional of the global model on $x_i$, $r_{i,c} = p^g_{i,c}/(1 - p^g_{i,y_i})$ for $c \neq y_i$, and $H(r_i) = -\sum_{c\neq y_i} r_{i,c}\log r_{i,c}$. Under (A0)–(A3), initializing the student at the global ($p^s_i = p^g_i$) and taking a small KD step of size $\eta_u > 0$ yields the first order change in evaluation cross entropy*

$$\Delta\ell_i(v) = \ell^+_i - \ell_i = \eta_u\left(p^g_{i,y_i} - p^{\mathrm{virt}}_{i,y_i}(v)\right)\left((1 - p^g_{i,y_i}) + \frac{\sum_{c\neq y_i}(p^g_{i,c})^2}{1-p^g_{i,y_i}}\right) + O(\eta^2_u).$$

*Since $p^{\mathrm{virt}}_{i,y_i}(v_{\min}) \leq \frac{1}{C} \leq p^{\mathrm{virt}}_{i,y_i}(v^\star)$, it follows that*

$$\Delta\ell_i(v_{\min}) \geq \Delta\ell_i(v^\star) \qquad (\text{strict whenever } p^{\mathrm{virt}}_{i,y_i}(v^\star) > \tfrac{1}{C}).$$

*Asymptotic comparison. If training on $(x_i, y_i)$ converges to the teacher $p^{\mathrm{virt}}_i(v)$, the evaluation CE on that sample changes from $-\log p^g_{i,y_i}$ to $-\log p^{\mathrm{virt}}_{i,y_i}(v)$, so the exact increase equals*

$$\left(-\log p^{\mathrm{virt}}_{i,y_i}(v)\right) - \left(-\log p^g_{i,y_i}\right) = \log\frac{p^g_{i,y_i}}{p^{\mathrm{virt}}_{i,y_i}(v)}.$$

*Because $p^{\mathrm{virt}}_{i,y_i}(v_{\min}) \leq \frac{1}{C} \leq p^{\mathrm{virt}}_{i,y_i}(v^\star)$, we also have*

$$\log\frac{p^g_{i,y_i}}{p^{\mathrm{virt}}_{i,y_i}(v_{\min})} \geq \log\frac{p^g_{i,y_i}}{p^{\mathrm{virt}}_{i,y_i}(v^\star)}.$$

*Thus, both in the local first order sense and in the teacher matching limit, $v_{\min}$ yields at least as large a forgetting signal as $v^\star$.*

**Lemma 4** (Teacher entropy is unimodal in $v$). *Fix $\{z_{i,c}^g\}_{c\neq y_i}$ and let $S = \sum_{c\neq y_i} e^{z_{i,c}^g}$, $r_c = \frac{e^{z_{i,c}^g}}{S}$, and $H(r_i) = -\sum_{c\neq y_i} r_c \log r_c$. With $p_y(v) = \frac{e^v}{e^v+S}$, the teacher entropy $H(v) = h(p_y(v)) + (1 - p_y(v))H(r_i)$ is strictly unimodal, maximized at $p_y(v^\star) = \frac{1}{1+e^{H(r_i)}}$, equivalently $v^\star = \sum_{c\neq y_i} r_c z_{i,c}^g$.*

*Sketch.* Differentiate $H(v)$ using $dp_y/dv = p_y(1 - p_y)$ to obtain $\frac{dH}{dv} = p_y(1 - p_y)\left[\log\frac{1-p_y}{p_y} - H(r_i)\right]$. The unique stationary point is a strict maximum. $\square$

**Remarks.** *Aggressiveness of $v$.* For geometry-preserving teachers, $p_{i,y_i}^{\mathrm{virt}}(v) = \frac{e^v}{e^v+S}$ increases strictly in $v$; the gap $p_{i,y_i}^g - p_{i,y_i}^{\mathrm{virt}}(v)$ and the first-order CE rise therefore grow as $v$ decreases.

*Why $p_{i,y_i}^{\mathrm{virt}} \leq 1/C$ for $v = \min z^g$.* Since $z_{i,c}^g \geq v$ for all $c \neq y_i$, $\sum_{c\neq y_i} e^{z_{i,c}^g} \geq (C-1)e^v$, hence $p_{i,y_i}^{\mathrm{virt}}(v) = \frac{e^v}{e^v+\sum_{c\neq y_i} e^{z_{i,c}^g}} \leq \frac{1}{C}$.

### C.4 BOUNDED PERTURBATION AND FEDAVG CONVERGENCE

We now show that FedQUIT induces a controlled change in parameters and that FedAvg, run on the remaining clients, converges from the resulting snapshot.

**Setup.** At the unlearning round, client $u$ receives $w_t$, constructs a virtual teacher by setting the true-class logit to $v$ and keeping all non-true logits, and trains the student (initialized at $w_t$) by minimizing $\mathrm{KL}(p^{\mathrm{virt}}\|p^s)$ over $T_u$ updates with learning rate $\eta_u$, yielding $w^{\bar{u}}$. The server then removes client $u$ and resumes FedAvg on $\{k \neq u\}$ from $w^{\bar{u}}$.

**Notation.** Let $z_i^g$, $p_i^g$ be global logits and probabilities; $z_i^s$, $p_i^s$ the student's; $e_{y_i}$ the basis vector at $y_i$. Define the non-true conditional

$$r_{i,c} := \frac{e^{z_{i,c}^g}}{\sum_{j\neq y_i} e^{z_{i,j}^g}} \ (c \neq y_i), \quad p_{i,c}^g = (1 - p_{i,y_i}^g)\, r_{i,c}, \ \sum_{c\neq y_i} r_{i,c} = 1,$$

and set $S_i = \sum_{c\neq y_i} e^{z_{i,c}^g}$, $p_y(v) = \frac{e^v}{e^v+S_i}$, so that

$$p_{i,y_i}^{\mathrm{virt}} = p_y(v), \qquad p_{i,c}^{\mathrm{virt}} = (1 - p_y(v))\, r_{i,c} \ (c \neq y_i).$$

Let $g_i := p_{i,y_i}^g - p_{i,y_i}^{\mathrm{virt}} \geq 0$ and denote minibatches by $\mathcal{B}_t$ of size $B$.

**Lemma 5** (Logit-gradient structure and norm). *For $\mathcal{L}_{\mathrm{KD}} = \mathrm{KL}(p_i^{\mathrm{virt}}\|p_i^s)$,*

$$\nabla_{z_i^s}\mathcal{L}_{\mathrm{KD}} = p_i^s - p_i^{\mathrm{virt}}.$$

*If $p_i^s = p_i^g$,*

$$\nabla_{z_i^s}\mathcal{L}_{\mathrm{KD}} = g_i\, e_{y_i} - g_i\, r_i, \qquad \left\|\nabla_{z_i^s}\mathcal{L}_{\mathrm{KD}}\right\|_2 = |g_i|\sqrt{1 + \|r_i\|_2^2} \leq \sqrt{2}\,|g_i| \leq \sqrt{2}\, p_{i,y_i}^g.$$

*Sketch.* Standard gradient identities; substitute the decompositions for non-true components and use $\|r_i\|_2 \leq 1$. $\square$

**Proposition 4** (FedQUIT induces a bounded parameter perturbation). *Assume (A4). If client $u$ performs $T_u$ KD updates with step size $\eta_u$ on minibatches $\{\mathcal{B}_t\}_{t=1}^{T_u}$,*

$$\|w^{\bar{u}} - w_t\| \leq \eta_u\, G \sum_{t=1}^{T_u} \left\|\frac{1}{B}\sum_{i\in\mathcal{B}_t}\left(p_i^s - p_i^{\mathrm{virt}}\right)\right\|_2 \leq \eta_u\, G\sqrt{2}\sum_{t=1}^{T_u}\overline{g}_t,$$

*where $\overline{g}_t := \frac{1}{B}\sum_{i\in\mathcal{B}_t}|p_{i,y_i}^s - p_{i,y_i}^{\mathrm{virt}}|$. At initialization $p_i^s = p_i^g$, $\overline{g}_t \leq 1$, hence*

$$\|w^{\bar{u}} - w_t\| \leq \eta_u\, G\sqrt{2}\, T_u.$$

*If $v = \min_c z_{i,c}^g$, then $p_{i,y_i}^{\mathrm{virt}} \leq \frac{1}{C}$ and $\overline{g}_t \leq \frac{1}{B}\sum_{i\in\mathcal{B}_t}\left(p_{i,y_i}^g - \frac{1}{C}\right)_+ \leq 1$, so the same bound holds and is typically tighter. Here $(a)_+ := \max\{a, 0\}$.*

*Sketch.* A KD step satisfies $\|w^+ - w\| \leq \eta_u \|\nabla_w z\|_{\mathrm{op}} \cdot \big\| \frac{1}{B} \sum_{i \in \mathcal{B}} (p_i^s - p_i^{\mathrm{virt}}) \big\|_2$ by the chain rule and Jensen. Apply Lemma 5 and sum over steps. □

**Theorem 2** (FedAvg convergence after resuming from $w^{\bar{u}}$). *Let* $F_{\setminus u}(w) = \sum_{k \neq u} \pi_k \mathbb{E}_{(x,y) \sim D_k}[\ell(h(x; w), y)]$ *be the objective after removing client* $u$. *Assume (A5)-(A8). Starting from the FedQUIT snapshot* $w^{\bar{u}}$, *the iterates converge to a stationary point of* $F_{\setminus u}$ *with the same rate as from* $w_t$, *with only an initial potential shift*

$$F_{\setminus u}(w^{\bar{u}}) - F_{\setminus u}(w_t) \leq \langle \nabla F_{\setminus u}(w_t), w^{\bar{u}} - w_t \rangle + \frac{L}{2} \|w^{\bar{u}} - w_t\|^2,$$

*which is controlled by Proposition 4.*

**Interpretation.** Because the teacher alters only the true-class logit while preserving non-true logits, the KD gradient at initialization has a one-dimensional structure $g_i \, e_{y_i} - g_i \, r_i$ with norm proportional to the true-class gap $g_i$. Via the bounded logit Jacobian, this yields an explicit bound on $\|w^{\bar{u}} - w_t\|$. Hence resuming FedAvg from $w^{\bar{u}}$ is a warm start near $w_t$, and standard convergence guarantees (Li et al., 2020) apply once client $u$ is excluded.

### C.5 APPROXIMATION TO RETRAINING

**Setup.** Recall the recovery-phase assumptions (A5)–(A8) and the FedAvg setup on

$$F_{\setminus u}(w) = \sum_{k \neq u} \pi_k \, \mathbb{E}_{(x,y) \sim D_k} \big[ \ell(h(x; w), y) \big] = \sum_{k \neq u} \pi_k F_k(w),$$

with unique minimizer $w_{\setminus u}^*$, condition number $\kappa = L/\mu$, and step-size schedule

$$\eta_t = \frac{2}{\mu(\gamma + t)}, \qquad \gamma = \max\{8\kappa, E\},$$

where $E$ is the number of local SGD steps per communication round.

Under (A5)–(A8) and this schedule, the FedAvg iterates on $F_{\setminus u}$ satisfy the generic convergence bound (Li et al., 2020).

$$\mathbb{E}\big[F_{\setminus u}(w_T)\big] - F_{\setminus u}^* \leq \frac{C_0}{\gamma + T}\Big(C_1 + \|w_0 - w_{\setminus u}^*\|^2\Big), \tag{3}$$

for some constants $C_0, C_1 > 0$ depending only on $(L, \mu, \{\sigma_k\}_{k \neq u}, G_{\mathrm{sgd}}, \Gamma_{\setminus u})$ and on the FedAvg hyperparameters.

On the FedQUIT side, Proposition 4 shows that the on-device KD phase produces a bounded perturbation of the current global model $w_t$. In particular, there exists a finite constant $\Delta_u > 0$ such that

$$\|w^{\bar{u}} - w_t\| \leq \Delta_u. \tag{4}$$

Under the conditions of Proposition 4 we can take, for example, $\Delta_u = \eta_u G \sqrt{2} \, T_u$, where $\eta_u$ is the unlearning step size, $T_u$ the number of KD steps, and $G$ the Jacobian bound from (A4).

**Definition 1** (Retraining baseline). *Fix an initialization* $w_0$. *The retraining baseline runs FedAvg on* $F_{\setminus u}$, *i.e., on clients* $k \neq u$ *only, starting from* $w_0$ *for a total of* $T$ *communication rounds, using the step-size schedule equation 3. We denote the resulting model by* $w_T^{\mathrm{re}}$.

**Definition 2** (FedQUIT baseline). *In contrast, FedQUIT executes three phases: (i) standard FedAvg on all clients up to round* $t$, *producing a model* $w_t$; *(ii) a FedQUIT unlearning round on client* $u$, *starting from* $w_t$ *and returning* $w^{\bar{u}}$; *(iii) FedAvg on the retained clients* $k \neq u$ *starting from* $w^{\bar{u}}$ *for* $T$ *recovery rounds, using the step-size schedule equation 3. We denote the resulting model by* $w_T^{\mathrm{FQ}}$.

**Theorem 3** (FedQUIT approximates retraining). *Assume (A5)–(A8). Let* $w_T^{\mathrm{re}}$ *and* $w_T^{\mathrm{FQ}}$ *be as in Definitions 1 and 2, respectively. Then there exist constants* $C_1', C_2' > 0$ *such that*

$$\mathbb{E}\big[F_{\setminus u}(w_T^{\mathrm{FQ}})\big] - \mathbb{E}\big[F_{\setminus u}(w_T^{\mathrm{re}})\big] \leq \frac{C_1'}{\gamma + T} \Delta_u^2 + \frac{C_2'}{(\gamma + t)(\gamma + T)}. \tag{5}$$

*In particular, if $\Delta_u$ is chosen as in equation 4 with $\Delta_u^2 = \mathcal{O}(\eta_u^2 G^2 T_u)$, the dominant term in equation 5 satisfies*

$$\mathbb{E}\big[F_{\backslash u}(w_T^{\mathrm{FQ}})\big] - \mathbb{E}\big[F_{\backslash u}(w_T^{\mathrm{re}})\big] \;=\; \mathcal{O}\!\left(\frac{\eta_u^2 G^2 T_u}{\gamma + T}\right). \tag{6}$$

*Moreover, the expected squared parameter distance obeys*

$$\mathbb{E}\big\|w_T^{\mathrm{FQ}} - w_T^{\mathrm{re}}\big\|^2 \;\leq\; \frac{C_3}{\gamma + T}\left(1 + \Delta_u^2\right) \tag{7}$$

*for some constant $C_3 > 0$ depending only on $(L, \mu, C_0, C_1)$ and on the gradient-noise and heterogeneity parameters.*

*Proof.* Apply the generic convergence bound equation 3 to the retraining baseline with initialization $w_0$ and output $w_T^{\mathrm{re}}$:

$$\mathbb{E}\big[F_{\backslash u}(w_T^{\mathrm{re}})\big] - F_{\backslash u}^* \;\leq\; \frac{C_0}{\gamma + T}\Big(C_1 + \|w_0 - w_{\backslash u}^*\|^2\Big). \tag{8}$$

Next, apply equation 3 to the FedQUIT recovery phase, initialized at $w_0^{\mathrm{FQ}} = w^{\bar{u}}$ with output $w_T^{\mathrm{FQ}}$:

$$\mathbb{E}\big[F_{\backslash u}(w_T^{\mathrm{FQ}})\big] - F_{\backslash u}^* \;\leq\; \frac{C_0}{\gamma + T}\Big(C_1 + \|w^{\bar{u}} - w_{\backslash u}^*\|^2\Big). \tag{9}$$

Subtracting equation 8 from equation 9 gives

$$\mathbb{E}\big[F_{\backslash u}(w_T^{\mathrm{FQ}})\big] - \mathbb{E}\big[F_{\backslash u}(w_T^{\mathrm{re}})\big] \;\leq\; \frac{C_0}{\gamma + T}\Big(\|w^{\bar{u}} - w_{\backslash u}^*\|^2 - \|w_0 - w_{\backslash u}^*\|^2\Big). \tag{10}$$

We now control $\|w^{\bar{u}} - w_{\backslash u}^*\|^2$ using the FedQUIT perturbation and the distance of $w_t$ to the optimum. By the triangle inequality and equation 4,

$$\|w^{\bar{u}} - w_{\backslash u}^*\| \;\leq\; \|w^{\bar{u}} - w_t\| + \|w_t - w_{\backslash u}^*\| \;\leq\; \Delta_u + \|w_t - w_{\backslash u}^*\|.$$

Using $(a+b)^2 \leq 2a^2 + 2b^2$,

$$\|w^{\bar{u}} - w_{\backslash u}^*\|^2 \;\leq\; 2\Delta_u^2 + 2\|w_t - w_{\backslash u}^*\|^2. \tag{11}$$

By $\mu$-strong convexity of $F_{\backslash u}$,

$$F_{\backslash u}(w_t) - F_{\backslash u}^* \;\geq\; \frac{\mu}{2}\,\|w_t - w_{\backslash u}^*\|^2,$$

so

$$\|w_t - w_{\backslash u}^*\|^2 \;\leq\; \frac{2}{\mu}\left(F_{\backslash u}(w_t) - F_{\backslash u}^*\right). \tag{12}$$

Applying equation 3 at time $t$ yields

$$\mathbb{E}\big[F_{\backslash u}(w_t)\big] - F_{\backslash u}^* \;\leq\; \frac{\tilde{C}_0}{\gamma + t}\Big(\tilde{C}_1 + \|w_0 - w_{\backslash u}^*\|^2\Big),$$

which, combined with equation 12, gives

$$\mathbb{E}\|w_t - w_{\backslash u}^*\|^2 \;\leq\; \frac{\tilde{C}_2}{\gamma + t} \tag{13}$$

for a constant $\tilde{C}_2 > 0$.

Plugging equation 13 into equation 11 and taking expectations,

$$\mathbb{E}\|w^{\bar{u}} - w_{\backslash u}^*\|^2 \;\leq\; 2\Delta_u^2 + \frac{2\tilde{C}_2}{\gamma + t}. \tag{14}$$

Substituting equation 14 into equation 10 and dropping the non-positive term $-\|w_0 - w_{\setminus u}^*\|^2$ yields

$$\mathbb{E}\big[F_{\setminus u}(w_T^{\mathrm{FQ}})\big] - \mathbb{E}\big[F_{\setminus u}(w_T^{\mathrm{re}})\big] \;\leq\; \frac{C_0}{\gamma + T}\left(2\Delta_u^2 + \frac{2\tilde{C}_2}{\gamma + t}\right).$$

Renaming constants as $C_1' = 2C_0$ and $C_2' = 2C_0\tilde{C}_2$ gives equation 5. The simplified form equation 6 follows by substituting $\Delta_u^2 = \mathcal{O}(\eta_u^2 G^2 T_u)$.

For the parameter distance, use

$$\|w_T^{\mathrm{FQ}} - w_T^{\mathrm{re}}\|^2 \;\leq\; 2\|w_T^{\mathrm{FQ}} - w_{\setminus u}^*\|^2 + 2\|w_T^{\mathrm{re}} - w_{\setminus u}^*\|^2,$$

combine strong convexity of $F_{\setminus u}$ with equation 3 applied to each run, and absorb constants into $C_3 > 0$ to obtain equation 7. $\qquad\square$

**Interpretation.** Under Assumptions (A4)–(A8), FedQUIT provides a warm start close to $w_t$ for the FedAvg run restricted to the retained clients. The FedQUIT unlearning phase induces a bounded perturbation of the snapshot,

$$\|w^{\bar{u}} - w_t\| \;\leq\; \Delta_u, \qquad \Delta_u \;\leq\; \eta_u\, G\, \sqrt{2}\, T_u,$$

as established in Proposition 4. Subsequently, FedAvg on $\{k \neq u\}$ converges according to the same rate as in the retraining baseline, and the deviation from retraining is controlled by this perturbation. In particular, the expected weight divergence satisfies

$$\mathbb{E}\big\|w_T^{\mathrm{FQ}} - w_T^{\mathrm{re}}\big\|^2 \;\leq\; \frac{C_3}{\gamma + T}\,(1 + \Delta_u^2),$$

showing that the gap between FedQUIT and ideal retraining decays at rate $O(1/(\gamma+T))$ and remains proportionally small throughout recovery.

## D EXTENDED EVALUATION AND FURTHER RESULTS

We run all the experiments on a machine with Ubuntu 22.04, equipped with 64 GB of RAM and one NVIDIA RTX A5000 as GPU (32GB memory). We developed the code with Python and with Python libraries; in our code repository, we provide the instructions to exactly reproduce our Python environment.

In this section, we provide further detail and experimental results that, for the sake of space, have been excluded from the main text. Table 6 reports all the settings we consider for client and sample unlearning.

**Overview.** This section provides additional experimental results and implementation details omitted from the main paper for space. Specifically, we include: (i) extended after-recovery results with MiT-B0 and with more local epochs (Sec. D.1); (ii) full baseline descriptions (Sec. D.2); (iii) full post-unlearning and post-recovery results to complete the main paper Table 1 (Sec. D.4); (iv) communication, computation, and storage cost estimation methodology (Sec. D.3); (v) fixed-budget evaluations at 5–50% retraining cost (Sec. D.5); (vi) sample-unlearning experiments for 50%, 10%,

| Model | Dataset | Clients | Non-IIDness | Pre-trained | Local Epochs |
|---|---|---|---|---|---|
| ResNet-18 | CIFAR-10 | 10 | LDA ($\alpha$=0.3) | No | 1 |
| ResNet-18 | CIFAR-100 | 10 | LDA ($\alpha$=0.1) or IID | No | 1 or 10 |
| MiT-B0 | CIFAR-100 | 10 | LDA ($\alpha$=0.1) | Yes | 1 |
| MiT-B0 | CUB-200 | 10 | LDA ($\alpha$=0.1) | Yes | 1 |
| ResNet-18 | CIFAR-100 | 100 | LDA ($\alpha$=0.1) | No | 1 |
| MiT-B0 | CUB-200 | 100 | LDA ($\alpha$=0.1) | Yes | 1 |
| Simple LSTM | Tiny-Shakespeare | 80 | Contiguous Chunks | No | 1 |

Table 6: Federated settings of reported results.

and 1% forget sets, including local retain accuracy (Sec. D.6); (vii) ablations on distillation loss (Sec. D.7) and virtual-teacher structure, including temperature effects (Sec. D.8); (viii) membership-inference attack definitions (Sec. D.10); and (ix) hyper-parameter tuning and preprocessing details for all methods.

## D.1 Further Results with MiT-B0 and with More Local Epochs

| Setting | Method | Efficacy | | | | | | Efficiency | | |
|---|---|---|---|---|---|---|---|---|---|---|
| | | Test Acc. | Retain Acc. ($\Delta \downarrow$) | Forget Acc. ($\Delta \downarrow$) | MIA$_{[Song]}$ ($\Delta \downarrow$) | MIA$_{[Ycom]}$ ($\Delta \downarrow$) | Avg. Gap ($\downarrow$) | Communication Bytes ($\times \uparrow$) | Computation FLOPs ($\times \uparrow$) | Storage Bytes ($\times \uparrow$) |
| CIFAR-100, Non-IID, MiT-B0, E=1, 10 Clients | Original | $75.0_{\pm0.0}$ | $84.7_{\pm0.3}$ | $84.3_{\pm5.6}$ | $77.0_{\pm8.6}$ | $74.0_{\pm6.4}$ | | — | — | — |
| | Retrain | $73.3_{\pm0.8}$ | $86.2_{\pm0.6}$ | $57.8_{\pm5.3}$ | $48.6_{\pm5.0}$ | $44.6_{\pm3.9}$ | | $1.19e^{10}$ (1.0×) | $3.82e^{15}$ (1.0×) | $1.33e^{07}$ (1.0×) |
| | PGA | $73.6_{\pm0.4}$ | $86.0\ (0.2_{\pm0.2})$ | $62.2\ (4.4_{\pm3.3})$ | $53.6\ (5.0_{\pm3.7})$ | $50.5\ (6.9_{\pm3.2})$ | 4.1 | $\mathbf{1.68e^{09}}$ (**7.1×**) | $\mathbf{5.70e^{14}}$ (**6.7×**) | $1.46e^{08}$ (0.0909×) |
| | MoDe | $73.4_{\pm0.8}$ | $84.7\ (1.5_{\pm0.9})$ | $62.9\ (5.2_{\pm2.2})$ | $54.7\ (6.1_{\pm3.1})$ | $50.3\ (5.9_{\pm2.9})$ | 4.7 | $5.55e^{09}$ (2.1×) | $1.78e^{15}$ (2.2×) | $2.66e^{07}$ (0.5×) |
| | FedAU | $73.6_{\pm0.7}$ | $86.0\ (0.2_{\pm0.4})$ | $63.7\ (5.9_{\pm1.9})$ | $55.2\ (5.9_{\pm2.3})$ | $50.2\ (5.6_{\pm1.3})$ | 4.4 | $3.54e^{09}$ (3.4×) | $1.13e^{15}$ (3.4×) | $1.33e^{07}$ (1.0×) |
| | NoT | $73.4_{\pm0.7}$ | $86.6\ (0.4_{\pm0.8})$ | $67.9\ (10.1_{\pm4.1})$ | $59.4\ (10.1_{\pm6.6})$ | $54.0\ (9.4_{\pm5.4})$ | 7.5 | $3.70e^{09}$ (3.2×) | $1.19e^{15}$ (3.2×) | $1.33e^{07}$ (1.0×) |
| | FedOSD | $73.5_{\pm0.7}$ | $86.4\ (0.2_{\pm0.6})$ | $59.8\ (2.0_{\pm1.8})$ | $51.3\ (2.7_{\pm2.0})$ | $46.6\ (2.0_{\pm1.5})$ | 1.7 | $3.11e^{09}$ (3.8×) | $9.94e^{14}$ (3.8×) | $2.66e^{07}$ (0.5×) |
| | FedQUIT | $73.4_{\pm0.9}$ | $\mathbf{86.3\ (0.1}_{\pm0.5})$ | $\mathbf{58.5\ (1.9}_{\pm1.5})$ | $\mathbf{48.9\ (2.6}_{\pm2.1})$ | $\mathbf{46.4\ (1.8}_{\pm1.1})$ | **1.6** | $2.13e^{09}$ (5.6×) | $6.82e^{14}$ (5.6×) | $1.33e^{07}$ (1.0×) |
| CIFAR-100, Non-IID, ResNet-18, E=10 | Original | $50.3_{\pm0.0}$ | $61.7_{\pm0.8}$ | $60.8_{\pm6.6}$ | $73.4_{\pm6.6}$ | $61.3_{\pm7.5}$ | | — | — | — |
| | Retrain | $48.0_{\pm0.8}$ | $63.6_{\pm0.7}$ | $31.6_{\pm4.5}$ | $42.1_{\pm5.2}$ | $31.2_{\pm4.9}$ | | $1.62e^{11}$ (1.0×) | $1.35e^{16}$ (1.0×) | $4.49e^{07}$ (1.0×) |
| | PGA | $48.3_{\pm1.0}$ | $61.9\ (1.7_{\pm0.4})$ | $34.7\ (3.1_{\pm3.9})$ | $45.3\ (3.2_{\pm4.0})$ | $33.5\ (2.3_{\pm3.7})$ | 2.6 | $3.64e^{09}$ (44.5×) | $3.01e^{14}$ (44.9×) | $4.94e^{08}$ (0.09×) |
| | MoDe | $48.1_{\pm0.9}$ | $60.7\ (2.9_{\pm1.2})$ | $35.3\ (3.9_{\pm3.6})$ | $45.4\ (4.7_{\pm4.1})$ | $34.0\ (3.6_{\pm3.3})$ | 3.8 | $2.10e^{10}$ (7.7×) | $1.76e^{15}$ (7.7×) | $8.98e^{07}$ (0.5×) |
| | FedAU | $48.2_{\pm1.0}$ | $58.7\ (4.9_{\pm0.3})$ | $36.8\ (4.5_{\pm3.7})$ | $47.1\ (6.1_{\pm3.6})$ | $34.7\ (2.8_{\pm3.5})$ | 4.6 | $1.13e^{10}$ (14.3×) | $9.45e^{14}$ (14.3×) | $4.49e^{07}$ (1.0×) |
| | NoT | $48.3_{\pm0.7}$ | $56.1\ (7.5_{\pm0.2})$ | $39.7\ (8.1_{\pm3.6})$ | $48.9\ (6.8_{\pm4.4})$ | $37.2\ (6.0_{\pm3.9})$ | 7.1 | $6.22e^{09}$ (26.0×) | $5.20e^{14}$ (26.0×) | $4.49e^{07}$ (1.0×) |
| | FedOSD | $48.2_{\pm0.9}$ | $62.9\ (0.7_{\pm0.9})$ | $34.8\ (3.2_{\pm3.0})$ | $46.0\ (3.9_{\pm2.1})$ | $33.9\ (2.7_{\pm2.0})$ | 2.6 | $9.69e^{09}$ (16.7×) | $8.10e^{14}$ (16.7×) | $8.98e^{07}$ (0.5×) |
| | FedQUIT | $48.3_{\pm1.0}$ | $\mathbf{63.3\ (0.3}_{\pm0.5})$ | $\mathbf{32.7\ (1.3}_{\pm3.3})$ | $\mathbf{44.8\ (2.3}_{\pm3.5})$ | $\mathbf{33.0\ (2.1}_{\pm3.1})$ | 1.5 | $\mathbf{1.46e^{09}}$ (**110.7×**) | $\mathbf{1.16e^{14}}$ (**116.9×**) | $4.49e^{07}$ (1.0×) |

Table 7: After-recovery performance of FedQUIT and other baselines for the setting CIFAR-100, MiT-B0 and with more local epochs during regular training for the setting CIFAR-100, ResNet-18 (10 per-round local epochs before unlearning). Results are expressed as *mean metric value (mean $\Delta$ $\pm$ standard deviation)*, with $\Delta$ representing the average absolute difference with *Retrain*. Lower $\Delta$ corresponds to better unlearning. The *Avg. Gap* column reports the average across $\Delta$. For efficiency metrics, higher $\times$ (reduction over *Retrain* baseline) are better.

Table 7 shows after-recovery results for two additional settings: (1) a setting with CIFAR-100 (Non-IID) and MiT-B0 as model architecture; (2) when the global model is trained with 10 local epochs per round instead of 1, for the setting with CIFAR-100 (Non-IID) and ResNet-18 as model architecture. The reported results confirm the trends already emerged in Table 1 of the main paper.

| Method | Comm. | Comp. | Storage | Symbols |
|---|---|---|---|---|
| Retrain | $2PBK_rR$ | $FNEK_rR$ | $PB$ | $P$: model params |
| | | | | $B$: bytes per param |
| | | | | $K$: clients |
| | | | | $R$: rounds |
| | | | | $F$: FLOPs per sample |
| | | | | $N$: samples per client |
| | | | | $E$: local epochs |
| | | | | $K_r$: retain clients |
| FedEraser | $2PBK_rR_{\mathrm{ret}}$ | $FNE_{\mathrm{cal}}K_rR_{\mathrm{ret}}$ | $KR_{\mathrm{ret}}PB$ | $R_{\mathrm{ret}}$: stored rounds |
| | | | | $E_{\mathrm{cal}}$: calibration epochs |
| PGA | $2PB$ | $FNE_{\mathrm{asc}}$ | $KPB$ | $E_{\mathrm{asc}}$: ascent epochs |
| MoDe | $2PB(K_rR_d + KR_m)$ | $FNE(K_rR_d + KR_m)$ | $2PB$ | $R_d$: degradation rounds |
| | | | | $R_m$: memory guidance rounds |
| FedAU | $P_cBR$ | $F_cNE$ | $PB$ | $P_c$: classifier params |
| | | | | $F_c$: classifier FLOPs |
| NoT | negligible | negligible | $PB$ | – |
| FedOSD | $2PBKR_u$ | $FNEKR_u$ | $PB$ | $R_u$: unlearning rounds |
| FedQUIT | $2PB$ | $FNE_u$ | $PB$ | $E_u$: unlearning epochs |
| Recovery (all baselines) | $2PBK_rR_{\mathrm{rec}}$ | $FNEK_rR_{\mathrm{rec}}$ | $PB$ | $R_{rec}$: recovery rounds |

Table 8: Per-method costs of the unlearning phase to unlearn one client (excluding the recovery phase, which is common to all methods) and cost of recovery phase for all methods (last row). Symbols are indicated in the last column.

## D.2 Baselines in Experiments

**GA:** Unconstrained gradient-ascent method. It performs local gradient-ascent on forget data, and training resumes from the unlearned model returned by the target client. The results for GA are reported in Table 10 and in Table 17.

**PGA (Halimi et al., 2022):** The work in (Halimi et al., 2022) applies projected gradient ascent (PGA) to eliminate the influence of the data to be forgotten, directly on-device (similarly to our method). PGA constrains the local weight updates to ensure that the model weights remain within an $l2$-norm ball around a carefully selected reference model. The reference model is computed by removing the last unlearning client's model update from the current global model. Halimi et al. perform early stopping if the distance between the unlearned model and the reference model is below a threshold, which must be tuned. **Note that:** (1) PGA requires clients to store their last model updates to build the so-called *reference model*, and, to do that, it maintains a state on clients; (2) in light of the previous point, PGA needs full participation of clients; (3) PGA needs to tune two thresholds (plus the standard training hyper-parameters).

**MoDe (Zhao et al., 2023):** When an unlearning request is issued, a multi-round, two-stage un-learning phase starts. The two stages are momentum degradation and memory guidance. Note that during the memory guidance (spanning multiple communication rounds, e.g. 5–10 rounds) the target client must be connected and actively performs training.

**NoT (Khalil et al., 2025):** When an unlearning request is issued, the server flips the sign of the global model's first-layer parameters and continues training from this modified model. Note that NoT yields the same unlearned model, regardless of which client is designated as the target.

**FedAU (Gu et al., 2024):** During regular training, FedAU learns an auxiliary module, which usu-ally has the same architecture of the classifier head of the local model. This module is trained round-by-round on local forget data with random labels, and on local retain data with original labels (in client unlearning case, local retain data are not available). When an unlearning request is issued, the server aggregates the parameters of the auxiliary module with the corresponding parameters of the global model (e.g., the classificaiton head) with a linear combination ruled by a factor $\lambda_{FedAU}$.

**FedOSD (Pan et al., 2025):** It introduces a two-stage, multi-round FU method: first, it enforces forgetting by training the target client with a custom Cross-Entropy and updating the server via an orthogonal steepest-descent direction; then, it post-trains while projecting retained gradients (or-thogonal to the reversion vector) to prevent conflicts and reversion. Note that, similarly to MoDe, FedOSD requires that the target client remains connected and active for the entire unlearning phase, which spans multiple communication rounds.

**Incompetent Teacher.** As a baseline for FedQUIT, we also implemented a naive solution that uses an incompetent teacher. This alternative does not use the global model at round $t$ to craft the teacher's output probabilities for unlearning samples, but it always outputs equally distributed probabilities over classes as an incompetent teacher, i.e., $\mathbf{p}^{\text{virt}}(c) = 1/C$. This baseline helped us investigate whether using the modified global model's output as a teacher is beneficial. An incompetent teacher can also be implemented by using an untrained model, which produces random predictions.

### D.3 COST ESTIMATION METHODOLOGY

Table 8 summarizes the per-method cost estimation we used. We report three complementary ef-ficiency metrics: *communication cost* (bytes), *computational cost* (FLOPs), and *storage overhead* (bytes). These metrics, preferred over wall clock time as in Khalil et al. (2025), are hardware and implementation agnostic, enabling fair comparisons across tasks and systems. They isolate algo-rithmic efficiency from factors such as device heterogeneity or parallelization. For each method we account for both the unlearning phase and the recovery phase. The recovery phase (for all methods) consists of standard FedAvg rounds among the $K_r$ retain clients until utility is restored.

| Setting | Method | Test Accuracy | | Retain Accuracy | | Forget Accuracy | | Unlearning Cost | | Unlearning+Recovery Cost | |
|---|---|---|---|---|---|---|---|---|---|---|---|
| | | After Unlearning | After Recovery | After Unlearning | After Recovery | After Unlearning | After Recovery | Comm. (Bytes) | Comp. (FLOPs) | Comm. (Bytes) | Comp. (FLOPs) |
| **CIFAR-100**, IID, ResNet-18, $E=1$ | Retrain | 58.3 ± 0.5 | | 82.7 ± 0.8 | | 58.0 ± 0.7 | | | – | $1.62e^{11}$ | $1.35e^{15}$ |
| | GA | 1.9 ± 0.1 | 58.5 ± 0.4 | 1.9 ± 0.2 | 74.7 ± 0.2 | 2.0 ± 0.5 | 76.1 ± 3.3 | $8.98e^{07}$ | $7.50e^{11}$ | $1.71e^{09}$ | $1.43e^{13}$ |
| | PGA | 1.0 ± 0.2 | 59.1 ± 0.6 | 1.0 ± 0.2 | 76.2 ± 0.8 | 1.0 ± 0.2 | 72.8 ± 1.0 | $8.98e^{07}$ | $3.75e^{12}$ | $9.86e^{09}$ | $8.54e^{13}$ |
| | MoDe† | 36.8 ± 5.6 | 58.7 ± 0.5 | 40.2 ± 0.9 | 73.9 ± 1.0 | 41.5 ± 6.5 | 69.0 ± 0.7 | $1.38e^{10}$ | $1.16e^{14}$ | $1.75e^{10}$ | $1.46e^{14}$ |
| | FedAU | 4.2 ± 1.3 | 59.0 ± 0.5 | 5.1 ± 1.2 | 75.5 ± 0.9 | 5.3 ± 1.2 | 74.7 ± 1.0 | $8.21e^{07}$ | $1.84e^{12}$ | $2.67e^{09}$ | $2.23e^{13}$ |
| | NoT | 12.2 ± 0.0* | 58.9 ± 0.4 | 10.3 ± 0.2 | 75.2 ± 0.6 | 13.4 ± 1.8 | 70.5 ± 0.8 | 0 | 0 | $5.25e^{09}$ | $4.39e^{13}$ |
| | FedOSD† | 4.5 ± 1.6 | 58.8 ± 0.5 | 4.0 ± 0.8 | 76.5 ± 0.9 | 3.8 ± 2.0 | 71.1 ± 0.9 | $8.08e^{09}$ | $6.75e^{13}$ | $1.05e^{10}$ | $8.78e^{13}$ |
| | FedQUIT | 25.5 ± 1.1 | 58.7 ± 0.4 | 27.2 ± 1.3 | 76.6 ± 0.8 | 23.5 ± 1.6 | 60.4 ± 1.1 | $8.98e^{07}$ | $7.50e^{11}$ | $8.17e^{09}$ | $6.82e^{13}$ |
| **CIFAR-100**, Non-IID, ResNet-18, $E=1$ | Retrain | 51.0 ± 1.4 | | 64.6 ± 1.2 | | 33.5 ± 4.5 | | | – | $1.62e^{11}$ | $1.35e^{15}$ |
| | GA | 1.0 ± 0.1 | 51.2 ± 0.7 | 1.0 ± 0.1 | 60.1 ± 1.5 | 1.0 ± 0.1 | 51.2 ± 7.6 | $8.98e^{07}$ | $7.50e^{11}$ | $1.71e^{09}$ | $1.43e^{13}$ |
| | PGA | 1.0 ± 0.2 | 51.4 ± 0.6 | 1.0 ± 0.2 | 62.9 ± 1.3 | 1.0 ± 0.2 | 37.4 ± 4.1 | $8.98e^{07}$ | $3.75e^{12}$ | $5.74e^{09}$ | $5.10e^{13}$ |
| | MoDe† | 17.5 ± 3.6 | 50.9 ± 1.1 | 15.8 ± 3.3 | 61.8 ± 1.3 | 4.5 ± 2.7 | 41.9 ± 3.8 | $1.38e^{10}$ | $1.16e^{14}$ | $2.19e^{10}$ | $1.83e^{14}$ |
| | FedAU | 4.8 ± 1.5 | 51.2 ± 1.2 | 5.3 ± 1.8 | 60.8 ± 1.4 | 5.7 ± 2.0 | 38.4 ± 2.2 | $8.21e^{07}$ | $1.84e^{12}$ | $1.29e^{10}$ | $1.08e^{14}$ |
| | NoT | 9.5 ± 0.0* | 51.3 ± 0.2 | 9.6 ± 0.1 | 57.8 ± 1.0 | 10.4 ± 1.3 | 43.3 ± 4.4 | 0 | 0 | $1.03e^{10}$ | $8.64e^{13}$ |
| | FedOSD† | 49.5 ± 0.9 | 51.6 ± 0.7 | 60.1 ± 1.5 | 63.1 ± 0.9 | 25.8 ± 1.3 | 37.3 ± 2.0 | $8.08e^{09}$ | $6.75e^{13}$ | $9.69e^{09}$ | $8.10e^{13}$ |
| | FedQUIT | 45.2 ± 1.0 | 52.0 ± 0.5 | 55.8 ± 1.2 | 63.2 ± 1.3 | 8.0 ± 2.5 | 34.6 ± 2.4 | $8.98e^{07}$ | $7.50e^{11}$ | $3.08e^{09}$ | $2.57e^{13}$ |
| **CIFAR-100**, Non-IID, ResNet-18, $E=10$ | Retrain | 48.0 ± 0.8 | | 63.6 ± 0.7 | | 31.6 ± 4.5 | | | – | $1.62e^{11}$ | $1.35e^{16}$ |
| | PGA | 1.0 ± 0.2 | 48.3 ± 1.0 | 1.0 ± 0.2 | 61.9 ± 0.4 | 1.0 ± 0.2 | 34.7 ± 3.9 | $8.98e^{07}$ | $3.75e^{12}$ | $3.64e^{09}$ | $3.01e^{14}$ |
| | MoDe† | 16.1 ± 2.9 | 48.1 ± 0.9 | 16.7 ± 2.8 | 60.7 ± 1.2 | 3.8 ± 1.1 | 35.3 ± 3.6 | $1.38e^{10}$ | $1.16e^{15}$ | $2.10e^{10}$ | $1.76e^{15}$ |
| | FedAU | 5.1 ± 1.3 | 48.3 ± 0.7 | 5.9 ± 0.9 | 58.7 ± 0.3 | 6.2 ± 2.3 | 36.8 ± 3.7 | $8.21e^{07}$ | $1.84e^{13}$ | $1.13e^{10}$ | $9.45e^{14}$ |
| | NoT | 10.5 ± 0.0* | 46.7 ± 2.2 | 8.9 ± 0.3 | 56.1 ± 0.2 | 12.5 ± 0.9 | 39.7 ± 3.6 | 0 | 0 | $6.22e^{09}$ | $5.20e^{14}$ |
| | FedOSD† | 46.5 ± 1.5 | 48.2 ± 0.9 | 59.8 ± 0.7 | 62.9 ± 0.9 | 20.0 ± 2.1 | 34.8 ± 3.0 | $8.08e^{09}$ | $6.75e^{14}$ | $9.69e^{09}$ | $8.10e^{14}$ |
| | FedQUIT | 43.5 ± 1.6 | 48.3 ± 1.0 | 54.9 ± 0.2 | 63.3 ± 0.5 | 6.1 ± 2.2 | 32.7 ± 3.3 | $8.98e^{07}$ | $7.50e^{11}$ | $1.46e^{09}$ | $1.16e^{14}$ |
| **CIFAR-10**, Non-IID, ResNet-18, $E=1$ | Retrain | 83.5 ± 1.6 | | 89.0 ± 0.6 | | 81.1 ± 8.0 | | | – | $1.62e^{11}$ | $1.35e^{15}$ |
| | PGA | 10.1 ± 0.2 | 84.0 ± 1.0 | 10.0 ± 0.2 | 88.7 ± 0.6 | 0.5 ± 0.2 | 84.3 ± 4.9 | $8.98e^{07}$ | $3.75e^{12}$ | $9.38e^{09}$ | $8.14e^{13}$ |
| | MoDe† | 44.3 ± 4.2 | 83.5 ± 1.6 | 59.5 ± 4.6 | 87.9 ± 0.9 | 18.5 ± 4.2 | 82.3 ± 1.7 | $1.38e^{10}$ | $1.16e^{14}$ | $2.40e^{10}$ | $2.01e^{14}$ |
| | FedAU | 30.4 ± 8.2 | 83.8 ± 1.4 | 34.5 ± 1.8 | 89.4 ± 0.7 | 32.4 ± 7.8 | 85.2 ± 2.6 | $8.21e^{07}$ | $1.84e^{11}$ | $5.17e^{09}$ | $4.32e^{13}$ |
| | NoT | 42.4 ± 0.0* | 83.8 ± 1.2 | 42.1 ± 0.9 | 83.5 ± 0.6 | 45.0 ± 6.7 | 84.3 ± 3.0 | 0 | 0 | $1.37e^{10}$ | $1.14e^{14}$ |
| | FedOSD† | 78.5 ± 1.5 | 83.7 ± 1.3 | 84.7 ± 0.6 | 88.6 ± 0.8 | 58.9 ± 2.4 | 83.6 ± 2.2 | $8.08e^{09}$ | $6.75e^{13}$ | $1.05e^{10}$ | $8.78e^{13}$ |
| | FedQUIT | 58.9 ± 3.0 | 83.7 ± 1.2 | 60.1 ± 5.9 | 88.8 ± 0.7 | 13.7 ± 8.0 | 81.7 ± 1.7 | $8.98e^{07}$ | $7.50e^{11}$ | $3.16e^{09}$ | $2.64e^{13}$ |
| **CIFAR-100**, Non-IID, MiT-B0, $E=1$ | Retrain | 73.3 ± 0.8 | | 86.2 ± 0.6 | | 57.8 ± 5.3 | | | – | $1.19e^{10}$ | $3.82e^{15}$ |
| | PGA | 10.1 ± 0.2 | 73.6 ± 0.4 | 12.3 ± 0.3 | 86.1 ± 0.2 | 0.6 ± 0.2 | 62.2 ± 3.3 | $2.66e^{07}$ | $4.25e^{13}$ | $1.68e^{09}$ | $5.70e^{14}$ |
| | MoDe† | 66.6 ± 2.1 | 73.4 ± 0.8 | 81.3 ± 0.5 | 84.7 ± 0.9 | 55.3 ± 2.4 | 62.9 ± 2.2 | $4.09e^{09}$ | $1.31e^{15}$ | $5.55e^{09}$ | $1.78e^{15}$ |
| | FedAU | 1.8 ± 1.1 | 73.6 ± 0.7 | 1.4 ± 1.5 | 86.0 ± 0.4 | 2.5 ± 1.2 | 63.7 ± 1.9 | $1.03e^{07}$ | $4.61e^{10}$ | $3.54e^{09}$ | $1.13e^{15}$ |
| | NoT | 41.3 ± 0.0* | 73.4 ± 0.7 | 45.9 ± 0.9 | 86.6 ± 0.8 | 44.0 ± 3.7 | 67.9 ± 4.1 | 0 | 0 | $3.70e^{09}$ | $1.19e^{15}$ |
| | FedOSD† | 58.8 ± 2.1 | 73.5 ± 0.7 | 74.2 ± 1.3 | 86.4 ± 0.6 | 3.5 ± 1.2 | 59.8 ± 1.8 | $2.39e^{09}$ | $7.65e^{14}$ | $3.11e^{09}$ | $9.94e^{14}$ |
| | FedQUIT | 50.4 ± 1.6 | 73.4 ± 0.9 | 60.1 ± 1.5 | 86.3 ± 0.5 | 14.6 ± 4.5 | 58.5 ± 1.5 | $2.66e^{07}$ | $8.50e^{12}$ | $2.13e^{09}$ | $6.82e^{14}$ |
| **CUB-200**, Non-IID, MiT-B0, $E=1$ | Retrain | 56.5 ± 1.0 | | 82.5 ± 0.7 | | 34.4 ± 6.2 | | | – | $1.19e^{10}$ | $3.82e^{15}$ |
| | PGA | 5.4 ± 0.2 | 57.1 ± 5.4 | 5.1 ± 0.4 | 82.1 ± 0.3 | 0.5 ± 0.1 | 39.8 ± 3.3 | $2.66e^{07}$ | $4.25e^{13}$ | $2.87e^{09}$ | $9.53e^{14}$ |
| | MoDe† | 43.4 ± 2.2 | 57.0 ± 1.2 | 68.2 ± 0.7 | 81.8 ± 0.4 | 24.5 ± 0.8 | 46.6 ± 12.2 | $4.09e^{09}$ | $1.31e^{15}$ | $6.67e^{09}$ | $2.14e^{15}$ |
| | FedAU | 0.8 ± 0.2 | 56.9 ± 1.2 | 1.1 ± 0.4 | 82.3 ± 0.5 | 1.9 ± 0.5 | 37.8 ± 2.8 | $1.03e^{07}$ | $4.61e^{10}$ | $6.29e^{09}$ | $2.01e^{15}$ |
| | NoT | 25.3 ± 0.0* | 57.0 ± 1.2 | 39.6 ± 0.7 | 81.2 ± 0.6 | 41.4 ± 6.1 | 54.7 ± 5.1 | 0 | 0 | $2.39e^{09}$ | $7.65e^{14}$ |
| | FedOSD† | 52.7 ± 1.2 | 57.0 ± 1.2 | 77.3 ± 0.8 | 80.1 ± 0.4 | 37.2 ± 5.2 | 39.5 ± 2.7 | $2.39e^{09}$ | $7.65e^{14}$ | $3.82e^{09}$ | $1.22e^{15}$ |
| | FedQUIT | 47.8 ± 1.9 | 57.0 ± 0.9 | 70.1 ± 0.6 | 82.3 ± 0.1 | 10.5 ± 1.3 | 38.0 ± 2.7 | $2.66e^{07}$ | $8.50e^{12}$ | $2.20e^{09}$ | $7.05e^{14}$ |
| **CIFAR-100**, Non-IID, ResNet-18, $E=1$, 100 clients | Retrain | 35.1 ± 0.6 | | 36.8 ± 0.9 | | 34.2 ± 5.2 | | | – | $5.39e^{11}$ | $4.50e^{15}$ |
| | MoDe† | 14.2 ± 3.1 | 36.2 ± 0.7 | 15.1 ± 3.2 | 38.0 ± 1.0 | 4.1 ± 2.3 | 38.9 ± 4.0 | $1.38e^{10}$ | $1.16e^{14}$ | $2.36e^{10}$ | $2.14e^{14}$ |
| | FedAU | 3.5 ± 1.1 | 36.0 ± 0.8 | 4.6 ± 1.4 | 34.7 ± 1.0 | 0.0 ± 0.1 | 39.9 ± 4.5 | $8.21e^{07}$ | $1.84e^{12}$ | $3.01e^{10}$ | $3.02e^{14}$ |
| | NoT | 1.7 ± 0.0* | 36.3 ± 0.6 | 1.7 ± 0.1 | 34.8 ± 0.9 | 2.9 ± 0.4 | 40.0 ± 5.0 | 0 | 0 | $3.75e^{10}$ | $3.75e^{14}$ |
| | FedOSD† | 28.8 ± 1.0 | 36.5 ± 0.8 | 30.5 ± 1.3 | 38.0 ± 1.0 | 22.1 ± 1.8 | 39.9 ± 3.5 | $8.08e^{09}$ | $6.75e^{13}$ | $1.11e^{10}$ | $9.75e^{13}$ |
| | FedQUIT | 24.4 ± 1.2 | 36.5 ± 0.7 | 24.8 ± 1.1 | 37.9 ± 1.0 | 22.6 ± 2.0 | 36.1 ± 3.8 | $8.98e^{07}$ | $7.50e^{11}$ | $6.09e^{09}$ | $6.08e^{13}$ |
| **CUB-200**, Non-IID, MiT-B0, $E=1$, 100 clients | Retrain | 43.8 ± 1.0 | | 56.7 ± 0.9 | | 41.3 ± 6.3 | | | – | $5.32e^{10}$ | $1.70e^{16}$ |
| | MoDe† | 33.4 ± 2.5 | 44.4 ± 1.1 | 42.1 ± 0.8 | 57.0 ± 0.9 | 18.6 ± 1.9 | 43.6 ± 5.0 | $4.09e^{09}$ | $1.31e^{15}$ | $7.55e^{09}$ | $2.42e^{15}$ |
| | FedAU | 5.5 ± 1.0 | 44.2 ± 1.0 | 5.6 ± 0.8 | 56.8 ± 0.9 | 6.0 ± 0.9 | 43.6 ± 4.7 | $1.03e^{08}$ | $4.61e^{11}$ | $8.08e^{09}$ | $2.55e^{15}$ |
| | NoT | 22.9 ± 0.0* | 44.5 ± 1.2 | 28.7 ± 0.2 | 56.5 ± 1.0 | 49.3 ± 5.1 | 64.0 ± 6.0 | 0 | 0 | $4.26e^{09}$ | $1.36e^{15}$ |
| | FedOSD† | 40.2 ± 1.1 | 44.5 ± 1.0 | 55.0 ± 1.0 | 56.7 ± 0.8 | 29.9 ± 4.8 | 43.0 ± 4.9 | $2.39e^{09}$ | $7.65e^{14}$ | $3.99e^{09}$ | $1.28e^{15}$ |
| | FedQUIT | 35.2 ± 1.4 | 44.5 ± 1.1 | 46.9 ± 0.9 | 56.9 ± 0.9 | 14.7 ± 1.5 | 39.7 ± 4.2 | $2.66e^{07}$ | $8.50e^{12}$ | $8.25e^{08}$ | $2.64e^{14}$ |

Table 9: Test, Retain and Forget Accuracy after unlearning (before recovery starts) and after recovery ends. MIAs are omitted for the sake of space. Reported standard deviations are computed over 10 experiments, each using a different client as the unlearning client. Cost columns show communication (bytes) and computation (FLOPs) respectively needed only for the unlearning phase and for both the unlearning and recovery phase. For *Retrain* we report the costs of retraining from scratch. Methods marked with † use a multi-round unlearning phase where the unlearning client is actively involved. *By design, NoT yields an identical unlearned model for any choice of unlearning client; therefore the standard deviation across clients is 0.

## D.4 EXTENDED RESULTS OF TABLE 1

Table 9 and Table 10 provide additional post-unlearning and post-recovery results complementing those reported in Table 1.

## D.5 RESULTS WITH FIXED COMMUNICATION BUDGET

In this subsection, we evaluate all methods under several fixed cumulative communication budgets, as reported in Table 11 (the trends for computation are very similar, as shown in Table 1). We consider five budgets corresponding to 5%, 10%, 20%, 25% and 50% of the cumulative communication cost required to retrain the model from scratch. We report results for CIFAR-100 with ResNet-18 and 10 clients, under both IID and non-IID settings. FedQUIT consumes only 5% (IID) and 1.9%

| Setting | Method | Test Acc. | Retain Acc. | Forget Acc. | Comm. (Bytes) | Comp. (FLOPs) |
|---|---|---|---|---|---|---|
| **CIFAR-100**, IID, ResNet-18, $E$=1 | Retrain | $58.3 \pm 0.5$ | $82.7 \pm 0.8$ | $58.0 \pm 0.7$ | – | – |
| | GA | $1.9 \pm 0.1$ | $1.9 \pm 0.2$ | $2.0 \pm 0.5$ | $8.98e^{07}$ | $7.50e^{11}$ |
| | PGA | $1.0 \pm 0.2$ | $1.0 \pm 0.2$ | $1.0 \pm 0.2$ | $8.98e^{07}$ | $3.75e^{12}$ |
| | MoDe$^\dagger$ | $36.8 \pm 5.6$ | $40.2 \pm 0.9$ | $41.5 \pm 6.5$ | $1.38e^{10}$ | $1.16e^{14}$ |
| | FedAU | $4.2 \pm 1.3$ | $5.1 \pm 1.2$ | $5.3 \pm 1.2$ | $8.21e^{07}$ | $1.84e^{12}$ |
| | NoT | $12.2 \pm 0.0^*$ | $10.3 \pm 0.2$ | $13.4 \pm 1.8$ | $0$ | $0$ |
| | FedOSD$^\dagger$ | $4.5 \pm 1.6$ | $4.0 \pm 0.8$ | $3.8 \pm 2.0$ | $8.08e^{09}$ | $6.75e^{13}$ |
| | FedQUIT | $25.5 \pm 1.1$ | $27.2 \pm 1.3$ | $23.5 \pm 1.6$ | $8.98e^{07}$ | $7.50e^{11}$ |
| **CIFAR-100**, Non-IID, ResNet-18, $E$=1 | Retrain | $51.0 \pm 1.4$ | $64.6 \pm 1.2$ | $33.5 \pm 4.5$ | – | – |
| | GA | $1.0 \pm 0.1$ | $1.0 \pm 0.1$ | $1.0 \pm 0.1$ | $8.98e^{07}$ | $7.50e^{11}$ |
| | PGA | $1.0 \pm 0.2$ | $1.0 \pm 0.2$ | $1.0 \pm 0.2$ | $8.98e^{07}$ | $3.75e^{12}$ |
| | MoDe$^\dagger$ | $17.5 \pm 3.6$ | $15.8 \pm 3.3$ | $4.5 \pm 2.7$ | $1.38e^{10}$ | $1.16e^{14}$ |
| | FedAU | $4.8 \pm 1.5$ | $5.3 \pm 1.8$ | $5.7 \pm 2.0$ | $8.21e^{07}$ | $1.84e^{12}$ |
| | NoT | $9.5 \pm 0.0^*$ | $9.6 \pm 0.1$ | $10.4 \pm 1.3$ | $0$ | $0$ |
| | FedOSD$^\dagger$ | $49.5 \pm 0.9$ | $60.1 \pm 1.5$ | $25.8 \pm 1.3$ | $8.08e^{09}$ | $6.75e^{13}$ |
| | FedQUIT | $45.2 \pm 1.0$ | $55.8 \pm 1.2$ | $8.0 \pm 2.5$ | $8.98e^{07}$ | $7.50e^{11}$ |
| **CIFAR-100**, Non-IID, ResNet-18, $E$=10 | Retrain | $48.0 \pm 0.8$ | $63.6 \pm 0.7$ | $31.6 \pm 4.5$ | – | – |
| | PGA | $1.0 \pm 0.2$ | $1.0 \pm 0.2$ | $1.0 \pm 0.2$ | $8.98e^{07}$ | $3.75e^{12}$ |
| | MoDe$^\dagger$ | $16.1 \pm 2.9$ | $16.7 \pm 2.8$ | $3.8 \pm 1.1$ | $1.38e^{10}$ | $1.16e^{15}$ |
| | FedAU | $5.1 \pm 1.3$ | $5.9 \pm 0.9$ | $6.2 \pm 2.3$ | $8.21e^{07}$ | $1.84e^{13}$ |
| | NoT | $10.5 \pm 0.0^*$ | $8.9 \pm 0.3$ | $12.5 \pm 0.9$ | $0$ | $0$ |
| | FedOSD$^\dagger$ | $46.5 \pm 1.5$ | $59.8 \pm 0.7$ | $20.0 \pm 2.1$ | $8.08e^{09}$ | $6.75e^{14}$ |
| | FedQUIT | $43.5 \pm 1.6$ | $54.9 \pm 0.2$ | $6.1 \pm 2.2$ | $8.98e^{07}$ | $7.50e^{11}$ |
| **CIFAR-10**, Non-IID, ResNet-18, $E$=1 | Retrain | $83.5 \pm 1.6$ | $89.0 \pm 0.6$ | $81.1 \pm 8.0$ | – | – |
| | PGA | $10.1 \pm 0.2$ | $10.0 \pm 0.2$ | $0.5 \pm 0.2$ | $8.98e^{07}$ | $3.75e^{12}$ |
| | MoDe$^\dagger$ | $44.3 \pm 4.2$ | $59.5 \pm 4.6$ | $18.5 \pm 4.2$ | $1.38e^{10}$ | $1.16e^{14}$ |
| | FedAU | $30.4 \pm 8.2$ | $34.5 \pm 1.8$ | $32.4 \pm 7.8$ | $8.21e^{07}$ | $1.84e^{11}$ |
| | NoT | $42.4 \pm 0.0^*$ | $42.1 \pm 0.9$ | $45.0 \pm 6.7$ | $0$ | $0$ |
| | FedOSD$^\dagger$ | $78.5 \pm 1.5$ | $84.7 \pm 0.6$ | $58.9 \pm 2.4$ | $8.08e^{09}$ | $6.75e^{13}$ |
| | FedQUIT | $58.9 \pm 3.0$ | $60.1 \pm 5.9$ | $13.7 \pm 8.0$ | $8.98e^{07}$ | $7.50e^{11}$ |
| **CIFAR-100**, Non-IID, MiT-B0, $E$=1 | Retrain | $73.3 \pm 0.8$ | $86.2 \pm 0.6$ | $57.8 \pm 5.3$ | – | – |
| | PGA | $10.1 \pm 0.2$ | $12.3 \pm 0.3$ | $0.6 \pm 0.2$ | $2.66e^{07}$ | $4.25e^{13}$ |
| | MoDe$^\dagger$ | $66.6 \pm 2.1$ | $81.3 \pm 0.5$ | $55.3 \pm 2.4$ | $4.09e^{09}$ | $1.31e^{15}$ |
| | FedAU | $1.8 \pm 1.1$ | $1.4 \pm 1.5$ | $2.5 \pm 1.2$ | $1.03e^{07}$ | $4.61e^{10}$ |
| | NoT | $41.3 \pm 0.0^*$ | $45.9 \pm 0.9$ | $44.0 \pm 3.7$ | $0$ | $0$ |
| | FedOSD$^\dagger$ | $58.8 \pm 2.1$ | $74.2 \pm 1.3$ | $3.5 \pm 1.2$ | $2.39e^{09}$ | $7.65e^{14}$ |
| | FedQUIT | $50.4 \pm 1.6$ | $60.1 \pm 1.5$ | $14.6 \pm 4.5$ | $2.66e^{07}$ | $8.50e^{12}$ |
| **CUB-200**, Non-IID, MiT-B0, $E$=1 | Retrain | $56.5 \pm 1.0$ | $82.5 \pm 0.7$ | $34.4 \pm 6.2$ | – | – |
| | PGA | $5.4 \pm 0.2$ | $5.1 \pm 0.4$ | $0.5 \pm 0.1$ | $2.66e^{07}$ | $4.25e^{13}$ |
| | MoDe$^\dagger$ | $43.4 \pm 2.2$ | $68.2 \pm 0.7$ | $24.5 \pm 0.8$ | $4.09e^{09}$ | $1.31e^{15}$ |
| | FedAU | $0.8 \pm 0.2$ | $1.1 \pm 0.4$ | $1.9 \pm 0.5$ | $1.03e^{07}$ | $4.61e^{10}$ |
| | NoT | $25.3 \pm 0.0^*$ | $39.6 \pm 0.7$ | $41.4 \pm 6.1$ | $0$ | $0$ |
| | FedOSD$^\dagger$ | $52.7 \pm 1.2$ | $77.3 \pm 0.8$ | $37.2 \pm 5.2$ | $2.39e^{09}$ | $7.65e^{14}$ |
| | FedQUIT | $47.8 \pm 1.9$ | $70.1 \pm 0.6$ | $10.5 \pm 1.3$ | $2.66e^{07}$ | $8.50e^{12}$ |
| **CIFAR-100**, Non-IID, ResNet-18, $E$=1, 100 clients | Retrain | $35.1 \pm 0.6$ | $36.8 \pm 0.9$ | $34.2 \pm 5.2$ | $5.39e^{11}$ | $4.50e^{14}$ |
| | MoDe$^\dagger$ | $14.2 \pm 3.1$ | $15.1 \pm 3.2$ | $4.1 \pm 2.3$ | $9.51e^{10}$ | $7.95e^{13}$ |
| | FedAU | $3.5 \pm 1.1$ | $4.6 \pm 1.4$ | $0.0 \pm 0.1$ | $8.21e^{07}$ | $2.05e^{12}$ |
| | NoT | $1.7 \pm 0.0^*$ | $1.7 \pm 0.1$ | $2.9 \pm 0.4$ | $0$ | $0$ |
| | FedOSD$^\dagger$ | $28.8 \pm 1.0$ | $30.5 \pm 1.3$ | $22.1 \pm 1.8$ | $8.98e^{09}$ | $7.50e^{12}$ |
| | FedQUIT | $24.4 \pm 1.2$ | $24.8 \pm 1.1$ | $22.6 \pm 2.0$ | $8.98e^{07}$ | $7.50e^{10}$ |
| **CUB-200**, Non-IID, MiT-B0, $E$=1, 100 clients | Retrain | $43.8 \pm 1.0$ | $56.7 \pm 0.9$ | $41.3 \pm 6.3$ | $5.31e^{10}$ | $2.04e^{14}$ |
| | MoDe$^\dagger$ | $33.4 \pm 2.5$ | $42.1 \pm 0.8$ | $18.6 \pm 1.9$ | $2.81e^{10}$ | $1.08e^{14}$ |
| | FedAU | $5.5 \pm 1.0$ | $5.6 \pm 0.8$ | $6.0 \pm 0.9$ | $1.03e^{07}$ | $6.14e^{10}$ |
| | NoT | $22.9 \pm 0.0^*$ | $28.7 \pm 0.2$ | $49.3 \pm 5.1$ | $0$ | $0$ |
| | FedOSD$^\dagger$ | $40.2 \pm 1.1$ | $55.0 \pm 1.0$ | $29.9 \pm 4.8$ | $2.66e^{09}$ | $1.02e^{13}$ |
| | FedQUIT | $35.2 \pm 1.4$ | $46.9 \pm 0.9$ | $14.7 \pm 1.5$ | $2.66e^{07}$ | $1.02e^{11}$ |

Table 10: Compact view with only the *After Unlearning* test/retain/forget accuracies and the *Unlearning Cost* (communication in bytes, computation in FLOPs), only unlearning phase. $^\dagger$: multi-round unlearning with active target client; $^*$: identical unlearned model for any target client (std = 0).

(non-IID) of the retraining budget (see Table 1); therefore, budget points at 10% and above occur well after FedQUIT has already fully recovered its utility. This analysis aims to determine whether forgetting remains effective later in the recovery phase, ruling out the possibility that, with budgets larger than strictly necessary to regain retraining-level utility, the forgetting process might reverse.

| Budget | Method | IID | | | Non-IID | | |
|---|---|---|---|---|---|---|---|
| | | Test Acc. ($\triangle$) | Retain Acc. ($\triangle$) | Forget Acc. ($\triangle$) | Test Acc. ($\triangle$) | Retain Acc. ($\triangle$) | Forget Acc. ($\triangle$) |
| 100% | Retrain | 58.31 | 82.72 | 58.02 | 51.02 | 64.61 | 33.52 |
| 5% | FedEraser | 13.28 (-45.03) | 16.45 (-66.27) | 12.68 (-45.34) | 9.84 (-41.18) | 11.00 (-53.61) | 1.84 (-31.68) |
| | PGA | 13.16 (-45.15) | 18.35 (-64.37) | 13.39 (-44.63) | 51.92 (+0.90) | 59.87 (-4.74) | 37.73 (+4.21) |
| | MoDe | 36.61 (-21.70) | 42.27 (-40.45) | 42.32 (-15.70) | 24.99 (-26.03) | 27.82 (-36.79) | 21.60 (-11.92) |
| | FedAU | 60.39 (+2.08) | 80.27 (-2.45) | 73.44 (+15.42) | 49.68 (-1.34) | 59.71 (-4.90) | 36.12 (+2.60) |
| | NoT | 60.06 (+1.75) | 77.96 (-4.76) | 71.54 (+13.52) | 51.20 (+0.18) | 60.12 (-4.49) | 42.78 (+9.26) |
| | FedOSD | 4.48 (-53.83) | 8.33 (-74.39) | 4.32 (-53.70) | 49.47 (-1.55) | 61.51 (-3.10) | 26.00 (-7.52) |
| | FedQUIT | 58.05 (-0.26) | 74.58 (-8.14) | 58.96 (+0.94) | 53.06 (+2.04) | 65.00 (+0.39) | 36.34 (+2.82) |
| 10% | FedEraser | 20.21 (-38.10) | 25.41 (-57.31) | 19.90 (-38.12) | 16.79 (-34.23) | 20.34 (-44.27) | 5.53 (-27.99) |
| | PGA | 59.04 (+0.73) | 76.53 (-6.19) | 72.61 (+14.59) | 52.75 (+1.73) | 65.98 (+1.37) | 38.88 (+5.36) |
| | MoDe | 58.91 (+0.60) | 74.64 (-8.08) | 69.12 (+11.10) | 49.53 (-1.49) | 57.56 (-7.05) | 36.96 (+3.44) |
| | FedAU | 60.17 (+1.86) | 81.37 (-1.35) | 71.50 (+13.48) | 51.72 (+0.70) | 63.46 (-1.15) | 39.48 (+5.96) |
| | NoT | 59.37 (+1.06) | 79.38 (-3.34) | 69.22 (+11.20) | 51.86 (+0.84) | 63.19 (-1.42) | 41.28 (+7.76) |
| | FedOSD | 59.52 (+1.21) | 79.32 (-3.40) | 71.83 (+13.81) | 53.18 (+2.16) | 65.32 (+0.71) | 38.77 (+5.25) |
| | FedQUIT | 59.90 (+1.59) | 81.42 (-1.30) | 60.86 (+2.84) | 54.55 (+2.53) | 67.09 (+2.48) | 36.38 (+2.86) |
| 20% | FedEraser | 29.16 (-29.15) | 34.28 (-48.44) | 29.26 (-28.76) | 23.58 (-27.44) | 26.45 (-38.16) | 10.23 (-23.29) |
| | PGA | 59.90 (+1.59) | 80.53 (-2.19) | 71.13 (+13.11) | 53.13 (+2.11) | 68.31 (+3.70) | 38.50 (+4.98) |
| | MoDe | 59.82 (+1.51) | 82.49 (-0.23) | 67.82 (+9.80) | 52.79 (+1.77) | 64.59 (-0.02) | 39.22 (+5.70) |
| | FedAU | 60.69 (+2.38) | 81.91 (-0.81) | 69.08 (+11.06) | 51.08 (-0.06) | 63.97 (-0.64) | 37.17 (+3.65) |
| | NoT | 60.63 (+2.32) | 82.39 (-0.33) | 68.12 (+10.10) | 53.42 (+2.40) | 67.68 (+3.07) | 41.51 (+7.99) |
| | FedOSD | 59.98 (+1.67) | 82.10 (-0.62) | 71.05 (+13.03) | 53.48 (+2.46) | 66.48 (+1.87) | 37.50 (+3.98) |
| | FedQUIT | 60.12 (+1.81) | 82.20 (-0.52) | 61.14 (+3.12) | 53.93 (+2.91) | 69.25 (+4.64) | 35.90 (+2.38) |
| 25% | FedEraser | 31.96 (-26.35) | 39.76 (-42.96) | 31.46 (-26.56) | 25.52 (-25.50) | 31.78 (-32.83) | 11.17 (-22.35) |
| | PGA | 60.35 (+2.04) | 83.43 (+0.71) | 69.78 (+11.76) | 54.02 (+3.00) | 69.51 (+4.90) | 40.16 (+6.64) |
| | MoDe | 60.67 (+2.36) | 79.45 (-3.27) | 70.00 (+11.98) | 53.74 (+2.72) | 66.86 (+2.25) | 39.70 (+6.18) |
| | FedAU | 61.09 (+2.78) | 85.96 (+3.26) | 68.46 (+10.44) | 53.25 (+2.23) | 68.84 (+4.23) | 40.47 (+6.95) |
| | NoT | 61.85 (+3.54) | 86.04 (+3.32) | 69.88 (+11.86) | 54.23 (+3.21) | 68.63 (+4.02) | 41.61 (+8.09) |
| | FedOSD | 61.12 (+2.81) | 84.70 (+1.98) | 70.88 (+12.86) | 54.40 (+3.38) | 69.58 (+4.97) | 39.30 (+5.78) |
| | FedQUIT | 61.21 (+2.90) | 86.13 (+3.41) | 62.36 (+4.34) | 55.29 (+4.27) | 75.29 (+10.68) | 37.24 (+3.72) |
| 50% | FedEraser | 45.48 (-12.83) | 56.28 (-26.44) | 45.42 (-12.60) | 37.76 (-13.26) | 41.32 (-23.29) | 19.99 (-13.53) |
| | PGA | 61.09 (+2.78) | 89.95 (+7.23) | 68.83 (+10.81) | 55.20 (+4.18) | 73.31 (+8.70) | 40.49 (+6.97) |
| | MoDe | 61.44 (+3.13) | 91.98 (+9.26) | 68.50 (+10.48) | 54.95 (+3.93) | 72.45 (+7.84) | 41.18 (+7.66) |
| | FedAU | 61.57 (+3.26) | 91.03 (+8.31) | 66.66 (+8.64) | 55.38 (+4.36) | 74.42 (+9.81) | 41.88 (+8.36) |
| | NoT | 61.05 (+2.74) | 90.59 (+7.87) | 66.36 (+8.34) | 55.10 (+4.08) | 73.27 (+8.66) | 40.39 (+6.87) |
| | FedOSD | 61.25 (+2.94) | 90.32 (+7.60) | 66.88 (+8.86) | 55.30 (+4.28) | 74.14 (+9.53) | 39.76 (+6.24) |
| | FedQUIT | 61.67 (+3.36) | 91.52 (+8.80) | 62.58 (+4.56) | 55.48 (+4.46) | 74.47 (+9.86) | 37.51 (+3.99) |

Table 11: Performance comparison under different fixed communication budgets. We evaluate five budget levels expressed as a percentage of the communication cost required to retrain the model from scratch ($100\% = 1.62e^{11}$ bytes): 5% ($8.10e^9$ bytes), 10% ($1.62e^{10}$ bytes), 20% ($3.24e^{10}$ bytes), 25% ($4.05e^{10}$ bytes), and 50% ($8.10e^{10}$ bytes). For each metric, values in parentheses indicate the gap with respect to the *Retrain* baseline, with negative gaps shown in red. Note that, as shown in Table 1 under the efficiency metrics, the 10% and 20% budget points occur significantly after most methods have already fully recovered their utility.

As shown in Table 11, FedQUIT consistently outperforms the other baselines, achieving both lower deltas in Forget Acc. and, often, higher Test and Retain Acc. values under the same communication budget. This advantage stems from FedQUIT's ability to rapidly recover model utility while remaining highly selective in forgetting.

## D.6 EXPERIMENTS WITH SAMPLE UNLEARNING

We conduct a complementary set of experiments on sample unlearning. We evaluate three forget-set sizes: 50% and 10% of the local data drawn at random, and an extreme 1% drawn from the least-represented local classes. Experiments are performed under both Non-IID and IID scenarios.

We did not include PGA (Halimi et al., 2022) in the comparison because it needs to know beforehand which are the forget data to build the reference model (see baseline description in D.2). We adapted MoDe and FedOSD by treating the unlearning samples as belonging to a virtual client throughout their multi-round unlearning phases. Table 14 reports the results.

Table 12 and Table 13 respectively report post-recovery and post-unlearning performance for the extreme 10% and 1% forget sets. These tables also include the local retain accuracy (Local Retain Acc.), computed on the portion of each client's data that should not be forgotten, to assess the selec-

| Setting | Method | Efficacy | | | | Efficiency | |
| | | Test Acc. | Retain Acc. ($\Delta \downarrow$) | Local Retain Acc. ($\Delta \downarrow$) | Forget Acc. ($\Delta\downarrow$) | Communication Bytes | Computation FLOPs |
|---|---|---|---|---|---|---|---|
| CIFAR-100 IID 10% Local | Original | 59.81 | 79.71 | 80.76 | 80.40 | | |
| | Retrain | 58.60 (0.00) | 79.30 (0.00) | 77.98 (0.00) | 57.60 (0.00) | $1.80e^{11}$ | $1.36e^{15}$ |
| | MoDe | 59.43 (0.83) | 78.67 (0.63) | 78.69 (0.71) | 62.93 (5.33) | $6.55e^{9}$ | $5.18e^{13}$ |
| | NoT | 59.02 (0.42) | 77.03 (2.27) | 76.31 (1.67) | 67.04 (9.44) | $4.49e^{9}$ | $3.41e^{13}$ |
| | FedAU | 59.24 (0.64) | 78.88 (0.42) | 78.01 (0.03) | 61.58 (3.98) | $2.69e^{9}$ | $2.05e^{13}$ |
| | FedOSD | 59.53 (0.93) | 78.77 (0.53) | 78.79 (0.81) | 59.93 (2.33) | $6.01e^{9}$ | $4.78e^{13}$ |
| | FedQUIT | 59.63 (1.03) | 79.17 (0.13) | 78.63 (0.65) | 58.73 (1.13) | $9.87e^{8}$ | $7.58e^{12}$ |
| CIFAR-100 Non-IID 10% Local | Original | 53.81 | 64.66 | 64.00 | 63.79 | | |
| | Retrain | 51.33 (0.00) | 62.30 (0.00) | 62.60 (0.00) | 48.91 (0.00) | $1.80e^{11}$ | $1.36e^{15}$ |
| | MoDe | 51.44 (0.11) | 62.03 (0.27) | 62.01 (0.59) | 52.13 (3.22) | $4.76e^{9}$ | $3.82e^{13}$ |
| | NoT | 51.48 (0.15) | 60.42 (1.88) | 61.62 (0.98) | 55.12 (6.21) | $5.39e^{9}$ | $4.10e^{13}$ |
| | FedAU | 51.58 (0.25) | 61.91 (0.39) | 62.02 (0.58) | 50.82 (1.91) | $3.59e^{9}$ | $2.73e^{13}$ |
| | FedOSD | 51.54 (0.21) | 62.13 (0.17) | 62.11 (0.49) | 50.43 (1.52) | $5.12e^{9}$ | $4.10e^{13}$ |
| | FedQUIT | 51.64 (0.31) | 62.53 (0.23) | 62.82 (0.22) | 47.93 (0.98) | $1.88e^{9}$ | $1.44e^{13}$ |
| CIFAR-100 IID 1% Local | Original | 59.87 | 79.71 | 80.65 | 88.00 | | |
| | Retrain | 59.29 (0.00) | 81.24 (0.00) | 80.93 (0.00) | 70.00 (0.00) | $1.80e^{11}$ | $1.35e^{15}$ |
| | MoDe | 60.10 (0.81) | 79.90 (1.34) | 79.80 (1.13) | 76.00 (6.00) | $5.65e^{9}$ | $4.50e^{13}$ |
| | FedAU | 60.10 (0.81) | 81.30 (0.06) | 82.55 (1.62) | 86.00 (16.00) | $1.80e^{9}$ | $1.35e^{13}$ |
| | NoT | 59.40 (0.11) | 79.00 (2.24) | 76.44 (4.49) | 76.00 (6.00) | $7.18e^{9}$ | $5.41e^{13}$ |
| | FedOSD | 60.00 (0.71) | 81.30 (0.06) | 82.55 (1.62) | 73.00 (3.00) | $5.12e^{9}$ | $4.05e^{13}$ |
| | FedQUIT | 60.10 (0.81) | 81.30 (0.06) | 80.50 (0.43) | 68.00 (2.00) | $1.88e^{9}$ | $1.43e^{13}$ |
| CIFAR-100 Non-IID 1% Local | Original | 51.81 | 63.79 | 64.08 | 66.67 | | |
| | Retrain | 50.84 (0.00) | 64.82 (0.00) | 64.94 (0.00) | 55.00 (0.00) | $1.80e^{11}$ | $1.35e^{15}$ |
| | MoDe | 51.53 (0.69) | 63.20 (1.62) | 64.30 (0.64) | 58.70 (3.70) | $6.19e^{9}$ | $5.18e^{13}$ |
| | FedAU | 51.10 (0.26) | 63.60 (1.22) | 64.10 (0.84) | 59.00 (4.00) | $5.39e^{9}$ | $4.05e^{13}$ |
| | NoT | 51.50 (0.66) | 64.40 (0.42) | 64.14 (0.80) | 58.67 (3.67) | $7.18e^{9}$ | $5.41e^{13}$ |
| | FedOSD | 51.93 (1.09) | 63.95 (0.87) | 64.53 (0.41) | 57.40 (2.40) | $4.04e^{9}$ | $3.38e^{13}$ |
| | FedQUIT | 52.00 (1.16) | 64.53 (0.29) | 65.14 (0.20) | 56.67 (1.67) | $9.87e^{8}$ | $7.51e^{12}$ |

Table 12: Post-recovery performance for sample unlearning on different settings and different amount of forget data, indicated on the left column. *Local Retain Acc.* report the accuracy on the retain data locally at the target client.

tivity of each method with such a particular subset of data. Table 14 reports the results for the 50% forget-set case.

**Results for 1%-10% Forget Data.** In the extreme settings (1% and 10% forget data), FedQUIT consistently outperforms all baselines in both IID and Non-IID regimes, exhibiting lower degradation immediately after unlearning and more accurate forgetting after recovery, while often requiring substantially lower computation and communication costs.

**Results for 50% Forget Data.** Under IID data, FedQUIT achieves the lowest Avg. Gap (1.2), significantly outperforming FedAU (6.2) and NoT (12.5). Notably, this comes with efficiency savings of roughly $25\times$ in communication and $42\times$ in computation, representing a favorable trade-off compared to methods like NoT and FedAU, which exhibit larger efficacy gaps despite lower cost. Under Non-IID data, FedQUIT exhibits the best Avg. Gap (2.9), while also delivering significant computational efficiency ($\sim 100\times$ reduction over *Retrain*). Overall, FedQUIT offers a balanced compromise between unlearning efficacy and efficiency, particularly in heterogeneous-data regimes where both communication and computation costs are significantly reduced.

### D.7 ABLATION ON DISTILLATION LOSS

We investigate the impact of different loss functions used for distillation between the virtual teacher and the student model (used in Eq. 2). We evaluated the default KL divergence against mean squared error (MSE), categorical cross-entropy (CE) with soft targets, and cosine similarity (CS). Table 15 summarizes the results. KL and CE perform similarly, particularly in the IID setting, with KL achieving slightly lower delta values across forget accuracy and MIA metrics. In the Non-IID setting, CE tends to over-forget, as indicated by lower absolute accuracy but higher delta values. Overall, MSE requires a shorter recovery phase (fewer rounds) but consistently performs worse in terms of forgetting (higher delta values in all metrics). These results align with our findings in Section 5.3: the teacher signal that aids retention lies in the relative non-true structure (ordering and odds ratios). KL and CE operate on the probability simplex and penalize deviations in log-odds,

| Setting | Method | Efficacy | | | | Efficiency | |
| | | Test Acc. | Retain Acc. ($\Delta \downarrow$) | Local Retain Acc. ($\Delta \downarrow$) | Forget Acc. ($\Delta\downarrow$) | Communication Bytes | Computation FLOPs |
|---|---|---|---|---|---|---|---|
| CIFAR-100 IID 10% Local | Original | 59.81 | 79.71 | 80.76 | 80.40 | | |
| | Retrain | 58.60 (0.00) | 79.30 (0.00) | 77.98 (0.00) | 57.60 (0.00) | $1.80e^{11}$ | $1.36e^{15}$ |
| | NoT | 11.00 (47.60) | 10.80 (68.50) | 11.00 (66.98) | 10.44 (47.16) | 0 | 0 |
| | FedAU | 34.60 (24.00) | 46.40 (32.90) | 40.30 (37.68) | 22.10 (35.50) | $8.21e^7$ | $1.84e^{12}$ |
| | FedOSD | 14.32 (44.28) | 12.47 (66.83) | 10.03 (67.95) | 3.41 (54.19) | $2.42e^9$ | $2.05e^{13}$ |
| | MoDe | 35.53 (23.07) | 44.58 (34.72) | 40.42 (37.56) | 26.03 (31.57) | $2.96e^9$ | $2.45e^{13}$ |
| | FedQUIT | 47.90 (10.70) | 59.60 (19.70) | 60.31 (17.67) | 35.00 (22.60) | $8.98e^7$ | $7.50e^{11}$ |
| CIFAR-100 Non-IID 10% Local | Original | 53.81 | 64.66 | 64.00 | 63.79 | | |
| | Retrain | 51.33 (0.00) | 62.30 (0.00) | 62.60 (0.00) | 48.91 (0.00) | $1.80e^{11}$ | $1.36e^{15}$ |
| | NoT | 9.10 (42.23) | 8.30 (54.00) | 8.30 (54.30) | 8.50 (40.41) | 0 | 0 |
| | FedAU | 32.30 (19.03) | 43.00 (19.30) | 9.00 (53.60) | 4.50 (44.41) | $8.21e^7$ | $1.84e^{12}$ |
| | FedOSD | 48.27 (3.06) | 56.38 (5.92) | 32.28 (30.32) | 7.29 (41.62) | $2.42e^9$ | $2.05e^{13}$ |
| | MoDe | 33.53 (17.80) | 44.63 (17.67) | 9.37 (53.23) | 9.02 (39.89) | $2.96e^9$ | $2.45e^{13}$ |
| | FedQUIT | 44.50 (6.83) | 55.20 (7.10) | 21.00 (41.60) | 0.90 (48.01) | $8.98e^7$ | $7.50e^{11}$ |
| CIFAR-100 IID 1% Atypical | Original | 59.87 | 79.71 | 80.65 | 88.00 | | |
| | Retrain | 59.29 (0.00) | 81.24 (0.00) | 80.93 (0.00) | 70.00 (0.00) | $1.80e^{11}$ | $1.35e^{15}$ |
| | MoDe | 37.32 (21.97) | 59.73 (21.51) | 52.00 (28.93) | 5.00 (65.00) | $2.96e^9$ | $2.47e^{13}$ |
| | FedAU | 58.73 (0.56) | 77.38 (3.86) | 85.90 (4.97) | 0.00 (70.00) | $8.21e^7$ | $1.84e^{12}$ |
| | NoT | 10.00 (49.29) | 10.00 (71.24) | 11.00 (69.93) | 0.00 (70.00) | 0 | 0 |
| | FedOSD | 58.73 (0.56) | 77.38 (3.86) | 85.90 (4.97) | 0.00 (70.00) | $2.42e^9$ | $2.03e^{13}$ |
| | FedQUIT | 49.26 (10.03) | 61.96 (19.28) | 64.10 (16.83) | 0.00 (70.00) | $8.98e^7$ | $7.50e^{11}$ |
| CIFAR-100 Non-IID 1% Local | Original | 51.81 | 63.79 | 64.08 | 66.67 | | |
| | Retrain | 50.84 (0.00) | 64.82 (0.00) | 64.94 (0.00) | 55.00 (0.00) | $1.80e^{11}$ | $1.35e^{15}$ |
| | MoDe | 42.20 (8.64) | 36.30 (28.52) | 37.80 (27.14) | 8.51 (46.49) | $2.96e^9$ | $2.48e^{13}$ |
| | FedAU | 24.50 (26.34) | 23.50 (41.32) | 6.70 (58.24) | 0.00 (55.00) | $8.21e^7$ | $1.84e^{12}$ |
| | NoT | 9.70 (41.14) | 9.10 (55.72) | 16.00 (48.94) | 15.12 (39.88) | 0 | 0 |
| | FedOSD | 49.30 (1.54) | 60.51 (4.31) | 61.20 (3.74) | 60.00 (5.00) | $2.42e^9$ | $2.02e^{13}$ |
| | FedQUIT | 50.79 (0.05) | 61.20 (3.62) | 63.30 (1.64) | 41.13 (13.87) | $8.98e^7$ | $7.50e^{11}$ |

Table 13: Post-unlearning performance for sample unlearning on different settings and different amount of forget data, indicated on the left column. *Local Retain Acc.* report the accuracy on the retain data locally at the target client.

| Setting | Method | Efficacy | | | | | Efficiency | |
| | | Test Acc. | Forget Acc. ($\Delta \downarrow$) | $MIA_{[Song]}$ ($\Delta \downarrow$) | $MIA_{[Yeom]}$ ($\Delta \downarrow$) | Avg. Gap ($\downarrow$) | Communication Bytes ($\times \uparrow$) | Computation FLOPs ($\times \uparrow$) |
|---|---|---|---|---|---|---|---|---|
| **CIFAR-100**, IID, ResNet-18, $E=1$ | Retrain | $59.4_{\pm 0.4}$ | $58.5_{\pm 0.5}$ | $56.0_{\pm 0.6}$ | $49.2_{\pm 0.7}$ | – | $1.80e^{11}$ (1.0×) | $1.42e^{15}$ (1.0×) |
| | MoDe | $59.5_{\pm 0.3}$ | $63.5$ $(5.0_{\pm 0.6})$ | $61.0$ $(3.8_{\pm 0.5})$ | $53.8$ $(3.7_{\pm 0.6})$ | 4.2 | $1.92e^{10}$ (9.4×) | $1.53e^{14}$ (9.3×) |
| | FedAU | $59.8_{\pm 0.3}$ | $65.7$ $(7.2_{\pm 1.1})$ | $61.8$ $(5.8_{\pm 0.9})$ | $54.9$ $(5.7_{\pm 0.9})$ | 6.2 | $\mathbf{8.08e^{08}}$ **(222.7×)** | $\mathbf{7.12e^{12}}$ **(199.3×)** |
| | NoT | $59.7_{\pm 0.5}$ | $72.4$ $(13.9_{\pm 0.7})$ | $68.1$ $(12.1_{\pm 1.0})$ | $60.7$ $(11.5_{\pm 1.0})$ | 12.5 | $5.65e^{09}$ (31.8×) | $4.99e^{13}$ (28.5×) |
| | FedOSD | $59.6_{\pm 0.6}$ | $62.1$ $(3.6_{\pm 0.6})$ | $59.2$ $(3.2_{\pm 0.5})$ | $52.2$ $(3.0_{\pm 0.5})$ | 3.3 | $9.87e^{09}$ (18.2×) | $8.55e^{13}$ (16.6×) |
| | FedQUIT | $59.6_{\pm 0.4}$ | $\mathbf{60.0}$ $\mathbf{(1.5_{\pm 0.6})}$ | $\mathbf{57.2}$ $\mathbf{(1.2_{\pm 0.5})}$ | $\mathbf{50.1}$ $\mathbf{(0.9_{\pm 0.5})}$ | **1.2** | $7.36e^{09}$ (24.5×) | $3.41e^{13}$ (41.6×) |
| **CIFAR-100**, Non-IID, ResNet-18, $E=1$ | Retrain | $49.0_{\pm 0.6}$ | $40.3_{\pm 2.4}$ | $38.9_{\pm 2.0}$ | $34.1_{\pm 1.8}$ | – | $1.80e^{11}$ (1.0×) | $1.42e^{15}$ (1.0×) |
| | MoDe | $49.3_{\pm 0.3}$ | $47.3$ $(7.0_{\pm 1.0})$ | $42.9$ $(3.9_{\pm 1.0})$ | $37.9$ $(3.8_{\pm 1.0})$ | 4.0 | $1.92e^{10}$ (9.4×) | $1.53e^{14}$ (9.3×) |
| | FedAU | $50.0_{\pm 0.2}$ | $47.2$ $(6.9_{\pm 1.2})$ | $45.1$ $(6.2_{\pm 1.1})$ | $40.2$ $(6.1_{\pm 1.1})$ | 6.4 | $\mathbf{8.08e^{08}}$ **(222.7×)** | $\mathbf{7.12e^{12}}$ **(199.3×)** |
| | NoT | $49.6_{\pm 0.8}$ | $49.0$ $(8.8_{\pm 1.4})$ | $47.0$ $(8.1_{\pm 1.3})$ | $41.8$ $(7.7_{\pm 1.2})$ | 8.2 | $6.46e^{09}$ (27.9×) | $5.70e^{13}$ (24.9×) |
| | FedOSD | $49.6_{\pm 0.5}$ | $45.9$ $(5.6_{\pm 1.0})$ | $43.7$ $(4.0_{\pm 1.0})$ | $38.6$ $(4.4_{\pm 1.0})$ | 3.2 | $9.87e^{09}$ (18.2×) | $8.55e^{13}$ (16.6×) |
| | FedQUIT | $49.7_{\pm 0.5}$ | $\mathbf{43.8}$ $\mathbf{(3.5_{\pm 1.0})}$ | $\mathbf{41.7}$ $\mathbf{(2.8_{\pm 1.0})}$ | $\mathbf{36.6}$ $\mathbf{(2.5_{\pm 1.0})}$ | 2.9 | $3.08e^{09}$ (58.5×) | $1.42e^{13}$ (99.6×) |

Table 14: Results for sample unlearning (half of the dataset is the forget data). For efficacy metrics, baseline results are expressed as *mean metric value (mean $\Delta \pm$ standard deviation)*, with $\Delta$ in parentheses representing the average absolute difference from the *Retrain* baseline. The last column (Avg. Gap) reports the average $\Delta$ across Forget Acc. and MIAs. Lower Avg. Gap indicates better unlearning. For efficiency metrics, higher × (reduction over Retrain baseline) are better.

thus preserving ranks and head–tail margins. In contrast, MSE measures coordinate-wise Euclidean error, disregards simplex geometry and relative ratios, and therefore averages modes, blurring ranks and compressing margins, yielding weaker forgetting.

### D.8 Ablation on Virtual Teacher's Non-True Structure

In Section 5.3, we investigated whether distilling the knowledge encoded in non-true output geometry maintains model utility during FedQUIT unlearning. We introduced a spectrum of virtual teachers, ordered from least to most informative in the non-true structure they retain: flatten (uniform over non-true classes), top-K (keeps only the head, collapses the tail), rank-only logit ladder (preserves rank but not gaps), probability ladder (preserves rank with equal steps in probability),

| Setting | Loss | R ($\downarrow$) | Forget Acc. | MIA (Song) | MIA (Yeom) |
|---|---|---|---|---|---|
| IID | KL Div. | 10.0 | 59.42 (**1.4**) | 60.64 (**5.0**) | 51.82 (**3.0**) |
| | CE | 10.3 | 59.63 (1.7) | 61.31 (5.7) | 52.78 (4.0) |
| | CS | 11.1 | 61.48 (3.5) | 62.73 (7.1) | 57.32 (8.4) |
| | MSE | **4.6** | 70.36 (12.4) | 69.10 (13.5) | 63.31 (14.4) |
| Non-IID | KL Div. | 3.7 | 34.62 (**3.5**) | 49.71 (**5.5**) | 33.90 (**3.8**) |
| | CE | 2.7 | 32.05 (4.0) | 49.56 (6.9) | 32.82 (4.0) |
| | CS | 3.5 | 35.01 (4.3) | 51.72 (8.0) | 35.43 (5.1) |
| | MSE | **1.1** | 39.12 (6.5) | 55.52 (12.6) | 38.51 (7.3) |

Table 15: Ablation on distillation loss for FedQUIT (CIFAR-100, ResNet-18, $E = 1$). Values in parentheses indicate the absolute mean gap with *Retrain* (smaller is better) after recovery.

| | Method | After Unlearning | | | R $\downarrow$ | After Recovery | | |
|---|---|---|---|---|---|---|---|---|
| | | Test Acc. ($\Delta \downarrow$) | Retain Acc. ($\Delta \downarrow$) | Forget Acc. ($\Delta \uparrow$) | | Test Acc. ($\Delta \downarrow$) | Retain Acc. ($\Delta \downarrow$) | Forget Acc. ($\Delta \downarrow$) |
| IID | Retrain | 58.31 | 82.72 | 58.02 | | 58.31 | 82.73 | 58.04 |
| | Flatten ($\tau \to \infty$) | 1.61 (56.7) | 1.61 (81.1) | 1.61 (56.4) | 16.9 | 59.20 (0.9) | 74.02 (8.7) | 67.10 (9.1) |
| | $\tau = 4.0$ | 1.80 (56.5) | 1.90 (80.8) | 1.80 (56.2) | 11.3 | 58.74 (0.4) | 74.92 (7.8) | 68.36 (10.3) |
| | $\tau = 3.0$ | 5.20 (53.1) | 5.20 (77.5) | 5.10 (52.9) | 10.0 | 58.67 (0.4) | 75.27 (7.5) | 68.73 (10.7) |
| | $\tau = 2.0$ | 14.65 (43.7) | 15.57 (67.2) | 13.56 (44.5) | 8.0 | 58.53 (0.2) | 74.54 (8.2) | 65.47 (7.5) |
| | $\tau = 1.5$ | 18.10 (40.2) | 19.00 (63.7) | 15.72 (42.3) | 9.7 | 58.53 (0.2) | 74.20 (8.5) | 61.35 (3.3) |
| | FedQUIT ($\tau = 1.0$) | 25.51 (32.8) | 27.23 (55.5) | 23.54 (34.5) | 10.0 | 58.70 (0.4) | 76.62 (6.1) | 59.42 (1.4) |
| N-IID ($\alpha = 0.1$) | Retrain | 51.02 | 64.61 | 33.52 | | 51.02 | 64.61 | 33.52 |
| | Flatten ($\tau \to \infty$) | 16.02 (35.0) | 18.40 (46.2) | 2.10 (31.4) | 1.2 | 52.60 (1.6) | 63.40 (1.2) | 48.50 (15.0) |
| | $\tau = 4.0$ | 22.90 (28.1) | 27.90 (36.7) | 0.90 (32.6) | 1.5 | 51.98 (1.0) | 63.20 (1.4) | 46.07 (12.6) |
| | $\tau = 3.0$ | 29.74 (21.3) | 36.70 (27.9) | 0.00 (33.5) | 2.0 | 52.36 (1.3) | 63.31 (1.3) | 44.39 (10.9) |
| | $\tau = 2.0$ | 35.85 (15.2) | 45.30 (19.3) | 1.40 (32.1) | 2.7 | 52.21 (1.2) | 63.60 (1.0) | 42.25 (8.7) |
| | $\tau = 1.5$ | 39.04 (12.0) | 49.60 (15.0) | 2.00 (31.5) | 2.7 | 51.35 (0.3) | 63.17 (1.4) | 38.23 (4.7) |
| | FedQUIT ($\tau = 1.0$) | 45.24 (5.8) | 55.82 (8.8) | 8.02 (25.5) | 3.7 | 52.04 (1.0) | 63.22 (1.4) | 34.62 (3.5) |

Table 16: Effect of temperature scaling for virtual teacher's non-true structure ($\tau=1.0$ is the default choice for FedQUIT). Results just after unlearning and after recovery. $R$ means recovery rounds. In parenthesis, gap with *Retrain*. ResNet-18, CIFAR-100.

and FedQUIT (preserves the entire non-true vector). Here, we provide details on these alternative virtual-teacher designs. Figure 4 provides a toy $C=10$ example that visualizes how each virtual-teacher edit acts on the teacher's output geometry. Rows correspond to one construction; within each row, the first panel shows teacher logits before the edit, the second panel shows the edited logits where the true-class logit is clamped to $v = \min_c z_c$ and the non-true logits are reshaped by the method (see below detail), and the third panel shows the post-edit probabilities sorted from highest to lowest (the true class is highlighted in red). By construction, the post-edit true-class probability is the same across methods. See below for formal definitions.

In addition to these structural variants, we also study the effect of temperature scaling applied exclusively to the non-true logits. Table 16 reports results for a range of temperatures. Increasing the temperature progressively smooths the non-true structure and shifts the behavior of FedQUIT toward that of flattened or highly smoothed teachers. A clear trend, consistent with the structure ablation, emerges: (i) higher temperatures progressively degrade retain and test utility immediately after unlearning; and (ii) higher temperatures lead to less effective forgetting after recovery.

**Alternative virtual teachers (structure ablation).** For a minibatch example $(\mathbf{x}_i, y_i)$ with teacher logits $\mathbf{z}_i^g = [z_{i,1}^g, \ldots, z_{i,C}^g]$, let $M = C - 1$, temperature $\tau > 0$, the per-sample anchor $v_i = \min_c z_{i,c}^g$, and the non-true exponential mass $S_i = \sum_{c \neq y_i} \exp(z_{i,c}^g / \tau)$. Denote by $r(c) \in \{0, \ldots, M-1\}$ the rank of class $c \neq y_i$ when sorting non-true classes by *descending* $z_{i,c}^g$ (so $r=0$ is the largest non-true logit). All variants below set the true-class logit to $v_i$ and differ only in how they reshape the non-true logits.

*Flatten (uniform over non-true; mass-preserving).* Replace all non-true logits by a single constant that preserves non-true mass:

$$\alpha_i = \tau \left( \log \sum_{c \neq y_i} e^{z_{i,c}^g / \tau} - \log M \right), \qquad z_{i,c}^{\text{virt}} = \begin{cases} v_i, & c = y_i, \\ \alpha_i, & c \neq y_i. \end{cases}$$

*Logit ladder (rank-only; mass-preserving).* Build a decreasing ladder over non-true logits with minimum $v_i$ and preserve non-true mass. Solve for $\lambda_i \geq 0$:

$$e^{v_i/\tau} \sum_{r=0}^{M-1} e^{\lambda_i r/\tau} \;=\; S_i,$$

then set $\beta_i = v_i + \lambda_i(M-1)$ and

$$z_{i,c}^{\text{virt}} = \begin{cases} v_i, & c = y_i, \\ \beta_i - \lambda_i\, r(c), & c \neq y_i. \end{cases}$$

This preserves the non-true order and controls head–tail contrast via the single slope $\lambda_i$.

*Probability ladder (rank-only with equal probability steps; mass-preserving).* Let $Z_i' = e^{v_i/\tau} + S_i$ and $p_{y_i} = e^{v_i/\tau}/Z_i'$. Define an equal-step ladder in probability over the non-true classes (worst to best) with step

$$\Delta_i \;=\; \frac{2\big(1 - (M+1)p_{y_i}\big)}{M(M-1)} \quad \text{(clipped at 0 if negative)},$$

and assign to the class of rank $r(c)$ the non-true probability $q_i(c) = p_{y_i} + (M-1-r(c))\,\Delta_i$. Convert back to logits while preserving non-true mass:

$$z_{i,c}^{\text{virt}} = \begin{cases} v_i, & c = y_i, \\ \tau\big(\log Z_i' + \log q_i(c)\big), & c \neq y_i. \end{cases}$$

*Top-K (keep head; collapse tail; mass-preserving).* Let $S_i$ be the indices of the $K$ largest non-true logits and $R_i$ the remaining non-true indices. Set

$$\alpha_{i,K} \;=\; \tau\left(\log \sum_{c \in R_i} e^{z_{i,c}^g/\tau} - \log |R_i|\right), \qquad z_{i,c}^{\text{virt}} = \begin{cases} v_i, & c = y_i, \\ z_{i,c}^g, & c \in S_i, \\ \alpha_{i,K}, & c \in R_i. \end{cases}$$

*Incompetent teacher (uniform over all classes; not mass-preserving).* Set a fully uniform target over classes, e.g.,

$$z_{i,c}^{\text{virt}} = \kappa \ \text{(same for all } c) \quad \Rightarrow \quad p_{i,c}^{\text{virt}} = 1/C.$$

All variants except the incompetent teacher satisfy $\sum_{c \neq y_i} \exp\big(z_{i,c}^{\text{virt}}/\tau\big) = S_i$; with the true-class fixed at $v_i$, this keeps the teacher's post-edit true-class probability unchanged. The student is then trained by KD to match the virtual-teacher probabilities on the forget batch via $\text{KL}\big(\mathbf{p}_i^{\text{virt}} \,\|\, \mathbf{p}_i^s\big)$.

### D.9 SPECIAL VALUES FOR $v$ TUNING

In Table 17, we report results for additional choices of the adaptive per-sample value $v$. We evaluate (i) a $v^\star$ that maximizes the virtual teacher's predictive entropy, and (ii) an extreme value $v \ll \min_c z_{i,c}^g$ such that $p_{i,y_i}^{\text{virt}} \approx 0$, comparing both against the default $v_i = \min_c z_{i,c}^g$. The entropy-maximizing choice $v^\star$ underforgets (theoretical intuition in Appendix 3), whereas the extreme choice yields no meaningful improvement over the default. These results support the parameter-free default $v_i = \min_c z_{i,c}^g$.

### D.10 MEMBERSHIP INFERENCE ATTACKS

In this section, we briefly describe Shokri's attack and Yeom's attack that we use in the experimental results. In general, MIA metrics reflect the information leakage of training algorithms about individual members of the training corpus. MIAs attempt to determine whether an individual sample was included in a model's training. The success rate indicates how many samples in $D_u^{forget}$ are correctly identified as training records. A lower MIA success rate implies less information about $D_u^{forget}$ in the model. We consider most effective the approach that minimizes the absolute difference from the retrained model's MIA value.

| Setting | $v$ Tuning | R ($\downarrow$) | Forget Acc. | MIA (Song) | MIA (Yeom) |
|---------|-----------|------|-------------|------------|------------|
| IID | Retrain | | 58.02 (0.0) | 55.60 (0.0) | 48.83 (0.0) |
| | FedQUIT ($v = \min_c z_{i,c}^g$) | 10.0 | 59.42 (1.4) | 60.64 (5.0) | 51.82 (3.0) |
| | FedQUIT ($v = v^\star$) | 9.8 | 63.91 (4.9) | 61.76 (56.1) | 52.15 (4.3) |
| | FedQUIT ($v \ll \min_c z_{i,c}^g$) | 10.5 | 59.38 (1.5) | 60.54 (4.9) | 51.90 (3.1) |
| Non-IID | Retrain | | 33.52 (0.0) | 44.06 (0.0) | 32.05 (0.0) |
| | FedQUIT ($v = \min_c z_{i,c}^g$) | 3.7 | 34.62 (3.5) | 49.71 (5.5) | 33.90 (3.8) |
| | FedQUIT ($v = v^\star$) | 2.5 | 36.03 (4.9) | 50.63 (6.4) | 35.10 (5.0) |
| | FedQUIT ($v \ll \min_c z_{i,c}^g$) | 4.3 | 31.52 (3.6) | 40.10 (5.7) | 29.98 (4.1) |

Table 17: Ablation on adaptive per-sample $v$ for FedQUIT (CIFAR-100, ResNet-18, $E = 1$). Values in parentheses indicate the absolute mean gap with *Retrain* (smaller is better) after recovery. $R$ means recovery rounds. Here, $v^\star$ denotes the per-sample value of $v$ that maximizes the virtual teacher's predictive entropy. Retrain rows are included for reference; gaps are zero by definition.

**Song's MIA (Song & Mittal, 2021).** This attack is implemented using the prediction confidence-based attack proposed in (Song & Mittal, 2021). This approach includes a training phase where a balanced dataset is sampled from data seen during training and data unseen during traning. For seen data, we use the retain dataset, which includes the training data but excludes any forget data; in federated settings, this retain dataset consists of the combined data from the remaining clients. For unseen data, we use a standard test dataset that the model has never encountered during training and is distinct from the forget dataset. The MIA predictor is trained to identify whether the target model's output resembles that of known (seen) data or unknown (unseen) data. To assess unlearning effectiveness, the MIA predictor is then applied to a model using the forget data as input.

**Yeom's MIA (Yeom et al., 2018).** This attack assumes a white-box setting in which the attacker has knowledge of the model's average training loss. A data sample is predicted to be a member of the training set if its model loss is below this average training loss; otherwise, it is classified as a non-member. It is important to note that this metric is used primarily for comparison with other baselines. In practice, an attacker would need access to each client's local training average loss, which is generally infeasible. Alternatively, the attacker would require knowledge of the global model's average loss across the union of all distributed client datasets, which is similarly challenging to obtain. In our implementation, we assume the latter scenario.

### D.11 HYPER-PARAMETER TUNING AND PRE-PROCESSING

In this Section, we report the hyper-parameter tuning of the various methods we used.

For fairness, we enforce full client participation, as adopted in (Halimi et al., 2022; Guo et al., 2024), even though our method naturally supports the standard partial participation setting.

**ResNet-18 on CIFAR-10/CIFAR-100.** We used a standard ResNet-18 (He et al., 2016) and employed Group Normalization layers, similar to other works with similar settings (e.g., (Kim et al., 2022)). We used SGD as a local optimizer with a learning rate set to 0.1, with a round-wise exponential decay of 0.998, 1 local epoch, local batch size of 32. We pre-processed the training images with random crop, horizontal flip and normalization layers. During unlearning routines, we only apply normalization. As server-side optimizer, we consider SGD with a learning rate set to 1.0 (standard FedAvg aggregation and update application).

**MiT-B0 on CIFAR-100/CUB-200.** We used a visual transformer, i.e., MiT-B0 (Xie et al., 2021), with approximately 3.6M parameters, initialized from a pre-trained model checkpoint trained on ImageNet-1k (69.27% accuracy on test data). We adapted the one-layer classification head to the task, initializing such a layer from scratch. We employed the AdamW optimizer with a client learning rate of 3e-4, with a round-wise exponential decay of 0.998, 1 local epoch, local batch size of 32, and weight decay regularization of 1e-3. As server-side optimizer, we consider SGD with a learning rate set to 1.0 (standard FedAvg aggregation and update application). The images are resized to a resolution of 224x224.

| Setting | Optimizer | $\eta_u$ (range; selected) | $E_u$ | Batch Size |
|---|---|---|---|---|
| CIFAR-100, ResNet-18, IID | Adam | $[1e-1, 1e-6]$; 1e-3 | 1 | 32 |
| CIFAR-100, ResNet-18, Non-IID | Adam | $[1e-1, 1e-6]$; 1e-4 | 1 | 32 |
| CIFAR-100, ResNet-18, Non-IID, Multiple Requests | Adam | $[1e-1, 1e-6]$; 5e-4 | 1 | 32 |
| CIFAR-10 , ResNet-18, Non-IID | Adam | $[1e-1, 1e-6]$; 1e-4 | 1 | 32 |
| CIFAR-100, MiT-B0, Non-IID | AdamW | $[1e-3, 1e-6]$; 5e-4 | 1 | 32 |
| CIFAR-100, MiT-B0, Non-IID, Multiple Requests | AdamW | $[1e-3, 1e-6]$; 5e-4 | 1 | 32 |
| CUB-200 , MiT-B0, Non-IID | AdamW | $[1e-3, 1e-6]$; 3e-4 | 1 | 32 |

Table 18: Hyper-parameter tuning for unlearning routine of FedQUIT.

| Setting | Optimizer | $\eta_u$ (range; selected) | $\eta_p$ (range; selected) | $U$ (range; selected) | $P$ (range; selected) |
|---|---|---|---|---|---|
| CIFAR-100, ResNet-18, IID | SGD | $[1e-1, 1e-6]$; **4e-2** | $[1e-1, 1e-6]$; **5e-2** | $[1, 20]$; **10** | $[1, 20]$; **10 (ES)** |
| CIFAR-100, ResNet-18, Non-IID | SGD | $[1e-1, 1e-6]$; **1e-6** | $[1e-1, 1e-10]$; **4e-4** | $[1, 20]$; **10** | $[1, 20]$; **10 (ES)** |
| CIFAR-10, ResNet-18, Non-IID | SGD | $[1e-1, 1e-6]$; **1e-5** | $[1e-1, 1e-6]$; **1e-3** | $[1, 20]$; **10** | $[1, 20]$; **10 (ES)** |
| CIFAR-100, MiT-B0, Non-IID | AdamW | $[1e-1, 1e-10]$; **1e-9** | $[1e-1, 1e-10]$; **1e-5** | $[1, 20]$; **10** | $[1, 20]$; **10 (ES)** |
| CUB-200, MiT-B0, Non-IID | AdamW | $[1e-1, 1e-10]$; **1e-9** | $[1e-1, 1e-10]$; **1e-6** | $[1, 20]$; **10** | $[1, 20]$; **10 (ES)** |

Table 19: Hyper-parameter tuning for the unlearning ($\eta_u$, $U$) and post-training ($\eta_p$, $P$) routines of **FedOSD** in federated settings. $U$=# unlearning rounds; $P$=# post-training rounds; ES=early stop on utility recovery.

**Unlearning Phase (Federated Settings)**   Here we report the hyper-parameter tuning for the unlearning routine of the various baselines we used in the paper.

**FedQUIT** . For FedQUIT we always used a distillation temperature set to 1, and one local epoch for unlearning $E_u$, with batch size as regular training. The per-setting learning rate for the unlearning routine is reported in Table 18. We used the same tuning for the Incompetent Teacher baseline.

**FedEraser** (Liu et al., 2021): we set the retention interval of rounds between stored updates to 1 (tuned in $\{1, 2\}$) and the calibration epochs (number of local epochs during calibration training) to 0.5.

**PGA** (Halimi et al., 2022).  we set the number of local epochs to 5 as prescribed in the original paper, we set gradient clipping threshold to 5, and we use a large batch size as in the original paper ($[256, 512]$), we tuned the early stopping threshold in $[2.2, 10.0]$ selecting the following one as best for settings: 9.0 (IID), 7.0 (Non-IID). We produced the *reference model* as described in the original paper. As in the original paper, we use $\frac{1}{3}$ of the mean distance to a random models as threshold for weight projection. We used SGD momentum (as in the original paper) with learning rate 1e-2 and momentum 0.9 for ResNet-18, and we used AdamW with learning 3e-4 for MiT-B0 as in regular training.

**MoDe** (Zhao et al., 2023). Five hyper-parameters: the number of memory-guidance and degradation rounds (respectively set to 10 and 6), learning rate for memory-guidance rounds (5e-4 for ResNet-18, 5e-5 for MiT-B0), learning rate for degradation rounds (the default learning rate of the specific setting), and a degradation parameter controlling momentum (0.95 as in the original paper).

**FedAU** (Gu et al., 2024). we used 10 local epochs to train the auxiliary module, and tuned $\lambda_{FedAU}$ for auxiliary module integration in $[0.0, 1.0]$, selecting $0.04$ as the best configuration.

**NoT** (Khalil et al., 2025). we negated the weights of the first layer of the neural network, following the prescription in the original paper.

**FedOSD** (Pan et al., 2025). Four hyper-parameters: the number of unlearning rounds, the number of post-training rounds, the learning rate for unlearning rounds, the learning rate during post-training rounds. Table 19 shows the hyper-parameter tuning we used in experiments.

### D.12 MULTIPLE UNLEARNING REQUESTS

FedQUIT also supports the case where multiple clients request unlearning simultaneously. Algorithm 3 summarizes the server-side procedure.

Let $\mathcal{U} \subseteq \{1, \dots, U\}$ denote the set of requesting clients, each wishing to remove the influence of its local forget subset $D_u^{\text{forget}}$. The server can incorporate multiple requests by running a single

| Setting | Method | Efficacy | | | | Efficiency | |
|---|---|---|---|---|---|---|---|
| | | Test Acc. | Forget Acc. ($\Delta \downarrow$) | MIA$_{[Song]}$ ($\Delta \downarrow$) | MIA$_{[Yeom]}$ ($\Delta \downarrow$) | Communication Bytes ($\times \uparrow$) | Computation FLOPs ($\times \uparrow$) |
| **Multiple Unlearning**, CIFAR-100, Non-IID, ResNet-18, $E$=1, 10 Clients | Original | $53.8_{\pm 0.0}$ | $63.1_{\pm 6.8}$ | $76.0_{\pm 7.6}$ | $61.1_{\pm 8.3}$ | — | — |
| | Retrain | $47.3_{\pm 2.1}$ | $30.1_{\pm 3.3}$ | $39.7_{\pm 3.9}$ | $27.9_{\pm 3.1}$ | $1.44\text{e}^{11}$ | $1.20\text{e}^{15}$ |
| | MoDe | $48.2_{\pm 2.0}$ | $35.7\ (5.6_{\pm 2.0})$ | $44.7\ (5.0_{\pm 2.7})$ | $32.2\ (4.1_{\pm 3.3})$ | $1.74\text{e}^{10}\ (8.3\times)$ | $1.36\text{e}^{14}\ (8.8\times)$ |
| | FedAU | $48.3_{\pm 2.3}$ | $35.6\ (5.7_{\pm 2.4})$ | $45.6\ (6.4_{\pm 3.6})$ | $33.6\ (5.7_{\pm 4.1})$ | $7.90\text{e}^{09}\ (18.2\times)$ | $6.60\text{e}^{13}\ (18.2\times)$ |
| | NoT | $48.1_{\pm 2.2}$ | $39.2\ (9.1_{\pm 2.8})$ | $50.5\ (11.3_{\pm 3.4})$ | $41.8\ (13.9_{\pm 4.5})$ | $4.74\text{e}^{09}\ (30.4\times)$ | $3.96\text{e}^{13}\ (30.3\times)$ |
| | FedQUIT | $48.5_{\pm 1.9}$ | $\mathbf{32.7}\ (\mathbf{2.6}_{\pm 1.2})$ | $\mathbf{41.9}\ (\mathbf{2.2}_{\pm 1.1})$ | $\mathbf{30.1}\ (\mathbf{2.2}_{\pm 2.0})$ | $\mathbf{2.33\text{e}^{09}}\ (\mathbf{61.7\times})$ | $\mathbf{1.95\text{e}^{13}}\ (\mathbf{61.5\times})$ |
| **Multiple Unlearning**, CIFAR-100, Non-IID, MiT-B0, $E$=1, 10 Clients | Original | $75.0_{\pm 0.0}$ | $84.6_{\pm 5.3}$ | $76.8_{\pm 9.0}$ | $73.8_{\pm 1.4}$ | — | — |
| | Retrain | $70.9_{\pm 1.3}$ | $54.3_{\pm 3.5}$ | $45.4_{\pm 3.6}$ | $43.5_{\pm 1.6}$ | $1.06\text{e}^{10}$ | $3.40\text{e}^{15}$ |
| | MoDe | $71.5_{\pm 0.6}$ | $59.4\ (5.1_{\pm 1.3})$ | $50.4\ (5.0_{\pm 1.8})$ | $47.6\ (4.1_{\pm 2.9})$ | $5.68\text{e}^{09}\ (1.8\times)$ | $1.69\text{e}^{15}\ (2.0\times)$ |
| | FedAU | $71.4_{\pm 0.9}$ | $58.9\ (4.6_{\pm 2.3})$ | $50.1\ (4.7_{\pm 2.3})$ | $47.2\ (3.7_{\pm 1.9})$ | $3.19\text{e}^{09}\ (3.3\times)$ | $1.02\text{e}^{15}\ (3.3\times)$ |
| | NoT | $71.4_{\pm 0.9}$ | $69.9\ (15.6_{\pm 6.3})$ | $62.5\ (16.6_{\pm 8.9})$ | $59.0\ (15.5_{\pm 7.5})$ | $6.80\text{e}^{08}\ (15.6\times)$ | $2.18\text{e}^{14}\ (15.6\times)$ |
| | FedQUIT | $71.6_{\pm 0.7}$ | $\mathbf{56.3}\ (\mathbf{2.0}_{\pm 1.0})$ | $\mathbf{47.1}\ (\mathbf{1.7}_{\pm 0.9})$ | $\mathbf{45.3}\ (\mathbf{1.8}_{\pm 1.0})$ | $\mathbf{4.78\text{e}^{08}}\ (\mathbf{22.2\times})$ | $\mathbf{1.53\text{e}^{14}}\ (\mathbf{22.2\times})$ |

Table 20: Post-recovery performance of FedQUIT and baselines when two clients out of ten request unlearning. The results are averaged over five different pairs of clients working as requesting clients. Results are expressed as *mean metric value (mean $\Delta \pm$ standard deviation)*, with $\Delta$ representing the average absolute difference with *Retrain*. Lower $\Delta$ corresponds to better unlearning.

| Setting | Method | Test Acc. | Retain Acc. | Forget Acc. | Comm. (Bytes) | Comp. (FLOPs) |
|---|---|---|---|---|---|---|
| **Multiple Unlearning** CIFAR-100, Non-IID ResNet-18, $E$=1, 10 Clients | Original | $53.8_{\pm 0.0}$ | $69.1_{\pm 1.8}$ | $63.1_{\pm 6.8}$ | | |
| | Retrain | $47.3_{\pm 2.1}$ | $66.6_{\pm 2.1}$ | $30.1_{\pm 3.3}$ | $1.44\text{e}^{11}$ | $1.20\text{e}^{15}$ |
| | MoDe$^\dagger$ | $8.7_{\pm 1.5}$ | $9.2_{\pm 1.0}$ | $2.3_{\pm 0.7}$ | $1.33\text{e}^{10}$ | $1.11\text{e}^{14}$ |
| | FedAU | $5.0_{\pm 1.1}$ | $5.4_{\pm 0.9}$ | $4.9_{\pm 1.1}$ | $8.21\text{e}^{07}$ | $1.84\text{e}^{12}$ |
| | NoT | $9.3_{\pm 0.0}^{*}$ | $9.4_{\pm 1.3}$ | $9.0_{\pm 1.1}$ | $0$ | $0$ |
| | FedQUIT | $38.1_{\pm 1.6}$ | $47.9_{\pm 0.6}$ | $19.9_{\pm 2.5}$ | $1.80\text{e}^{08}$ | $1.50\text{e}^{12}$ |
| **Multiple Unlearning** CIFAR-100, Non-IID MiT-B0, $E$=1, 10 Clients | Original | $75.0_{\pm 0.0}$ | $85.5_{\pm 1.3}$ | $84.6_{\pm 5.3}$ | | |
| | Retrain | $70.9_{\pm 1.3}$ | $87.2_{\pm 1.1}$ | $54.3_{\pm 3.5}$ | $1.06\text{e}^{10}$ | $3.40\text{e}^{15}$ |
| | MoDe$^\dagger$ | $54.3_{\pm 3.0}$ | $66.2_{\pm 0.9}$ | $42.5_{\pm 5.1}$ | $3.93\text{e}^{09}$ | $1.26\text{e}^{15}$ |
| | FedAU | $1.7_{\pm 0.2}$ | $2.1_{\pm 0.4}$ | $1.9_{\pm 0.3}$ | $1.03\text{e}^{07}$ | $4.61\text{e}^{10}$ |
| | NoT | $41.5_{\pm 0.0}^{*}$ | $45.5_{\pm 1.2}$ | $41.3_{\pm 2.9}$ | $0$ | $0$ |
| | FedQUIT | $57.6_{\pm 2.8}$ | $69.5_{\pm 1.0}$ | $36.7_{\pm 4.5}$ | $5.32\text{e}^{08}$ | $1.70\text{e}^{13}$ |

Table 21: Post-unlearning performance of multiple unlearning when two clients out of ten request unlearning, across two settings. The results are averaged over five different pairs of clients working as requesting clients. $^\dagger$: multi-round unlearning with active target client; $^{*}$: identical unlearned model for any target client (std = 0).

*unlearning round* in which all requesting clients participate. Specifically, each $u \in \mathcal{U}$ receives the current global model $w_t$, executes FedQUIT locally (Algorithm 2) on its forget data, and returns $w_t^{\bar{u}}$ to the server.

After collecting all unlearned snapshots, the server constructs the sanitized global model $w_t^{\bar{\mathcal{U}}}$ by performing a standard FedAvg-like (McMahan et al., 2017) weighted aggregation:

$$w_t^{\bar{\mathcal{U}}} \ = \ \sum_{u \in \mathcal{U}} \frac{n_u}{N_{\mathcal{U}}} \, w_t^{\bar{u}},$$

where $n_u = |D_u^{\text{forget}}|$ denotes the size of the forget dataset of client $u$ and $N_{\mathcal{U}} = \sum_{u \in \mathcal{U}} n_u$. Training then resumes from $w_t^{\bar{\mathcal{U}}}$ using the standard FedAvg procedure while excluding the clients in $\mathcal{U}$ from subsequent participation.

Table 20 and Table 21 report, respectively, the after-recovery and after-unlearning results when two clients request unlearning simultaneously, across the two settings shown on the left. **Findings.** FedQUIT achieves the best retained performance immediately after unlearning (higher Test and Retain accuracy), provides the most accurate post-recovery unlearning metrics, and outperforms all baselines in terms of efficiency (lower communication and computation costs to recover utility).

**Excluded baselines.** FedOSD (Pan et al., 2025) is inherently designed for single-client unlearning: its optimization and guarantees rely on aligning with the gradient of one target client while enforcing orthogonality to all others. Extending this one-versus-rest formulation to multiple simultaneous forgetters would require redefining the update rule and orthogonality constraints, something not addressed in the orignal paper and not straightforward. PGA (Halimi et al., 2022) requires to

---

**Algorithm 3** Server-side handling of multiple unlearning requests in `FedQUIT`.

---

**Input:** Current global model $w_t$; set of unlearning clients $\mathcal{U}$; local forget dataset $D_u^{\text{forget}}$, $n_u = \sum |D_u^{\text{forget}}|$

**Output:** Unlearned global model $w_t^{\bar{\mathcal{U}}}$

1: **Server initiates unlearning round**
2: **for** each $u \in \mathcal{U}$ in parallel **do**
3:     Send $w_t$ to client $u$
4:     Client $u$ locally executes Algorithm 2 on $D_u^{\text{forget}}$
5:     Client returns unlearned snapshot $w_t^{\bar{u}}$

6: **Server performs:**                         ▷ FedAvg-like aggregation of unlearned models
7: Compute total weight $N_{\mathcal{U}} \leftarrow \sum_{u \in \mathcal{U}} n_u$
8: $w_t^{\bar{\mathcal{U}}} \leftarrow \sum_{u \in \mathcal{U}} \frac{n_u}{N_{\mathcal{U}}} w_t^{\bar{u}}$

9: $w_{t+1} \leftarrow w_t^{\bar{\mathcal{U}}}$                               ▷ Initialize next global round
10: Resume standard FedAvg on clients $k \notin \mathcal{U}$

---

build a reference model without the last contribution of the requesting client; the extension of this mechanism with multiple requests is not explicit in the original paper.

### D.13 EXPERIMENTS ON OTHER TASKS

**Tiny-Shakespeare Dataset.** We evaluate our method on the *Tiny Shakespeare* corpus (Karpathy, 2015), framed as a character-level next-token prediction task.

**Experimental Design.** We build a vocabulary by sorting the unique characters appearing in the training corpus, obtaining 65 symbols (letters, digits, punctuation, and whitespace). The corpus is encoded into integer token IDs and partitioned across 80 clients by dividing the token sequence into 80 contiguous, equally sized segments. Each client locally generates supervised examples using a sliding window of length 80, where the model predicts the next character. Both the original model (before unlearning) and the retraining baseline are trained for 200 FedAvg rounds with one local epoch per client. In each federated round, we use a participation rate of 0.1. We use Adam with learning rate of 1e-3.

**Findings.** Table 22 reports the results for the Tiny-Shakespeare setting. Consistently with the trends observed in the other configurations (e.g., Tables 1, 9, and 10), FedQUIT emerges as the most efficient baseline for recovering model utility, requiring the lowest communication and computation budgets to return to retrain-level test accuracy. This efficiency stems from the characteristics of the FedQUIT model immediately after unlearning (left part of Table 22): it exhibits selective forgetting that leads to only minor degradation in test and retain accuracy, while inducing a larger drop in forget accuracy. This selective degradation makes the subsequent recovery procedure considerably lighter. Moreover, FedQUIT achieves the most accurate forgetting among the compared baselines, showing the smallest absolute gap from the *Retrain* forget accuracy, confirming its strong ability to remove target information while quickly recovering overall utility.

## E EXTENDED RELATED WORK

The main text (Section 6) positions FedQUIT within existing families of FU methods based on their core algorithmic mechanisms. Here, we provide an *orthogonal perspective*: instead of grouping methods by mechanism, we compare them along key *practical requirements and deployment constraints*, such as reliance on historical information, proxy data, client availability, architectural modifications, or extensive hyperparameter tuning. This perspective highlights dimensions that are often overlooked in algorithmic taxonomies but are critical in realistic FL deployments, especially under privacy, scalability, and efficiency considerations. In each subsection we contrast representative methods with FedQUIT, emphasizing how our design avoids these additional assumptions and complexities.

**Historical information.** Methods that remove a client by reconstructing or compensating its past contributions require storing server-side histories of per-client updates. Note that: (1) Linking per-

| Method | After Unlearning | | | After Recovery | | | Efficiency | |
|---|---|---|---|---|---|---|---|---|
| | Test Acc. | Retain Acc. | Forget Acc. | Test Acc. | Retain Acc. ($\Delta \downarrow$) | Forget Acc. ($\Delta \downarrow$) | Comm. | Comp. |
| Original | $57.93_{\pm 0.2}$ | $61.56_{\pm 0.3}$ | $66.42_{\pm 0.2}$ | $57.93_{\pm 0.2}$ | $61.56_{\pm 0.3}$ | $66.42_{\pm 0.3}$ | $5.36e^9$ | $1.26e^{15}$ |
| Retrain | $57.43_{\pm 0.2}$ | $61.29_{\pm 0.3}$ | $58.60_{\pm 1.0}$ | $57.43_{\pm 0.2}$ | $61.29_{\pm 0.3}$ | $58.60_{\pm 1.0}$ | $5.36e^9$ | $1.26e^{15}$ |
| MoDe | $50.92_{\pm 0.5}$ | $53.55_{\pm 0.5}$ | $56.38_{\pm 1.0}$ | $57.75_{\pm 0.3}$ | $60.00\,(1.29_{\pm 0.8})$ | $61.30\,(2.70_{\pm 1.5})$ | $2.79e^9$ | $6.57e^{14}$ |
| FedAU | $14.1_{\pm 0.9}$ | $15.2$ | $14.3_{\pm 1.6}$ | $57.82_{\pm 0.9}$ | $60.95\,(0.34_{\pm 0.5})$ | $60.12\,(1.52_{\pm 0.8})$ | $4.72e^9$ | $1.11e^{15}$ |
| NoT | $1.0_{\pm 0.0}$ | $1.2_{\pm 0.2}$ | $1.3_{\pm 0.3}$ | $57.51_{\pm 0.5}$ | $58.31\,(2.98_{\pm 0.4})$ | $59.90\,(1.30_{\pm 0.3})$ | $6.43e^9$ | $1.52e^{15}$ |
| FedOSD | $53.10_{\pm 0.6}$ | $55.9_{\pm 0.8}$ | $51.33_{\pm 1.1}$ | $57.82_{\pm 0.1}$ | $60.81\,(0.48_{\pm 0.5})$ | $60.80\,(2.20_{\pm 1.2})$ | $2.57e^9$ | $6.07e^{14}$ |
| FedQUIT | $53.18_{\pm 0.5}$ | $56.29_{\pm 0.6}$ | $49.10_{\pm 1.0}$ | $57.90_{\pm 0.2}$ | $61.53\,(\mathbf{0.24}_{\pm 0.4})$ | $57.90\,(\mathbf{0.70}_{\pm 0.9})$ | $\mathbf{4.56}e^8$ | $\mathbf{1.07}e^{14}$ |

Table 22: Performance (post–unlearning and post–recovery) on the Tiny-Shakespeare dataset. For each method, the table reports test, retain, and forget accuracies after the unlearning phase, as well as the corresponding metrics after the recovery phase. The Efficiency block summarizes the communication (bytes) and computational (FLOPs) costs incurred during unlearning and recovery. In parentheses, we also report the absolute gap with *Retrain*, where lower values indicate better alignment with the ideal retraining baseline. Lower efficiency costs are preferable.

client update histories to specific clients requesting unlearning undermines FL's privacy design; (2) to some extent it violates the *ephemeral updates* requirement of the FL framework (Kairouz et al., 2021); and (3) at scale, considering a massive number of clients (Dean et al., 2012), several FL rounds, and multiple simultaneous learning tasks, the corresponding storage requirements might become impractical. **FedEraser** (Liu et al., 2021) (and other similar work as (Liu et al., 2022; Wu et al., 2022b;c)) is a representative example that relies on historical updates and calibration to rebuild a sanitized model. In contrast, *FedQUIT does not store or access any history; it performs a short on device unlearning phase guided only by the current global model.*

**Availability of proxy data.** Several unlearning and model repair procedures ((Liu et al., 2022; Wu et al., 2022b;c; Zhang et al., 2025)) rely on public or proxy data to regain utility, which is difficult in privacy sensitive deployments and can mismatch the federation distribution. Furthermore, to be useful, such a public dataset should be semantically similar (and balanced across classes) to the data in the federation (Nayak et al., 2021); assuming the existence and leveraging the usage of a semantically similar public dataset in FL has been identified before as unrealistic (Mora et al., 2024b). *FedQUIT does not use proxy data; it crafts a virtual teacher directly from the current global model on the forget batch at the requesting client.*

**Client availability and coordination.** Procedures that require multiple rounds keep the requesting client active for an extended period and increase coordination cost. **MoDe** Zhao et al. (2023) and **FedOSD** Pan et al. (2025) run multi-stage or multi-round routines where the target client must stay connected across rounds. *FedQUIT performs a single on-device unlearning round and the client can disconnect immediately after, while the server resumes standard training.*

**Stateful client.** In cross device settings clients are expected to be stateless. **PGA** Halimi et al. (2022) requires maintaining client-side state such as the last update to build their reference model and for projections and clipping. *FedQUIT does not require client-side state beyond what is used in ordinary training.*

**Architectural changes.** Altering the training stack complicates deployment. **FedAU** Gu et al. (2024) introduces an auxiliary head trained during normal learning and later merges it into the global model. *FedQUIT keeps the model architecture unchanged.*

**Direct weight manipulations and selectivity.** Server-side weight transforms can be fast but may be weak or not selective. **NoT** Guo et al. (2024) flips the sign of the first layer and resumes training, producing essentially the same unlearned model regardless of which client requested removal. This lack of selectiveness is evident in our empirical results. In contrast, *FedQUIT demonstrates targeted removal of contributions (see Section 5.1).*

**Extensive Tuning.** Several approaches introduce extra schedules, projections, and thresholds beyond standard training hyperparameters. **PGA** (Halimi et al., 2022) requires projection radii, clip-

ping, and ascent settings; **MoDe** (Zhao et al., 2023) and **FedOSD** (Pan et al., 2025) add stage-specific step sizes and numbers of rounds. *FedQUIT is parameter light: temperature is set to 1 at default, batch size equals the usual local batch size, the number of local epochs for unlearning is 1 by default, the per-sample parameter $v_i$ is data adaptive, and the only routine knob is the local unlearning learning rate.*

**Efficiency.** Multi-round and server-client alternating procedures (e.g., (Pan et al., 2025; Zhao et al., 2023)) increase communication and computation, and history-based designs (e.g., (Liu et al., 2021; 2022; Wu et al., 2022a;b)) increase storage, as we proved in our empirical results. Empirically, *FedQUIT attains gaps close to retrain with considerably lower cumulative communication and computation and no extra storage, because it uses one local unlearning round (one local epoch) without proxy data or historical updates.*

**Summary.** These properties suggest that FedQUIT may be suitable for constrained devices, thanks to its single-round, history-free procedure and absence of architectural changes, though evaluating this on real hardware is an interesting direction for future work.

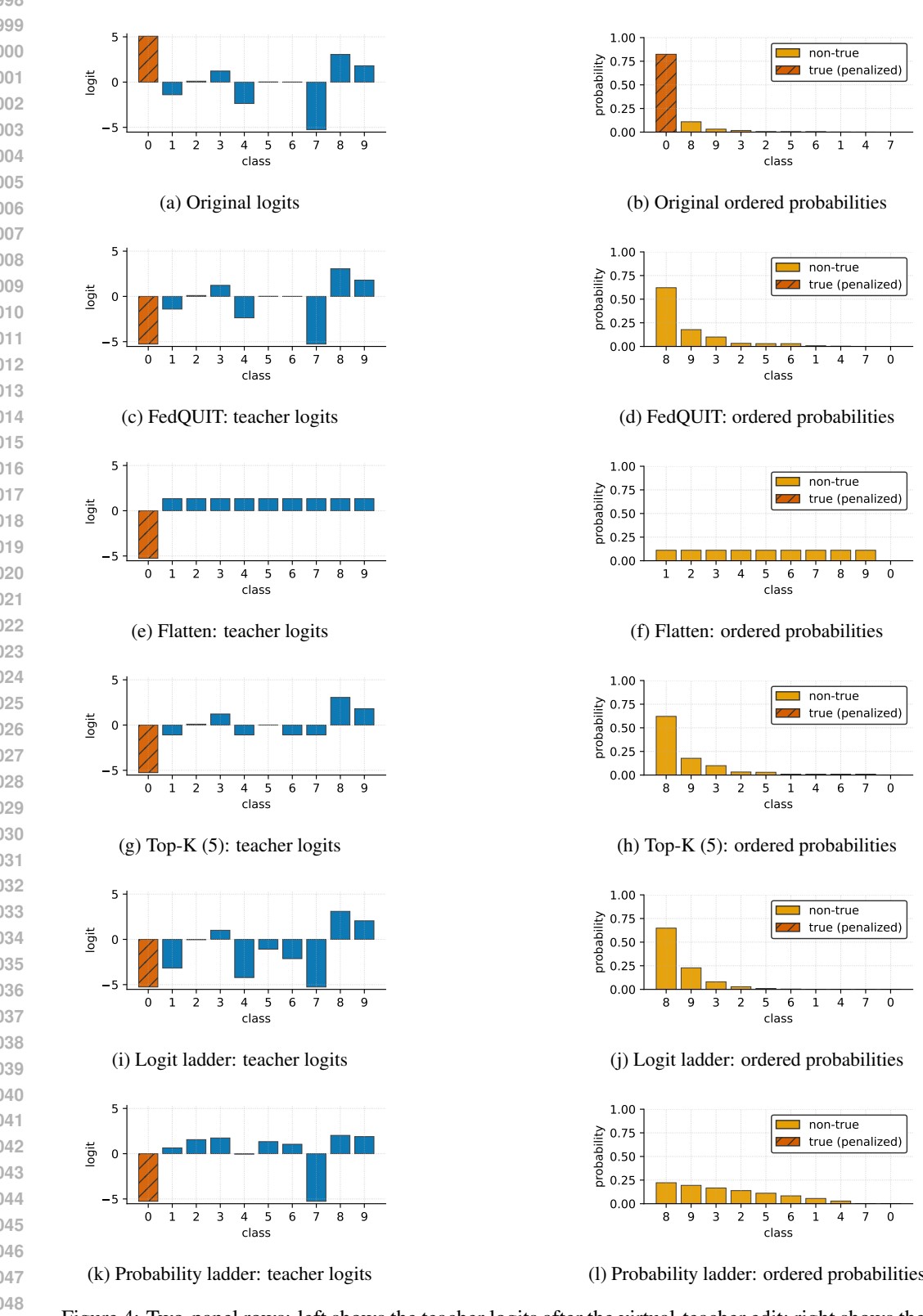

Figure 4: Two-panel rows: left shows the teacher logits after the virtual-teacher edit; right shows the corresponding post-edit probabilities sorted from most to least probable. The first row reports the original model for reference.

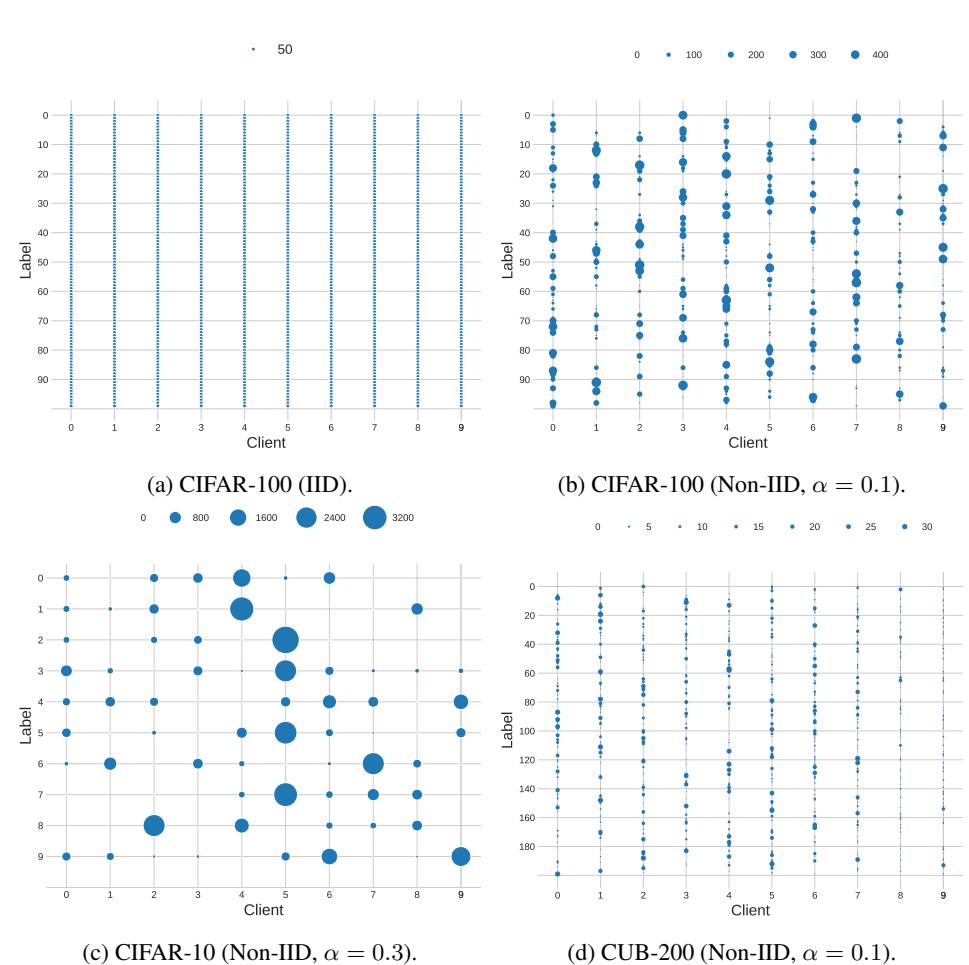

(a) CIFAR-100 (IID).

(b) CIFAR-100 (Non-IID, $\alpha = 0.1$).

(c) CIFAR-10 (Non-IID, $\alpha = 0.3$).

(d) CUB-200 (Non-IID, $\alpha = 0.1$).

Figure 5: Label distribution across clients (0-9) for CIFAR-100 (IID and Non-IID, $\alpha = 0.1$), CIFAR-10 (Non-IID, $\alpha = 0.3$) and CUB-200 (Non-IID, $\alpha = 0.1$).

