# OpenReview forum: "FedQUIT: On-Device Federated Unlearning via a Quasi-Competent Virtual Teacher"
_ICLR.cc/2026/Conference — Submitted to ICLR 2026_

### Official Review · Reviewer_Eci2 · 2025-10-20

**Soundness:** 3
**Presentation:** 3
**Contribution:** 2
**Rating:** 4
**Confidence:** 2

**Summary:**

This paper proposes a federated unlearning (FU) algorithm called FedQUIT, which aims to execute 'forget' requests directly on client devices without requiring historical gradients or additional public data on the server side. The core design idea is to modify the logits of the global model on the data to be forgotten i.e. penalizing the logits of the true class while preserving the relationships among non-true classes, thereby constructing a virtual teacher; the local model (student model) then mimics this teacher through one round of on-device knowledge distillation, after which the standard FedAvg training process resumes. In the experimental section, on CIFAR-10, CIFAR-100, and CUB-200, compared with six state-of-the-art methods (FedEraser, PGA, MoDe, FedAU, NoT, FedOSD), the results show higher forgetting effectiveness, along with lower communication and computation overhead. The paper also includes a theoretical analysis, proving that the parameter perturbation induced by FedQUIT is bounded and preserves the convergence properties of FedAvg.

**Strengths:**

1. No need to modify the FedAvg framework, just add a single round of distillation step to deploy.
2. The boundedness analysis of parameter perturbations and the convergence proof of FedAvg after forgetting provide formal guarantees for the stability of the method, filling the theoretical gap in many empirical federated forgetting works.
3. The experiments cover a wide range of topics, including multiple datasets, multiple architectures (ResNet-18 and MiT-B0), and comparisons with various SOTA methods.

**Weaknesses:**

1. The paper frames the "quasi-competent virtual teacher" as a key innovation, but it is an incremental adaptation of KD for FU. Prior work (e.g., FedNTD, Lee et al., 2022) already uses KD with modified true-class logits in FL (albeit for heterogeneity, not unlearning), and the "preserve non-true class geometry" idea is standard in KD (Hinton et al., 2015). The only novel tweak, i.e. setting v to min logit, is a hyperparameter choice, not a conceptual breakthrough.
2. All proofs rely on idealized smoothing conditions and lack rigorous definitions for non-independent and identically distributed (Non-IID) cases.
3. This paper only analyzes the sensitivity of v value.

**Questions:**

1. Table 1 labels FedEraser as "limited in effectiveness", lacking comparative results and not tested in large-scale scenarios. Does this mean that FedQUIT is less effective than FedEraser, and that its advantage may diminish as the number of clients increases?
2. What is the rationale for fixing the distillation temperature τ to 1 during training? Have you tried using higher τ to smooth the probability distribution?
3. Section D.8 excludes MoDe and FedOSD from the sample oblivion experiment due to "unclear how to use holdout data". However, MoDe's memory bootstrapping phase and FedOSD's post-training phase explicitly allow clients to use holdout data, why can't these baselines be adapted? Please either include these baselines in the sample oblivion experiment or provide a detailed technical explanation for the exclusion.
4. How is "on-device" implemented? Are the computing and storage limitations of mobile devices fully considered?
5. Does FedQUIT support forgetting multiple clients simultaneously? If multiple requests arrive in parallel, will the algorithm still converge?

---

> ### Author Response · Authors · 2025-11-22
> **Response to Reviewer Eci2 (Weakness 1, Weakness  3)**
>
> **W1 — On the novelty of the quasi-competent virtual teacher.**
>
> We thank the reviewer for the opportunity to clarify the conceptual distinction. Beyond the different goals (heterogeneity mitigation vs. unlearning), FedQUIT and FedNTD (Lee et al., 2022) differ technically in two fundamental ways (see also Appendix A.2):
>
> 1. **Treatment of the true-class logit.**
>
>     FedNTD **removes** the true-class logit from distillation, whereas FedQUIT **explicitly penalizes** it.
>
>     This step is essential in unlearning: penalizing the true-class logit generates the forgetting signal, while preserving the non-true geometry maintains useful semantic structure. Our Lemma 1 formalizes why this mechanism is necessary for selective forgetting.
>
> 2. **Objective design.**
>
>     FedNTD uses KL as a regularizer on top of cross-entropy, whereas FedQUIT performs pure KL minimization on forget data only. The entire update is driven by the virtual teacher, which makes FedQUIT a genuine distillation-based unlearning procedure.
>
>
> The choice of $v = \min_{c} z^{g}_{i,c}$ (see Sec. 3 “Choice of $v$") is not a hyperparameter tweak but a **parameter-free, sample-adaptive construction** that preserves non-true geometry while enforcing forgetting on the true class. FedNTD has no analogous mechanism, since the true-class logit is absent from its distillation rule. We further support the choice theoretically and empirically (Appendix A.2 and Section 5.3).
>
> We also note that recent FU methods use much simpler perturbations (e.g., random labels in FedAU (Gu et al., 2024) or weight negation in NoT (Khalil et al., 2025)). Our experiments show that FedQUIT outperforms all these state-of-the-art baselines both in efficacy and efficiency.
>
> Gihun Lee et al.,  Preservation of the  global knowledge by not-true distillation in federated learning. In Advances in Neural Information Processing Systems, 2022.
>
> Khalil et al., Not: Federated unlearning via weight negation. In Proceedings of the Computer Vision and Pattern Recognition Conference, 2025.
>
> Gu et al., Unlearning during learning: an efficient federated machine unlearning method. In Proceedings of the Thirty-Third International Joint Conference on Artificial Intelligence, 2024.
>
> ---
> **W3 —** *“This paper only analyzes the sensitivity of*  $v$*.”*
>
> We thank the reviewer and clarify that our analysis extends beyond the sensitivity of $v$. Specifically:
>
> 1. **Full FU metrics** (Sec. 5.1, Table 1, Appendix Tables 9-10): test/retain/forget accuracy, MIAs, post-unlearning and post-recovery performance, and communication/computation costs
> 2. **Entropy analysis** (Sec. 5.2): FedQUIT’s output entropy closely matches that of Retrain.
> 3. **Teacher-structure ablation** (Sec. 5.3): flatten, top-K, rank-only ladder, probability ladder, FedQUIT.
> 4. **New temperature-scaling ablation** (Sec. 5.3 and Appendix D.8): varying the temperature on non-true logits; higher $\tau \rightarrow$ smoother teacher $\rightarrow$ degraded retention and weaker forgetting.
> 5. **Sensitivity of** $v$ (Sec. 5.4): effect of varying $v$ and the behavior of the adaptive per-sample choice.
> 6. **Eight client-unlearning settings** (list of settings in Appendix Table 6), including two new 100-client scenarios, compared against six baselines plus straightforward baselines such as random labels and uncostrained gradient ascent.
> 7. **Six sample-unlearning settings** (Sec. 5.1, Appendix D.6): 1% atypical forget sets, 10% IID/Non-IID, and 50% Non-IID (four settings added in the revision).
> 8. **Distillation-loss ablation** (Appendix D.7): KL (default) vs. MSE, CE, cosine similarity.
> 9. **Multiple-client unlearning** (Appendix D.12).
> 10. **Experiments beyond image classification** (Appendix D.13).
>
> Overall, the paper provides a broad and comprehensive set of ablations and analyses that extend beyond the sensitivity of $v$, covering multiple dimensions of the teacher design, distillation behavior, and unlearning dynamics.

---

> ### Author Response · Authors · 2025-11-22
> **Response to Reviewer Eci2 (Question 1, Question 2)**
>
> **Q1 — Clarifying FedEraser’s effectiveness vs. efficiency (and large-scale behavior)**
>
> We thank the reviewer for raising this point.
>
> **1. Effectiveness vs. efficiency in Table 1.** FedEraser is highly *effective* at forgetting, but *inefficient* in communication, computation, and storage. As described in Algorithm 1 of FedEraser, the retain clients must run a full set of calibration rounds comparable to retraining, and the server must store all historical updates (scaling with rounds $\times$ clients). Its strong effectiveness is therefore expected: FedEraser is essentially a form of retraining, whereas **FedQUIT completes unlearning in a single round**, without stored updates and without multi-round calibration.
>
> **2. Empirical evidence of FedEraser’s low efficiency.** In the revision, we added a fixed-budget evaluation (new Table 2; Appendix Table 11). When all methods are constrained to the **same communication budget**, even substantial budgets such as **50% of the retraining budget**, FedEraser performs significantly worse than every other baseline. This confirms that FedEraser’s effectiveness depends on consuming retraining-level resources, while FedQUIT preserves strong unlearning quality without requiring such costs.
>
> **3. Behavior in large-scale FL.** FedEraser’s own authors note that its performance degrades as the number of clients increases and collapses if calibration rounds are reduced. Furthermore, Algorithm 1 in (Liu et al., 2021) requires coordinated participation of retain clients during several rounds of calibration, which is unrealistic under partial participation.
>
> **4. Larger-scale experiments.** We added **two new 100-client experiments** (new blocks in Table 1, described in Sec. 4), with partial participation. We do not include FedEraser in the comparison for the reasons above. These confirm the same trend: **FedQUIT remains effective and highly efficient, outperforming all the considered baselines.**
>
> Liu et al., Federaser: Enabling efficient client-level data removal from federated learning models. In 2021 IEEE/ACM 29th International Symposium on Quality of Service (IWQOS), pp. 1–10, 2021.
>
> ---
> **Q2 — On fixing the distillation temperature and using higher $\tau$.**
>
> We thank the reviewer. We now include **new temperature-scaling experiments** and clarify their connection to our teacher-structure ablations.
>
> **Temperature-scaling experiments.** We apply temperature scaling ($\tau\in{1.5,2.0,3.0,4.0}$). As shown in Appendix D.8 (Table 16), higher $\tau$ steadily degrades post-unlearning utility and post-recovery forgetting accuracy, pushing FedQUIT toward the behaviour of flattened or highly smoothed teachers.
>
> **Connection to teacher-structure ablations.** This trend matches our existing ablations on teacher structure, which correspond to increasingly smoothed teachers. Across all variants, removing or smoothing non-true-class geometry harms performance of FedQUIT.
>
> **Why $\tau=1$ in the main method.** We fix $\tau=1$ because (i) our theoretical guarantees rely on the raw global-model logit geometry, which temperature scaling would modify, and (ii) empirical evidence already showed that smoother/higher-temperature teachers perform worse. We now make this explicit in Sec. 5.3.
>
> **Changes to the paper:** updated Sec. 5.3; added temperature-scaling results in Appendix D.8 (Table 16).

---

> ### Author Response · Authors · 2025-11-22
> **Response to Reviewer Eci2 (Question 3, Question  4)**
>
> **Q3 — Inclusion of MoDe and FedOSD in Sample-Unlearning Experiments.**
>
> We thank the reviewer for the insightful comment. **We have now included both MoDe and FedOSD** in the sample-unlearning experiments.
>
> Originally, we excluded them because their papers do not specify algorithms for sample-level unlearning. However, following the suggestion in FedOSD (Pan et al., 2025), **we adapted both methods by treating the forget samples as a “virtual client’’ throughout their multi-round unlearning phases**. Section D.6 has been updated accordingly, and both baselines are now reported.
>
> **Additional sample-unlearning experiments.** As part of the revision, we also added **1% (statistically atypical)** and **10% (random)** forget-set experiments under both IID and Non-IID settings (ResNet-18). For IID, the 1% atypical case corresponds to forgetting all samples from a random class. Results for the 1% Non-IID case appear in **new Table 4**; complete results (post-unlearning and post-recovery) are provided in **Appendix D.6**.
>
> **Findings.** At **1% forget data**, FedQUIT consistently outperforms all baselines (including MoDe and FedOSD) showing lower degradation after unlearning, more accurate forgetting after recovery, and substantially lower cost (≈4×). Similar trends hold for the 10% and 50% settings.
>
> **Changes to the paper:** updated Sec. 5.1; added Table 4; added Appendix Tables 12–13; extended Appendix D.6 to include MoDe and FedOSD.
>
> ---
> **Q4 —  On-device computation and memory requirements.** FedQUIT is designed so that any client capable of participating in standard FL can also perform on-device unlearning without additional hardware assumptions. The on-device phase is intentionally lightweight and closely matches the computation and memory footprint of a round with a single local epoch of FedAvg.
>
> Concretely, the client only needs to (with reference to our Algorithm 1):
>
> (1) run a forward pass of the current global model on its forget data to obtain teacher logits (negligible memory, computation, and storage),
>
> (2) modify these logits to construct the virtual teacher (negligible computation), and
>
> (3) run one epoch of knowledge distillation using the same model architecture and the same batch size as in regular training (similar cost to a standard FedAvg round).
>
> Thus, the overall cost is comparable to a normal local training round. If the client forgets all its local data, the cost is approximately identical to a standard round; if it forgets only a subset, the computation requirements are strictly lower, since only that subset is processed.

---

> ### Author Response · Authors · 2025-11-23
> **Response to Reviewer Eci2 (Question 5)**
>
> **Q1. Multiple Unlearning Requests.**
>
> We thank the reviewer for this request, which gave us the opportunity to provide a useful explicit extension. In the revised manuscript, we include both the algorithmic procedure to handle multiple simultaneous unlearning requests and the corresponding empirical results for two different settings (Appendix D.12), and we now explicitly reference this extension in Section 5 of the main paper.
>
> **Algorithm.** FedQUIT also supports the case where multiple clients request unlearning simultaneously. Let $\mathcal{U} \subseteq \{1,\ldots,U\}$ be the set of requesting clients, each aiming to remove its local forget subset $D_u^{\mathrm{forget}}$.
> In a single unlearning round, every $u \in \mathcal{U}$ receives the current global model $w_t$, applies FedQUIT locally (Algorithm 1) to obtain an unlearned snapshot $w_t^{\bar u}$, and returns it to the server.
> The server then constructs a sanitized global model via a FedAvg-style aggregation:
> $
> w_t^{\bar{\mathcal{U}}}
> = \sum_{u \in \mathcal{U}}
> \frac{n_u}{N_{\mathcal{U}}} w_t^{\bar u},
> $
> with $n_u = |D_u^{\mathrm{forget}}|$ and $N_{\mathcal{U}} = \sum_{u\in\mathcal{U}} n_u.$
> Training resumes from $w_t^{\bar{\mathcal{U}}}$ while excluding all clients in $\mathcal{U}$ from further participation.
>
> **New Experimental Results.** We provide results for multiple simultaneous unlearning requests. We consider CIFAR-100 (Non-IID) with either ResNet-18 or MiT-B0 and 10 clients, where two clients request unlearning simultaneously. We average results over five random pairs of clients and report mean and standard deviation. We do not include PGA (Halimi et al., 2022)  and FedOSD (Pan et al., 2025), see below.
>
> **Findings.** FedQUIT achieves the best retained performance immediately after unlearning (higher Test and Retain accuracy), provides the most accurate post-recovery unlearning metrics, and outperforms all baselines in terms of efficiency (lower communication and computation costs to recover utility).
>
> **Excluded Baselines and Adaptations.** FedOSD is inherently designed for **single-client** unlearning: its optimization and guarantees rely on aligning with the gradient of **one** target client while enforcing orthogonality to all others. Extending this one-versus-rest formulation to multiple simultaneous forgetters would require redefining the update rule and the orthogonality constraints, something not addressed in the original paper and not straightforward. PGA requires constructing a reference model without the last contribution of the requesting client; how to generalize this mechanism to multiple simultaneous requests is not explicit in the original paper. We adapt MoDe (Zhao et al., 2023) by providing all unlearning clients with the same degradation model and applying memory guidance in parallel.
>
> **Changes to the paper.** We added a dedicated section in **Appendix D.12** as well as new **Appendix Table 20** and **Appendix Table 21**, and we included an **explicit reference to these experiments in Section 5**, where we introduce the research question (point 1).
>
> **References**
>
> Halimi et al., Federated Unlearning, arXiv:2207.05521, 2022.
>
> Pan et al., Federated Unlearning with Gradient Descent and Conflict Mitigation, AAAI 2025.
>
> Zhao et al., Federated unlearning with momentum degradation. IEEE Internet of Things Journal, 2023.

---

> ### Author Response · Authors · 2025-11-24
> **Response to Reviewer Eci2 (Weakness 2)**
>
> **Clarification of Smoothness Assumptions and Explicit Non-IID Definition.**
>
> Thank you for pointing this out. We acknowledge that the previous version did
> not state the Non-IID assumption rigorously. In the revision, we now clarify
> the precise Non-IID condition used in our analysis, following the bounded
> heterogeneity definition of Li et al. (2020). We also emphasize that
> Lemma 1 and Lemma 2 do not depend on the client data distribution.
>
> Finally, we note that smoothness-type assumptions are standard in
> state-of-the-art FU theory: for example, NoT (Khalil et al., 2025) and
> FedOSD (Pan et al., 2025) both rely on Lipschitz/smoothness assumptions in
> their theoretical analyses. Our assumptions are therefore aligned with
> those used in existing FU methods.
>
> **Explicit Non-IID assumption.**
> We now make explicit the bounded heterogeneity condition of Li et al. (2020).
> In **Assumption (A7) in Appendix C.2** we define
>
> $\Gamma = F^* - \sum_{k\neq u} \pi_k F_k^*$,
>
> where $F^* = \min_w F_{\setminus u}(w)$ is the global optimum after
> removing client $u$, and $F_k^* = \min_w F_k(w)$ are the local optima. This Non-IID constant measures the discrepancy between global and local optima, equals $0$ in the IID case. We assume $\Gamma < \infty$, i.e., the
> mismatch between global and local minima is bounded.
> Our convergence theorem (Theorem 1)
> now explicitly assumes $\Gamma < \infty$.
>
> **References.**
>
> Li et al., On the convergence of FedAvg on non-IID data. ICLR 2020.
>
> Pan et al., Federated Unlearning with Gradient Descent and Conflict
> Mitigation. AAAI 2025.
>
> Khalil et al., NoT: Federated Unlearning via Weight Negation. CVPR 2025.

---

### Official Review · Reviewer_ST2i · 2025-10-27

**Soundness:** 2
**Presentation:** 3
**Contribution:** 2
**Rating:** 4
**Confidence:** 5

**Summary:**

This paper introduces FedQUIT, a novel federated unlearning (FU) framework designed for on-device execution. The method addresses a client's request to remove its data's influence from a global model by employing a single-round, history-free, and proxy-free unlearning procedure based on knowledge distillation. The core idea is a "quasi-competent virtual teacher," a modified version of the global model that, on the forget data, penalizes the true-class logit while preserving the relational geometry of non-true class logits. This simultaneously induces forgetting and retains generalizable knowledge. Extensive experiments demonstrate that FedQUIT achieves a superior unlearning-recovery trade-off compared to six state-of-the-art baselines, while significantly reducing communication and computational overhead relative to retraining from scratch.

**Strengths:**

1. The method's on-device, single-round, history-free, and proxy-free design makes it exceptionally practical for real-world federated learning, particularly in resource-constrained cross-device settings where client availability is intermittent and storing historical updates is infeasible.

2. The concept of the "quasi-competent virtual teacher" is a significant contribution. This carefully crafted distillation target provides a sophisticated way to balance the removal of specific information with the retention of general knowledge, proving more effective than naive supervision perturbation techniques.

3. The experimental validation is a major strength. The paper compares FedQUIT against six strong and relevant baselines across multiple datasets, data distributions (IID and non-IID), and model architectures, using a comprehensive set of efficacy and efficiency metrics that convincingly support its claims.

4. The paper provides solid theoretical insights that justify the forgetting signal and the stability of the post-unlearning model for continued training. These theoretical claims are further substantiated by strong empirical results and insightful ablation studies that analyze the key design choices.

**Weaknesses:**

1. While the specific construction of the virtual teacher is novel, the broader concept of "supervision perturbation" has been explored. The paper would benefit from a more detailed discussion in the related work section that explicitly contrasts its sophisticated soft-target manipulation against simpler hard-label modifications (e.g., label randomization) to better highlight its unique contributions.

2. The provided theory justifies the unlearning mechanism but lacks formal guarantees on how closely the resulting model approximates the gold-standard retrained model. The paper should be strengthened by adding an analysis that bounds the divergence between the unlearned and retrained models.

3. The method's reliance solely on the local forget set creates a risk of "catastrophic unlearning," where small or atypical requests could disproportionately damage the global model. A robustness analysis should be conducted using highly sparse forget requests to evaluate this potential failure mode.

4. The paper fails to justify the necessity of its knowledge distillation framework over a more direct alternative, namely simple gradient ascent on the forget data. An essential ablation study comparing against unconstrained gradient ascent should be added to validate the design choice.

5. The core assumption that the teacher's non-true class geometry is beneficial knowledge is questionable, as it may propagate biases from a poorly calibrated global model. The analysis would be more convincing with ablations on more robust teacher designs, such as applying temperature scaling to the non-true logits.

6. The experimental recovery protocol, which stops training once a target test accuracy is met, may unfairly penalize baselines by halting their recovery prematurely across all metrics. A more robust evaluation should compare all methods after a fixed recovery budget or by reporting the cost-versus-efficacy Pareto frontier.

**Questions:**

1. The concept of supervision perturbation for unlearning has been explored. Could you elaborate on the key advantages of your sophisticated soft-target manipulation compared to simpler hard-label modifications like label randomization, particularly in terms of knowledge retention and recovery efficiency?

2. Your theoretical analysis effectively shows that unlearning occurs and convergence can resume. However, a key claim is that FedQUIT approximates the retrained model. Is it possible to provide any theoretical insights or bounds on the divergence between the model produced by FedQUIT and the ideal, retrained-from-scratch model?

3. How robust is FedQUIT when the unlearning request pertains to a very small or statistically atypical subset of a client's data? Is there a risk that the KD process, focused on this unrepresentative set, could disproportionately damage the model's performance on retained data (i.e., catastrophic unlearning)?

4. To better justify the complexity of the proposed KD framework, could you comment on how FedQUIT compares to a more direct unlearning baseline, such as simple, unconstrained gradient ascent on the forget data's loss function? What are the hypothesized benefits of preserving non-true class geometry over simply maximizing the error on the forget samples?

5. The quality of the "quasi-competent" teacher relies on the global model's outputs. In scenarios where the global model is poorly calibrated or overconfident in its incorrect predictions, could preserving this flawed geometry be detrimental? Have you considered or experimented with more robust teacher designs, such as applying temperature scaling to the non-true logits?

6. The recovery protocol stops when a baseline matches the retrained model's test accuracy. Could this criterion mask deficiencies in other unlearning metrics by halting recovery prematurely for some methods? How might the comparative results change if all methods were evaluated after a fixed recovery budget?

---

> ### Author Response · Authors · 2025-11-22
> **Response to Reviewer ST2i (Weakness/Question 1, Weakness/Question 3)**
>
> **W1 / Q1 - FedQUIT vs. simpler hard-label modifications.**
>
> We thank the reviewer for this helpful comment. We agree that contrasting FedQUIT with simpler hard-label perturbations (e.g., random labels) better highlights our contribution.
>
> In the revised manuscript we: (i) **expanded the related-work discussion** to clarify this distinction; (ii) **added a dedicated analysis in Appendix A.1** formalizing the connection between random labels and teacher–student KD; and (iii) **added random-label supervision as a new baseline in Table 5**.
>
> 1. **Random hard labels correspond to a “random teacher.”**
>
>     As shown in **Appendix A.1**, using random hard labels under cross-entropy is equivalent to KD with a one-hot random teacher. This places label randomization within the same conceptual framework as other teacher-based perturbations.
>
> 2. **But random labels destroy the non–true-class geometry.**
>
>     Both random-hard and uniform teachers eliminate the structure among non-true logits. In contrast, **FedQUIT preserves this geometry**, modifying only the true-class logit while retaining the relative relationships among non-true classes, a key factor behind its stable recovery and selective forgetting. Our ablations in Section 5.3 show that removing this geometry severely harms performance.
>
> 3. **Only FedQUIT satisfies the assumptions behind our theoretical guarantees.**
>
>     Our formal results (Proposition 1; Lemma 1–2; Theorem 1) rely on preserving the ordering and relative magnitudes of the non-true logits. Random-label or uniform-teacher supervision violates these assumptions and thus cannot guarantee bounded perturbation or stable unlearning under our theoretical framework.
>
> 4. **Empirical comparison with random-label supervision.**
>
>     The new “Random’’ baseline in Table 3 confirms this: random-label supervision collapses test and retain accuracy after unlearning, requires longer recovery, and fails to forget reliably (large post-recovery gaps vs. retrain). These behaviors align with the theoretical limitations noted above.
>
>
> **In summary,** while random-label strategies are indeed forms of supervision perturbation, FedQUIT differs fundamentally by preserving the structured non–true-class geometry that underpins both its empirical performance and its theoretical guarantees.
>
> ---
> **W3 / Q3 - Robustness to Small or Atypical Forget Requests.** We thank the reviewer for  helping us strengthen the robustness analysis.
>
> To answer this question, we extended the sample unlearning case that in the original submission considered a subset of 50% random data samples as the forget data.
> We anticipate that FedQUIT remains robust even when unlearning very small or statistically atypical subsets, and our new experiments show no evidence of catastrophic unlearning. Below, we detail the extended analysis.
>
> We now use two additional forget sets: we unlearn the **1% of the local data drawn from the least-represented local classes (statistically atypical)**, and the **10% of the local data drawn randomly**.
>
> We evaluate this new sample-unlearning settings in IID and non-IID settings, with ResNet-18. For the IID case, the statistically atypical case translates to forget all the samples from a random class.
>
> We evaluate these new sample-unlearning settings under both IID and Non-IID scenarios using ResNet-18. For the IID case, the “statistically atypical’’ setting corresponds to forgetting all samples from a random class.
>
> New Table 4 reports the results for the 1% setting in the Non-IID scenario, while **Appendix D.6** provides complete results for all configurations (IID and Non-IID), both post-unlearning and post-recovery.
>
> **Findings.** In the extreme 1% forget-set setting (statistically atypical, Non-IID), FedQUIT outperforms all baselines, exhibiting both lower degradation immediately after unlearning and more accurate forgetting after recovery, while also requiring substantially lower computation and communication costs than second-best baseline (≈4×). Similar trends hold for the less extreme cases (see Appendix D.6). We find **no evidence of “catastrophic unlearning’’**, which FedQUIT is explicitly designed to avoid.
>
> **Changes to the paper:** Added new Table 4, updated Sec. 5.1 with sample-unlearning discussion, added Appendix Tables 12 and 13, and updated Appendix D.6.
>
> If the reviewer has further suggestions for challenging or adversarial data partitions that they consider relevant for evaluating robustness, or potential breaking points for our method, we are happy to test those as well.

---

> ### Author Response · Authors · 2025-11-22
> **Response to Reviewer ST2i (Weakness/Question 4, Weakness/Question 5, Weakness/Question 6)**
>
> **W4 / Q4. FedQUIT vs. unconstrained gradient ascent.** We thank the reviewer for this suggestion. We have added an unconstrained gradient-ascent (GA) baseline in two settings (**Appendix D.2, Tables 9–10**). GA is omitted from the main Table 1 because it yields illusory fast recovery but fails to achieve meaningful forgetting, making comparisons misleading.
>
> Two baselines already included, **FedOSD** (Pan et al., 2025) and **PGA** (Halimi et al., 2022), are controlled versions of GA. FedOSD bounds the loss to prevent gradient explosion; PGA constrains updates within an $\ell_2$ ball with clipping and early stopping. Both underscore that **unconstrained GA is fundamentally unstable**.
>
> The issue is structural: cross-entropy is unbounded above, so GA rapidly drives gradients to explode, collapsing model utility and producing behaviour close to random networks (Pan et al., 2025). Even when GA increases forget loss, the model typically **relearns** the forgotten information during recovery.
>
> FedQUIT, by contrast, yields stable and selective forgetting, supported by our theoretical guarantees (Proposition 1, Lemma 2, Theorem 1), which rely on preserving non-true-class geometry to bound perturbations.
>
> **Findings.** Our new GA experiments confirm this: GA severely damages test and retain accuracy (Appendix Table 10) and fails to achieve lasting forgetting (large post-recovery gaps vs. retrain, Appendix Table 9).
>
> Halimi et al., Federated Unlearning, arXiv:2207.05521, 2022.
>
> Pan et al., Federated Unlearning with Gradient Descent and Conflict Mitigation, AAAI 2025.
>
> ---
>
> **W5 / Q5 — Temperature-Scaled Teachers.**
>
> We appreciate the reviewer’s suggestion and now include (i) **new temperature-scaling experiments** on the non-true logits, and (ii) an explanation of how these results **connect to the teacher-structure ablations** already in the paper.
>
> **Temperature-scaling experiments.** Following the suggestion, we evaluate FedQUIT with temperature scaling applied only to non-true logits ($\tau \in {1.5,2.0,3.0,4.0}$). Results (Appendix D.8, Table 16) show that increasing $\tau$ steadily degrades both post-unlearning utility and post-recovery forgetting accuracy, and pushes FedQUIT toward the behaviour of flattened or highly smoothed teachers. This provides a direct robustness analysis on the teacher design.
>
> **Connection to existing ablations.** These findings align with the teacher-structure variants already in the paper. The alternatives (flatten, probability ladder, logit ladder, Top-K) correspond to increasingly smoothed, or effectively higher-temperature, teachers. As shown in Section 5.3 and Appendix D.8, moving toward these smoother teachers consistently worsens performance, confirming that **preserving non-true-class geometry is crucial** for stable and selective unlearning.
>
> **On poorly calibrated global models.** We agree that if the global model were severely miscalibrated, its outputs could reflect biased geometry. However, this issue originates from the underlying FL training, not from the unlearning mechanism. Moreover, FedQUIT explicitly **penalizes the true-class logit** on forget data (the part most affected by overconfidence) while retaining only the relative geometry among non-true classes. Addressing extreme global miscalibration remains an open problem for the broader FU literature, and no prior work has studied FU under such conditions.
>
> **Changes to the paper:** updated Section 5.3; new temperature-scaling experiments in Appendix D.8 (new Table 16).
>
> ---
>
> **W6 / Q6 - Fixed Budget.** We thank the reviewer for the observation. Below, we provide new results under fixed communication budgets to directly address the concern.
>
> **New comparison under fixed cumulative budgets.** As suggested, we evaluate all methods under fixed cumulative communication budgets (computation shows similar trends; see Table 1). We consider **5%, 10%, 20%, 25%, and 50%** of the total communication cost required to retrain from scratch. **A new paragraph has been added in Section 5.1, with numerical results in Table 2 and extended results in Appendix D.5 and Table 11.**
>
> Experiments were run on CIFAR-100 with ResNet-18 under both IID and Non-IID settings.
>
> Crucially, Table 1 shows that **10% and 20% budgets occur well after most methods have already fully recovered**. Thus, these fixed-budget evaluations assess whether forgetting remains stable once recovery is complete, and whether additional budget could cause forgetting to reverse (precisely the reviewer’s concern).
>
> **Findings.** Across all fixed-budget settings (Table 2 and Appendix Table 11), **FedQUIT consistently outperforms baselines**, achieving lower deltas in Forget Acc. and often higher Test and Retain Acc. under the same budget. This is due to its ability to restore utility quickly while maintaining strong selectivity in forgetting.
>
> **Changes to the paper:** new paragraph in Section 5.1, new Table 2, extended results in Appendix D.5 and Table 11.

---

> ### Author Response · Authors · 2025-11-27
> **Response to Reviewer ST2i (Weakness/Question 2)**
>
> **W2 – *“Is it possible to provide any theoretical insights or bounds on the divergence between the model produced by FedQUIT and the ideal, retrained-from-scratch model?”***
>
> We thank the reviewer for the insightful question. In the revised version, we extended our theoretical analysis and added **Theorem 3** in **Appendix C.5**, which directly addresses this concern.
>
> The paper already included two core theoretical results:
>
> 1. **Lemma 2**, which shows that FedQUIT introduces a *bounded model perturbation* on the original global model during unlearning;
> 2. **Theorem 1**, which establishes that, after unlearning, FedAvg still converges at the *same rate* as in the standard (unperturbed) setting.
>
> To further connect FedQUIT to the gold-standard retraining baseline, the new **Theorem 3** demonstrates that, under standard assumptions in FL theory, the divergence between the FedQUIT unlearned model and the ideal “retrain-from-scratch” model is **governed precisely by this bounded perturbation** from Lemma 2. Moreover, we show that this gap **decreases along the recovery rounds**, providing a formal characterization of how FedQUIT approaches the retrained solution.
>
> Formal statements and complete proofs are provided in Appendix C.5. We also added a forward reference to this result in Section 3 of the main paper.
>
> **Changes:** Added Appendix C.5 (Theorem 3), and updated Section 3 accordingly.

---

### Official Review · Reviewer_SXnp · 2025-10-28

**Soundness:** 3
**Presentation:** 2
**Contribution:** 2
**Rating:** 4
**Confidence:** 4

**Summary:**

This paper proposes FedQUIT, a novel on-device federated unlearning (FU) algorithm designed to address the "right to be forgotten" in federated learning (FL) while preserving model generalization. FedQUIT adopts a teacher-student framework: a quasi-competent virtual teacher is constructed by modifying the output of the current global model (penalizing the true-class logit of forget data to induce forgetting, while preserving non-true class output relationships to retain useful knowledge), and the local model (student) mimics the teacher via knowledge distillation (minimizing KL divergence). Evaluations across CIFAR-10, CIFAR-100, and CUB-200 datasets (both IID and non-IID distributions) with ResNet-18 and MiT-B0 models show that FedQUIT outperforms six state-of-the-art FU baselines in unlearning efficacy (closer to the "Retrain" gold standard) and significantly reduces communication (15–60×) and computational (15–60×) overhead compared to retraining from scratch.

**Strengths:**

+ The idea proposed in this paper is straightforward and easy to understand. FedQUIT performs unlearning directly on the requesting client’s device in a single round, eliminating the need for multi-round client-server coordination (a flaw of MoDe and FedOSD) and reducing latency.
+ Unlike methods like FedEraser (relying on historical updates) and Wu et al. (2022b) (relying on proxy data), FedQUIT only uses the current global model and local forget data, avoiding privacy and storage risks from historical data.
+ The paper provides rigorous theoretical analyses, including proofs that FedQUIT induces a controlled forgetting signal (Lemma 1), bounded parameter perturbation (Lemma 2), and maintains FedAvg convergence guarantees after unlearning (Theorem 1).

**Weaknesses:**

- It is still unclear what the intuition is behind the whole framework design, and why only a knowledge distillation loss can be used to achieve the goal of both efficiency and preserving performance.

- While the paper mentions extending FedQUIT to sample unlearning, experiments only cover 50% forget data in a single setting (CIFAR-100), lacking validation on varying forget data ratios (e.g., 10%, 30%) or other datasets.

- It seems that only a marginal performance gain has been obtained from the designed framework, i.e, 1~3%. Furthermore, in Table 1, many of the results of FedQUIT were highlighted, but they are not always the best ones, e.g., CIFAR-10, Non-IID, ResNet-18, E=1, Retain Acc. (88.8%). Please check and clarify.
​
- Evaluations use a single GPU (NVIDIA RTX A5000) and do not test FedQUIT on resource-constrained devices (e.g., smartphones or other devices), where its "on-device" advantage is most critical.

**Questions:**

1. How does FedQUIT perform when multiple clients simultaneously request unlearning? The current design focuses on single-client unlearning, but real FL systems may face concurrent requests—does it maintain efficacy/efficiency under such scenarios?

2. For highly imbalanced forget data (e.g., 1% vs. 90% of a client’s data), does the default $\(v=min_c z_i^g\)$ still work, or is a dynamic v-adjustment mechanism needed?

3. Can FedQUIT be extended to other FL tasks beyond image classification (e.g., NLP or time-series prediction), and what modifications would be required for non-classification model outputs?

4. In cross-silo FL (where clients are organizations with large datasets), does FedQUIT’s 1-round unlearning still outperform baselines, or does dataset size increase its local computational cost significantly?

---

> ### Author Response · Authors · 2025-11-22
> **Response to Reviewer SXnp (Weakness 1, Weakness 3)**
>
> **W1. Unclear intuition.**
>
> We thank the reviewer for the helpful feedback. To address this concern, we have revised the paragraph *“Rationale and Intuition’’* in **Section 3** to more clearly articulate the motivation behind our design. We also summarize the key intuition here.
>
> Our framework is grounded in two observations:
>
> (1) **High-confidence predictions,** where the true-class logit significantly dominates, are strong indicators of memorization (Song & Mittal, 2019).
>
> (2) **Preserving the inter-class structure** is essential for maintaining model utility (Hinton et al., 2015).
>
> Our virtual teacher integrates these insights by (i) **reducing confidence on the true label** (lowering the corresponding logit) to induce forgetting, while (ii) **preserving the geometry among non-true classes**, ensuring the student retains information necessary for utility recovery. Crucially, this transformation is performed locally at the client, using a modified version of the current global model as the teacher. This enables one-round, communication-efficient federated unlearning without requiring access to retain data.
>
> **Why a KD-only loss is appropriate.** Adding a cross-entropy (CE) term on hard labels together with the KD loss would introduce conflicting objectives: CE would push the student to reinforce the true class for forget samples, whereas the virtual teacher is deliberately constructed to penalize confidence in that class. Using a KD-only objective avoids this conflict and aligns the training objective with the intended forgetting behavior.
>
> **Ablation on KD Loss.** We also refer the reviewer to Appendix D.7, where we ablate the distillation losses used in Eq. (2): KL divergence (default), mean squared error (MSE), categorical CE with soft targets, and cosine similarity. As shown in Table 15, **effective retention requires preserving the relative geometry among non-true classes** (their ratios and ordering). KL and soft-target CE enforce this directly on the probability simplex, while MSE does not respect this structure and instead averages modes, which weakens forgetting and degrades utility recovery (consistent with our analysis in Section 5.3).
>
> Overall, these ablations support our intuition and justify why a **geometry-preserving KD objective** is the appropriate choice for local federated unlearning.
>
> **References**
>
> Song and Mittal, Systematic evaluation of privacy risks of machine learning models, USENIX Security, 2021.
>
> Hinton et al., Distilling the knowledge in a neural network, arXiv:1503.02531, 2015.
>
> ---
>
> **W3.** *“It seems that only a marginal performance gain has been obtained... results of FedQUIT were highlighted, but they are not always the best ones. Please check and clarify.”*
>
> We thank the reviewer for this observation. Below, we first clarify that in Table 1 we highlight the method with the **smallest absolute gap to Retrain**. Then, we also provide new experiments with **fixed communication budget** where comparing raw Test/Retain Accuracy (higher is better) is more natural.
>
> **Lower absolute gap with Retrain.**  In federated unlearning, the objective is to match as closely as possible the behavior of the *Retrain* model, which represents the ideal state where the forget data was never used. Therefore, **smaller absolute gaps to Retrain** directly indicate better unlearning performance. This criterion is standard in the FU literature (e.g., Khalil et al., 2025; Romandini et al., 2024). In Table 1, highlighted values correspond to the **closest match to Retrain (lower absolute gap)**, not to the highest raw accuracies. Similarities in Test and Retain Acc. across methods are expected because all methods are evaluated after utility recovery, so these metrics do not distinguish selective forgetting. The differences emerge in unlearning-specific metrics (Forget Accuracy and MIA scores), where FedQUIT consistently shows smaller gaps and, in addition, delivers the largest efficiency gains.
>
> **Fixed-budget evaluation (higher Test/Retain Acc. is better).** We added a fixed-budget comparison where baselines are allocated 5%, 10%, 20%, 25%, and 50% of the communication cost of full retraining. This analysis appears in Section 5.1 (Table 2), with full results in Appendix D.5 and Table 11. In this setting, especially for larger budgets (>10%), **higher Test/Retain Accuracy** reflects more effective recovery. Across all budgets, FedQUIT consistently outperforms baselines, achieving higher Test and Retain Accuracy under the same communication budget and lower deltas in Forget Accuracy.
>
> **References**
>
> Khalil et al., Not: Federated unlearning via weight negation. In Proceedings of the Computer Vision and Pattern Recognition Conference, 2025.
>
> Romandini et al., Federated unlearning: A survey on methods, design guidelines, and evaluation metrics. IEEE Transactions
> on Neural Networks and Learning Systems, 2024.

---

> ### Author Response · Authors · 2025-11-22
> **Response to Reviewer SXnp (Question 2, Question 4)**
>
> **Q2. Dynamic $v$-adjustment mechanism under imbalanced or extreme forget ratios.**
>
> We thank the reviewer for the helpful question. We first clarify that FedQUIT, in its default form, does not rely on a fixed $v$ and then provide new results with extreme forget ratios.
>
> The default rule in Section 3
>
> $v_i = \min_c z^g_{i,c}$
>
> already gives a per-sample, data-adaptive adjustment. For each forget sample $x_i$ with true label $y_i$, we take the teacher logits $z^g_i$ and build the virtual teacher logits $z^{\text{virt}}_i$
> by just **replacing the true-class logit with the smallest logit among all classes**:
>
> $z_i^{virt}  = v_i = \min_c z^g_{i, c}$
>
> **This adaptive rule is used in all experiments.** Section 5.4 shows that even the best tuned fixed-$v$ alternatives do not outperform it. Across all settings, the above adaptive $v_i$ achieves the best or second-best performance relative to all baselines. For this reason, no additional $v$-adjustment mechanism is needed.
>
> ---
>
> **New experiments on imbalanced forget ratios.**
>
> We **added new sample-unlearning experiments** beyond the 50% setting, and we now unlearn:
>
> - **1% statistically atypical samples** (1% of local data composed of the samples from the least-represented local classes), and
> - **10% of random samples.**
>
> We consider both IID and Non-IID scenarios.
>
> New **Table 4** reports the most extreme case (1%, Non-IID case); complete post-unlearning and post-recovery results for all new settings in **Appendix D.6** (Tables 12 and 13). Discussion of new results appear in a **new paragraph in Sec. 5.1**.
>
> **Findings.** Even in the extreme 1% case, our default adaptive-$v$ FedQUIT consistently outperforms all baselines: lower degradation immediately after unlearning, stronger forgetting after recovery, and $\approx 4\times$ lower cost. Similar trends hold for the 10% and 50% percent settings. These results confirm that the default data-adaptive rule already provides the theoretical and empirical behavior needed to handle very small as well as large forget sets.
>
> **Changes to the paper:** added Table 4, updated Section 5.1, expanded Appendix D.6.
>
> ---
> **Q4 (first part). Cross-silo FL with large per-client datasets.**
>
> We thank the reviewer for raising this point. In the revised version of the paper, our experimental design covers two complementary regimes: (i) a limited number of clients (10) with large local datasets (for CIFAR-100, each client holds approximately 5,000 samples), and (ii) a larger number of clients (100) with smaller local datasets (approximately 500 samples per client for CIFAR-100, and approximately 50 for CUB-200). These two settings allow us to evaluate both the effect of large and small per-client datasets on FedQUIT’s one-round unlearning.
>
> The regime with **a limited number of clients and large per-client datasets** is most relevant to the reviewer’s cross-silo scenario, where each participant holds a substantial fraction of the global dataset. In our CIFAR-100 experiments with 10 clients, each client indeed stores approximately 5,000 samples (more imbalanced in non-IID case). Also for this setting, FedQUIT performs **exactly one local unlearning round**, consisting of a single pass over the forget set. The results in Table 1 show that FedQUIT (i) provides strong selective forgetting, and (ii) significantly reduces cumulative communication and computation with respect to most or all other baselines.
>
> **Q4 (second part). Does dataset size increase the local computational cost significantly?**
>
> The computational cost of FedQUIT scales linearly with the amount of forget data, and for client-unlearning it is approximately equal to running **one standard local FedAvg round** (same batch size, single epoch). By contrast:
>
> - FedOSD (Pan et al.) and MoDe (Zhao et al.) require several unlearning rounds, leading to higher cost (Table 3 and Appendix Table 10).
> - FedAU (Gu et al.) trains an auxiliary module every round and often fails to unlearn precisely in near-IID regimes (Table 1).
> - NoT (Khalil et al.) has no client-side cost but does not deliver selective unlearning and needs long recovery.
> - PGA (Halimi et al.) requires more local epochs, increasing computation.
>
> Overall, our 10-client experiments already resembles a cross-silo setting with relatively large per-client datasets, and consistently show that FedQUIT’s **single-round unlearning** is both effective and more efficient than all baselines. We would be glad to evaluate any additional cross-silo configuration the reviewer suggests.
>
> **References**
>
> Halimi et al., Federated Unlearning, arXiv:2207.05521, 2022.
> Gu et al., Unlearning during learning: an efficient federated machine unlearning method, IJCAI, 2024.
>
> Zhao et al., Federated unlearning with momentum degradation. IEEE Internet of Things Journal, 2023.
>
> Pan et al., Federated Unlearning with Gradient Descent and Conflict Mitigation, AAAI 2025.
>
> Khalil et al., Not: Federated unlearning via weight negation, CVPR, 2025.

---

> ### Author Response · Authors · 2025-11-23
> **Response to Reviewer SXnp (Question 1, Weakness 4)**
>
> **Q1. Multiple Unlearning Requests.**
>
> We thank the reviewer for this request, which allows us to provide a useful explicit extension. In the revised manuscript, we include both the algorithmic procedure to handle multiple simultaneous unlearning requests and the corresponding empirical results for two different settings (Appendix D.12), and we now explicitly reference this extension in Section 5 of the main paper.
>
> **Algorithm.** FedQUIT also supports the case where multiple clients request unlearning simultaneously. Let $\mathcal{U} \subseteq \{1,\ldots,U\}$ be the set of requesting clients, each aiming to remove its local forget subset $D_u^{\mathrm{forget}}$.
> In a single unlearning round, every $u \in \mathcal{U}$ receives the current global model $w_t$, applies FedQUIT locally (Algorithm 1) to obtain an unlearned snapshot $w_t^{\bar u}$, and returns it to the server.
> The server then constructs a sanitized global model via a FedAvg-style aggregation:
> $
> w_t^{\bar{\mathcal{U}}}
> = \sum_{u \in \mathcal{U}}
> \frac{n_u}{N_{\mathcal{U}}} w_t^{\bar u},
> $
> with $n_u = |D_u^{\mathrm{forget}}|$ and $N_{\mathcal{U}} = \sum_{u\in\mathcal{U}} n_u.$
> Training resumes from $w_t^{\bar{\mathcal{U}}}$ while excluding all clients in $\mathcal{U}$ from further participation.
>
> **New Experimental Results.** We provide results for multiple simultaneous unlearning requests. We consider CIFAR-100 (Non-IID) with either ResNet-18 or MiT-B0 and 10 clients, where two clients request unlearning simultaneously. We average results over five random pairs of clients and report mean and standard deviation. We do not include PGA (Halimi et al., 2022)  and FedOSD (Pan et al., 2025), see below.
>
> **Findings.** FedQUIT achieves the best retained performance immediately after unlearning (higher Test and Retain accuracy), provides the most accurate post-recovery unlearning metrics, and outperforms all baselines in terms of efficiency (lower communication and computation costs to recover utility).
>
> **Excluded Baselines and Adaptations.** FedOSD is inherently designed for **single-client** unlearning: its optimization and guarantees rely on aligning with the gradient of **one** target client while enforcing orthogonality to all others. Extending this one-versus-rest formulation to multiple simultaneous forgetters would require redefining the update rule and the orthogonality constraints, something not addressed in the original paper and not straightforward. PGA requires constructing a reference model without the last contribution of the requesting client; how to generalize this mechanism to multiple simultaneous requests is not explicit in the original paper. We adapt MoDe (Zhao et al., 2023) by providing all unlearning clients with the same degradation model and applying memory guidance in parallel.
>
> **Changes to the paper.** We added a dedicated section in **Appendix D.12** as well as new **Appendix Table 20** and **Appendix Table 21**, and we included an **explicit reference to these experiments in Section 5**, where we introduce the research question (point 1).
>
> **References**
>
> Halimi et al., Federated Unlearning, arXiv:2207.05521, 2022.
>
> Pan et al., Federated Unlearning with Gradient Descent and Conflict Mitigation, AAAI 2025.
>
> Zhao et al., Federated unlearning with momentum degradation. IEEE Internet of Things Journal, 2023.
>
> ---
> **W4. Evaluations on resource-constrained devices.**
>
> We thank the reviewer for the comment. Evaluating on real smartphones or severely resource-constrained devices is indeed valuable future work. Our current experiments follow the standard FU evaluation protocol used in prior work (e.g., Gu et al., 2024; Khalil et al., 2025; Halimi et al., 2022; Pan et al., 2025), and none of these baselines provide on-device measurements either.
>
> FedQUIT’s on-device computation is intentionally lightweight: a client performs one forward pass on its forget data, constructs the virtual teacher by modifying logits, and runs a single epoch of distillation, **approximately matching** the cost of a standard FedAvg round, and even less when only part of the data is forgotten.
>
> To further assess scalability toward cross-device FL, we added experiments with **100 clients and partial participation** (CIFAR-100/ResNet-18 and CUB-200/MiT-B0), now included in Table 1 and Tables 9–10 in the Appendix. These confirm the trends from the 10-client setting: strong unlearning quality and large efficiency gains over baselines.
>
> Finally, we softened the Appendix summary (end of Appendix) to avoid overclaiming about constrained-device scenarios, since we do not evaluate directly on real hardware.
>
> **References**
>
> Khalil et al. Not: Federated unlearning via weight negation, CVPR, 2025.
>
> Gu et al., Unlearning during learning: an efficient federated machine unlearning method, IJCAI, 2024
>
> Halimi et al., Federated Unlearning, arXiv:2207.05521, 2022.
>
> Pan et al., Federated Unlearning with Gradient Descent and Conflict Mitigation, AAAI 2025.

---

> ### Author Response · Authors · 2025-11-26
> **Response to Reviewer SXnp (Question 3)**
>
> ***Q3. “Can FedQUIT be extended to other FL tasks beyond image classification (e.g., NLP or time-series prediction), and what modifications would be required for non-classification model outputs?”***
>
> We thank the reviewer for the observation. FedQUIT relies on manipulating the teacher’s logits and applying a distillation loss (e.g., KL) on the unlearned client’s data. This mechanism is directly applicable to any task whose model outputs a categorical distribution, such as next-token prediction in NLP or multi-class time-series classification.
>
> To clarify this point, **we have added a new set of experiments on a *next-token prediction* task using the Tiny Shakespeare corpus** (Karpathy, 2015), with 80 clients and partial participation.
>
> **Findings.**
>
> New Table 22 (Appendix D.13) reports the results for this LSTM-based Tiny Shakespeare setting. Consistent with all previously evaluated configurations, FedQUIT remains the most efficient baseline in terms of communication and computation required to recover full model utility. This behavior stems from the properties of the FedQUIT model immediately after unlearning: it achieves *selective forgetting*, displaying only a mild degradation in test and retain accuracy while causing a substantial drop in forget accuracy. This selective degradation makes the subsequent recovery phase significantly lighter. Moreover, FedQUIT achieves the most accurate forgetting among all baselines, showing the smallest gap from the *Retrain* forget accuracy and thus confirming its strong ability to remove target information while rapidly restoring overall utility.
>
> **Extension to non-classification tasks.**
>
> For tasks where the model does not produce logits (e.g., continuous regression or forecasting), FedQUIT would require adapting the teacher–student objective by defining degraded continuous targets and replacing the KL term with a regression-based loss. While conceptually feasible, this extension would require designing an analogue of  “logit manipulation” for continuous outputs, which is beyond the scope of the present work.
>
> **Changes to the paper.** We added Appendix D.13 with the experimental design and results for the Tiny Shakespeare setting, added Table 22 with numerical results, and included an explicit reference to these new experiments in Section 4 of the main paper.
>
> **References**
>
> Andrej Karpathy. char-rnn. https://github.com/karpathy/char-rnn, 2015.

---

### Official Review · Reviewer_2DZF · 2025-11-01

**Soundness:** 3
**Presentation:** 3
**Contribution:** 3
**Rating:** 4
**Confidence:** 4

**Summary:**

This paper proposes FedQUIT, an on-device federated unlearning framework that enables clients to remove their data influence without accessing historical updates or auxiliary datasets. The key idea is to use a quasi-competent virtual teacher that penalizes the true-class logit while preserving inter-class relations, achieving efficient forgetting through one-round knowledge distillation. Theoretical analysis proves bounded parameter shifts and preserved convergence under FedAvg.

**Strengths:**

- The proposed quasi-competent teacher is a novel intermediate-level concept, clearly distinguished from prior “incompetent teacher” or “random-label teacher” paradigms. By penalizing the true-class prediction while preserving the inter-class structure, it effectively balances forgetting and knowledge retention.

- The evaluation metrics are well designed and cover both unlearning efficacy and efficiency, offering a comprehensive view of model performance.

- The idea itself is interesting and provides a fresh perspective on on-device federated unlearning through knowledge distillation.

- The theoretical analysis is mathematically sound and aligns well with the algorithm design

**Weaknesses:**

- The authors announce that this paper is in on-device federated learning settings in line 81. However, the experiment results under different numbers of clients are not clear. Could authors provide the results on the settings of more clients.

- The paper claims that FedQUIT is more efficient than MoDe and FedOSD. However, those baselines inherently require more local epochs or communication rounds. To ensure fairness, it would be better to compare all methods under the same computation or communication cost budget and observe their relative performance.

- The writing is generally clear, but the technical novelty needs to be emphasized more strongly. Knowledge distillation has already been used in federated unlearning (e.g., Wu et al., 2022a; 2022b), yet the paper only briefly mentions that those methods rely on historical records. The authors should provide a more detailed comparison highlighting the conceptual and technical differences between FedQUIT and these prior KD-based unlearning approaches.

**Questions:**

- How about the results under different numbers of client settings?

- If the budget is fixed, how about the performance gap?

---

> ### Author Response · Authors · 2025-11-22
> **Response to Reviewer 2DZF (Weakness/Question 1, Weakness/Question 2)**
>
> **W1/Q1 – Experiment with more clients.** We thank the reviewer for the suggestion. Below, we (1) clarify the meaning of “on-device” in our setting and (2) provide additional experiments with 100 clients, as requested.
>
> **1. On-device unlearning.** In our paper, “on-device” refers to where unlearning is executed:
> the FU procedure is performed locally on the client device that issues the request, using only its
> own forget data and without relying on historical information or auxiliary datasets. This notion of
> on-device unlearning is orthogonal to the size of the federation.
>
> **2. Additional experiments with 100 clients.** We added two new settings with 100 clients:
> - **CUB-200, MiT-B0, Non-IID ($\alpha=0.1$)**, fine-tuned from pretrained weights (200 rounds of FedAvg before unlearning).
> - **CIFAR-100, ResNet-18, Non-IID ($\alpha=0.1$)**, trained from scratch (600 rounds of FedAvg before unlearning).
>
>     In both settings we use a participation rate of 0.1.
>
>     The main post-recovery results are now reported in the two new blocks added at the bottom of our updated **Table 1**, and the full corresponding metrics (post-unlearning and post-recovery) are included in **Tables 9–10 in the Appendix**, consistent with the reporting used for all other settings.
>
>
> **Findings.** As shown in the updated Table 1, FedQUIT continues to be the best-performing method across both new settings. It achieves the lowest absolute forget-metric gaps and remains the most efficient method, with approximately **74.1×** (CIFAR-100) and **64.5×** (CUB-200) reductions in computation/communication cost compared to retraining. FedQUIT also significantly outperforms the second-best efficient baseline, with relative gains of **5×** (CUB-200) and **1.72×** (CIFAR-100).
>
> **Changes in the paper.** We updated **Section 4 “Experimental Design”** to describe these new settings. We updated **Table 1** (main paper) and **Tables 9–10** (Appendix) to include the two new 100-client settings and integrated them into the results section. All trends observed in the original submission remain consistent.
>
> ---
> **W2/Q2 - Comparison under Fixed Cost Budget.**
>
> We thank the reviewer for the observation. We agree that methods such as MoDe and FedOSD inherently require more local epochs or communication rounds, and a fixed cost budget can be a meaningful way to compare methods with different cost structures. Below, we (1) clarify what was already accounted for in the original submission, and (2) present new results under fixed computation/communication budgets, as explicitly requested.
>
> **1. Clarification on fairness of cost reporting in the original submission.** Our cumulative communication and computation costs already incorporated the additional rounds required by multi-round unlearning methods such as MoDe and FedOSD. This is reflected in Table 1, Table 3, and Appendix Tables 9 and 10. For example, Table 3 reports the cost of the unlearning phase alone (before any recovery) and shows that MoDe and FedOSD require **two or more orders of magnitude** more communication and computation than FedQUIT. Furthermore, Section D.3 of the appendix details the cost-estimation methodology used for each method to ensure comparability. Thus, the reported efficiency differences already fully account for the inherent cost structure of the baselines.
>
> **2. New comparison under fixed cumulative cost budgets.** To directly address the reviewer’s suggestion, we additionally evaluate all methods under fixed cumulative communication budgets (the trends for computation are very similar, as emerging in Table 1). We consider five budgets corresponding to **5%, 10%, 20%, 25%**, 5**0%** of the total communication cost required for retraining the model from scratch. **We added a dedicated paragraph in Section 5.1, with numerical results in new Table 2, and results in Appendix D.5 and Table 11.**
>
> We report results for CIFAR-100 with ResNet-18 and 10 clients, under both IID and non-IID settings.
>
> Note that, as shown in Table 1 under the efficiency metrics, the 10% and 20% budget points occur significantly after most methods have already fully recovered their utility.
> Therefore, budgets at 10% and 20% occur well after FedQUIT has fully recovered utility. This analysis aims to determine whether forgetting remains effective later in the recovery phase, ruling out the possibility that, with budgets larger than strictly necessary to regain retraining-level utility, the forgetting process might reverse.
>
> **Findings.** As shown in Table 2 and Appendix Table 11, FedQUIT consistently outperforms the other baselines, achieving both lower deltas in Forget Acc. and, often, higher Test and Retain Acc. values under the same communication budget. This advantage stems from FedQUIT’s ability to rapidly recover model utility while remaining highly selective in forgetting.
>
> **Changes to the paper:** new paragraph in Section 5.1, new Table 2, and extended results in Appendix D.5 and Table 11.

---

> ### Author Response · Authors · 2025-11-22
> **Response to Reviewer 2DZF (Weakness 3)**
>
> **W3 – Emphasizing Technical Novelty vs. KD-based FU Approaches.**
>
> We thank the reviewer for pointing out the need to better highlight the conceptual novelty of FedQUIT relative to prior FU methods that employ knowledge distillation (KD) (e.g., Wu et al., 2022a; 2023). **We have strengthened the Related Work section (Section 6) accordingly**. Here, we summarize the key distinctions.
>
> **(1) Prior KD-based FU methods rely on historical updates and public/auxiliary data.**
>
> Methods such as Wu et al. (2022a; 2023) perform unlearning by first subtracting the historical updates associated with the target client and then distilling the knowledge of the pre-unlearning global model using a public dataset. This paradigm implicitly assumes the existence of public data that is (i) semantically aligned with the federation and (ii) sufficiently balanced across classes (conditions that have been shown to be critical for KD effectiveness (Nayak et al., 2021) but are often unrealistic in privacy-sensitive FL deployments).
>
> **(2) FedQUIT is fully on-device and does not rely on any public, auxiliary, or historical data.**
>
> FedQUIT constructs a virtual teacher directly from the current global model, requiring no historical client updates and no public dataset. KD is performed solely on the target client’s forget data, entirely on-device, during a single unlearning round. This avoids the feasibility, privacy, and distribution-shift issues inherent in the use of external KD datasets.
>
> **(3) Conceptual difference: KD as supervision perturbation vs. KD for fast recovery.**
>
> Whereas Wu et al. (2022a; 2023) use KD on external data primarily as a recovery accelerator, FedQUIT uses KD in a fundamentally different way: as a controlled supervision-perturbation mechanism that penalizes the true-class output while preserving non-true logits. This is a substantially different technical formulation from recovery-oriented distillation and directly targets the forgetting signal.
>
> **Changes to the paper. These conceptual and technical distinctions are now explicitly articulated in Section 6 of the revised manuscript.** An extended discussion on the key practical requirements and limits of related work can be found in Appendix E.
>
>
> Wu et al., Federated unlearning with knowledge distillation, arXiv preprint arXiv:2201.09441, 2022a.
>
> Wu et al., Unlearning backdoor attacks in federated learning, In ICLR 2023 Workshop on Backdoor Attacks and Defenses in Machine Learning, 2023.
>
> Nayak et al., Effectiveness of Arbitrary Transfer Sets for Data-free Knowledge Distillation. In Proc. of the IEEE/CVF Winter Conference on Applications of Computer Vision, pp. 1430–1438, 2021.
>
> Liu et al., Federaser: Enabling efficient client-level data removal from federated learning models. In 2021 IEEE/ACM 29th International Symposium on Quality of Service (IWQOS), pp. 1–10, 2021.
>
> Li et al., The right to be forgotten in federated learning: An efficient realization with rapid retraining. In IEEE INFOCOM 2022-IEEE Conference on Computer Communications, pp. 1749–1758. IEEE, 2022.

---

### Author Response · Authors · 2025-11-27
**Summary of Changes**

We thank the Area Chair and all Reviewers for their constructive feedback. Below we summarize the main revisions, with locations in the revised manuscript (main paper and appendix). For each major or minor change we track here, we report in parenthesis the Reviewer that requested the change and the question (Q) or weaknesses (W) we addressed.

**Major Changes**

- **Experiments with more clients** (Reviewer **2DZF, Q1**).

    We added two new experimental settings with 100 clients and partial participation.

    **Changes:** updated Section 4 (experimental design); revised numerical results in Table 1 (main paper) and Tables 9–10 (Appendix); discussion expanded in Section 5.1.

- **Multiple simultaneous unlearning requests** (Reviewer **SXnp, Q1**; Reviewer **Eci2, Q5**).

    We added both the algorithmic procedure and the empirical evaluation for handling multiple unlearning requests.

    **Changes:** new subsection in Appendix D.12; results in Tables 20–21 (Appendix); referenced in Section 5 (main paper).

- **Performance under fixed communication budget** (Reviewer **2DZF, Q2**; Reviewer **ST2i, Q6**).

    We added results under five fixed cumulative communication budgets (with analogous trends for computation).

    **Changes:** new paragraph in Section 5.1; numerical results in new Table 2 (main paper); extended results in Appendix D.5 and Table 11 (Appendix).

- **Sample unlearning with highly imbalanced or atypical forget data** (Reviewer **SXnp, Q2**; Reviewer **ST2i, Q3**).

    We extended sample-unlearning experiments to include: (i) 1% of local data from the least-represented classes (statistically atypical), and (ii) 10% of local data sampled uniformly at random.

    **Changes:** new Table 4; additional discussion in Section 5.1; extended results in Tables 12–13 (Appendix); updated Appendix D.6.

- **Inclusion of MoDe and FedOSD in sample-unlearning experiments** (Reviewer **Eci2, Q3**).

    **Changes:** added baselines to Table 4 (main paper) and Tables 12–13 (Appendix); extended Appendix D.6.

- **FedQUIT vs. simpler random-label supervision** (Reviewer **ST2i, Q1**).

    We framed random-label supervision within the same teacher-based perturbation family and added it as a baseline in our ablation study.

    **Changes:** expanded Section 6 (related work); detailed analysis in Appendix A.1; added “Random” baseline in Table 5; additional discussion in Section 5.3.

- **FedQUIT vs. unconstrained gradient ascent (GA)** (Reviewer **ST2i, Q4**).

    We added an unconstrained GA baseline in two settings (GA fails to induce meaningful forgetting).

    **Changes:** referenced in Section 4; results in Tables 9–10; expanded discussion in Appendix D.2.

- **Temperature-scaled teachers** (Reviewer **ST2i, Q5**; Reviewer **Eci2, Q2**).

    We added temperature-scaling experiments and linked them explicitly to our teacher-structure ablation.

    **Changes:** updated Section 5.3; extended experiments and discussion in Appendix D.8; new Table 16.

- **Experiments beyond image classification** (Reviewer **SXnp, Q3**). We added experiments on a next-token prediction task using the Tiny Shakespeare corpus, and the results align with the trends observed in our original settings.
- **Changes:** Added Appendix D.13 with the experimental design, added Table 22 with numerical results, and included an explicit reference to these experiments in Section 4 of the main paper.
- **Technical novelty relative to KD-based FU methods** (Reviewer **2DZF, W3**).

    We clarified the specific technical differences from history-based KD approaches, highlighting FedQUIT’s unique design.

    **Changes:** expanded Related Work in Section 6.

- **Theoretical insights on the divergence between the FedQUIT unlearned model and the retrained model** (Reviewer **ST2i, Q2**). We extended the theoretical analysis and showed that the divergence between the FedQUIT unlearned model and an ideal retrained model is governed by the bounded perturbation introduced during unlearning, and this gap decreases over the recovery rounds.

    **Changes**: Added Section C.5 (Theorem 3), added reference in Section 3 (main paper).
- **Explicit Non-IID assumption for Theorem 1** (Reviewer **Eci2, W2**).

    **Changes:** new Assumption (A7) in Appendix C.2; updated theorem statement.

- **Clarified the “Rationale and Intuition”** (Reviewer **SXnp, W1**).

    We rewrote the paragraph in Section 3 to more clearly articulate the motivation behind FedQUIT’s design principles.


**Minor Changes**

- **Softened claims about constrained-device scenarios** (Reviewer **Eci2, Q4**).

    **Changes:** We revised the conclusion of Appendix E to avoid overclaiming.

- **Improved visibility of assumptions.** We slightly reworked the presentation of assumptions in Appendix C to make (A5)–(A8) more prominent and easier to find.

---

### Meta-Review · Area_Chair_7Ahw · 2026-01-05

**Summary:**

1. The proposed framework is largely an incremental adaptation of established knowledge distillation techniques. But the authors overstate its novelty.
2. It is not clear how a simple KD loss modification can enhance performance in multiple aspects.
3. The authors announce that this paper is in on-device federated learning settings, but lack experiments on more clients or resource-constrained devices.
4. The performance gain seems to be only marginal.

**Reviewer Concerns:**

While point 3 was addressed, points 1, 2, and 4 are still outstanding.

**Reviewer Scores:**

Remain the same

---

### Decision · Program_Chairs · 2026-01-26

Reject